# Mitochondrial fission controls astrocyte morphogenesis and organization in the cortex

Maria Pia Rodriguez Salazar[1], Sprihaa Kolanukuduru[1,2], Valentina Ramirez[1,3], Boyu Lyu[4], Gracie Manigault[1], Gabrielle Sejourne[1], Hiromi Sesaki[5], Guoqiang Yu[6,7], and Cagla Eroglu[1,8,9,10]

**Dysfunctional mitochondrial dynamics are a hallmark of devastating neurodevelopmental disorders such as childhood refractory epilepsy. However, the role of glial mitochondria in proper brain development is not well understood. We show that astrocyte mitochondria undergo extensive fission while populating astrocyte distal branches during postnatal cortical development. Loss of mitochondrial fission regulator, dynamin-related protein 1 (Drp1), decreases mitochondrial localization to distal astrocyte processes, and this mitochondrial mislocalization reduces astrocyte morphological complexity. Functionally, astrocyte-specific conditional deletion of Drp1 induces astrocyte reactivity and disrupts astrocyte organization in the cortex. These morphological and organizational deficits are accompanied by loss of perisynaptic astrocyte process (PAP) proteins such as gap junction protein connexin 43. These findings uncover a crucial role for mitochondrial fission in coordinating astrocytic morphogenesis and organization, revealing the regulation of astrocytic mitochondrial dynamics as a critical step in neurodevelopment.**

## Introduction

Mitochondria are dynamic organelles that move, divide, and fuse in response to changing cellular states (Friedman and Nunnari, 2014; Youle and van der Bliek, 2012). Localized functions of mitochondria within cells are governed by fusion, fission, and trafficking processes, collectively referred to as mitochondrial dynamics (Ni et al., 2015). Mitochondrial dynamics are particularly important for the function and development of highly compartmentalized cells such as neurons, where mitochondria within distal compartments perform key functions including ion buffering, energy production, and local protein translation (Billups and Forsythe, 2002; López-Doménech and Kittler, 2023; Rangaraju et al., 2019). Furthermore, distal mitochondria regulate cellular morphology of neurons through focalized ATP generation for actin polymerization at growth cones, and disruptions in mitochondrial dynamics result in stunted neurite growth (Courchet et al., 2013; Smith and Gallo, 2018; Steketee et al., 2012). Indeed, mutations in the genes that control the mitochondrial dynamics' machinery cause severe neurodevelopmental disorders, such as forms of childhood refractory epilepsy (Abati et al., 2022; Vanstone et al., 2016).

Despite the clear importance of mitochondrial dynamics in brain development, the role of this organelle in non-neuronal brain cells is less well understood. In particular, astrocytes are the most abundant glial cells in the brain and have highly ramified processes that infiltrate the neuropil in the brain parenchyma (Verkhratsky et al., 2017). These fine astrocyte processes interact with synapses, vasculature, and other glia (Allen and Lyons, 2018), where they regulate synapse formation, neurotransmitter and ion buffering, and maintain the blood–brain barrier (Alvarez et al., 2013; Chung et al., 2015). Intriguingly, astrocyte peripheral processes are loaded with mitochondria at higher densities than the surrounding neuropil (Lovatt et al., 2007), and their mitochondria are recruited to perisynapses in response to neuronal activity (Stephen et al., 2015), linking astrocyte mitochondrial dynamics and synapse function. Indeed, mature astrocyte mitochondria localize to functional microdomains in their arbors, where mitochondria regulate distal calcium dynamics to provide metabolic support in response to neuronal activity (Agarwal et al., 2017). Furthermore, recent work established that immature astrocytes have a high oxidative capacity and require mitochondrial biogenesis for proper

[1]The Department of Cell Biology, Duke University Medical Center, Durham, NC, USA; [2]The Department of Psychology and Neuroscience, University of North Carolina at Chapel Hill, Chapel Hill, NC, USA; [3]The Department of Psychology and Neuroscience, Duke University, Durham, NC, USA; [4]Bradley Department of Electrical and Computer Engineering, Virginia Polytechnic Institute and State University, Arlington, VA, USA; [5]Department of Cell Biology, John Hopkins University School of Medicine, Baltimore, MD, USA; [6]Department of Automation, Tsinghua University, Beijing, China; [7]IDG/McGovern Institute for Brain Research, Tsinghua University, Beijing, China; [8]Aligning Science Across Parkinson's (ASAP) Collaborative Research Network, Chevy Chase, MD, USA; [9]The Department of Neurobiology, Duke University Medical Center, Durham, NC, USA; [10]Howard Hughes Medical Institute, Duke University Medical Center, Durham, NC, USA.

Correspondence to Cagla Eroglu: cagla.eroglu@duke.edu.

morphogenesis and synapse formation (Zehnder et al., 2021). However, the role of mitochondrial dynamics in astrocyte development is unknown.

Beyond their morphological complexity, astrocytes further regulate brain homeostasis by tiling the entire brain parenchyma in evenly dispersed, nonoverlapping domains. This organization enables astrocytes to form an extensive network that facilitates long-distance communication via gap junction coupling between the distal processes of neighboring astrocytes (Bushong et al., 2002; Giaume et al., 2010). Tiling is absent in immature astrocytes, becomes established as astrocytes mature, and is regulated by the same machinery that ensures proper gap junction coupling (Baldwin et al., 2021). Connexin 43 (Cx43) is the most abundant gap junction protein in the brain and is almost exclusively expressed in astrocytes (Rash et al., 2001). Beyond its gap junction roles, Cx43 has cell adhesion functions that drive astrocyte morphogenesis and migration (Kameritsch et al., 2012; Lagos-Cabré et al., 2019). Importantly, both tiling and gap junction coupling are disrupted in many forms of brain injury and disease, such as epilepsy and traumatic brain injury, indicating the importance of the proper establishment of astrocyte networks in brain homeostasis (Cheung et al., 2023; Clasadonte et al., 2017; Hösli et al., 2022; Oberheim et al., 2008).

Here, we investigated the role of mitochondrial dynamics in astrocyte morphogenesis, development, and function. We found that astrocyte mitochondria robustly increase in number and decrease in size to populate distal astrocyte processes during postnatal development, linking mitochondrial fission to astrocyte morphogenesis. Dynamin-related protein 1 (Drp1), encoded by the gene *Dnm1l*, is a highly conserved GTPase that controls mitochondrial fission (Smirnova et al., 2001). Upon mitochondrial division, Drp1 is recruited to the outer mitochondrial membrane to perform GTP hydrolysis-driven scission (Smirnova et al., 2001). Complete loss of Drp1 is embryonically lethal in mice (Wakabayashi et al., 2009b), and mutations in human *DNM1L* are linked to aggressive forms of developmental delay and childhood refractory epilepsy (Fahrner et al., 2016; Liu et al., 2021). Despite the causal link between Drp1 function and proper brain development, the role of Drp1 in astrocyte morphogenesis and development is unknown.

## Results

### Mitochondria increase in number and decrease in size during cortical astrocyte morphogenesis

In mature astrocytes, mitochondria occupy the entirety of the astrocyte arbor (Lovatt et al., 2007), and this mitochondrial dispersion is remodeled in response to injury or disease (Gollihue and Norris, 2020; Motori et al., 2013). However, how the mitochondrial network is established and modified in developing astrocytes is not well understood. How do mitochondrial transport, fission, or fusion dynamics orchestrate the distribution of astrocytic mitochondria during morphogenesis? To address this question, we first investigated how the mitochondrial network transforms during astrocyte maturation to populate the arbors. To do so, we used postnatal astrocyte labeling by electroporation (PALE) (Stogsdill et al., 2017) to

fluorescently label sparse populations of astrocytes in the developing mouse V1 visual cortex. We used mito-EGFP–floxed mice (Agarwal et al., 2017), in which EGFP is targeted to the inner mitochondrial membrane protein cytochrome c oxidase, to label all mitochondria in Cre-expressing cells. We electroporated two plasmids into the ventricles of P0 mice—a pCAG-Cre plasmid to turn on the mitochondrial GFP and an mCherry-CAAX plasmid to label astrocyte membranes. Then using confocal microscopy, we imaged and analyzed astrocytes from layers 2/3, 4, and 5 of the developing visual cortex. This approach allowed us to visualize and quantify astrocyte morphology and their mitochondrial content throughout postnatal cortical development using confocal microscopy and morphometric analyses (Fig. 1 A).

We focused our studies on the first 3 wk of postnatal astrocyte development (P4, P7, P14, and P21, Fig. 1 B), because cortical astrocytes become morphologically mature during this period (Clavreul et al., 2019; Stogsdill et al., 2017). As expected, astrocytes undergo robust growth and elaboration, increasing their overall cell volume ~20-fold between P4 and P21 (Fig. 1 C). Concurrently, we found that total astrocyte mitochondrial volume per cell increased fivefold between P4 and P14 and then remained constant between P14 and P21 (Fig. 1 D). We next quantified total mitochondrial volume per cell normalized to astrocyte cell volume, hereafter referred to as total mitochondrial occupancy. We found total mitochondrial occupancy decreased during development due to the robust increase in astrocyte cell volume (Fig. 1 E). Importantly, astrocyte mitochondria increased in number and decreased in size on average during these first two postnatal weeks (Fig. 1, F and G), indicating mitochondrial division (fission) occurs concurrently with astrocyte morphogenesis.

### Mitochondria occupy fine astrocyte processes concurrently with morphogenesis *in vitro*

The recruitment of mitochondria to the distal edge of developing cell processes in other compartmentalized cells is required for cytoskeletal remodeling and branch outgrowth (Cunniff et al., 2016; Smith and Gallo, 2018). However, the spatiotemporal distribution of mitochondria during astrocyte arborization and whether astrocyte mitochondria are present at the tips of their developing peripheral processes is not known. The highly complex morphology of astrocytes *in vivo* obstructs our ability to track and analyze individual astrocyte processes and their mitochondria. Therefore, we investigated how mitochondria are distributed throughout the astrocyte arbor during morphological growth using an *in vitro* astrocyte–neuron co-culture system. Astrocytes cultured on top of neurons gain a complex morphology compared with astrocytes cultured alone or with non-neuronal cells (Stogsdill et al., 2017). Thus, we used astrocyte–neuron co-cultures to stimulate astrocyte ramification and track how mitochondrial number, size, and distribution change during process elaboration. We isolated and transfected primary cortical rat astrocytes with cytosolic GFP and MitoDsRed constructs to label astrocytes and their mitochondria, respectively. Transfected astrocytes were then plated onto rat cortical neuron monolayers for 4, 12, 24, and 48 h (Fig. 2, A and B).

We developed a MATLAB-based image analysis program, Seg_Astro, to quantify branch number, mitochondrial number,

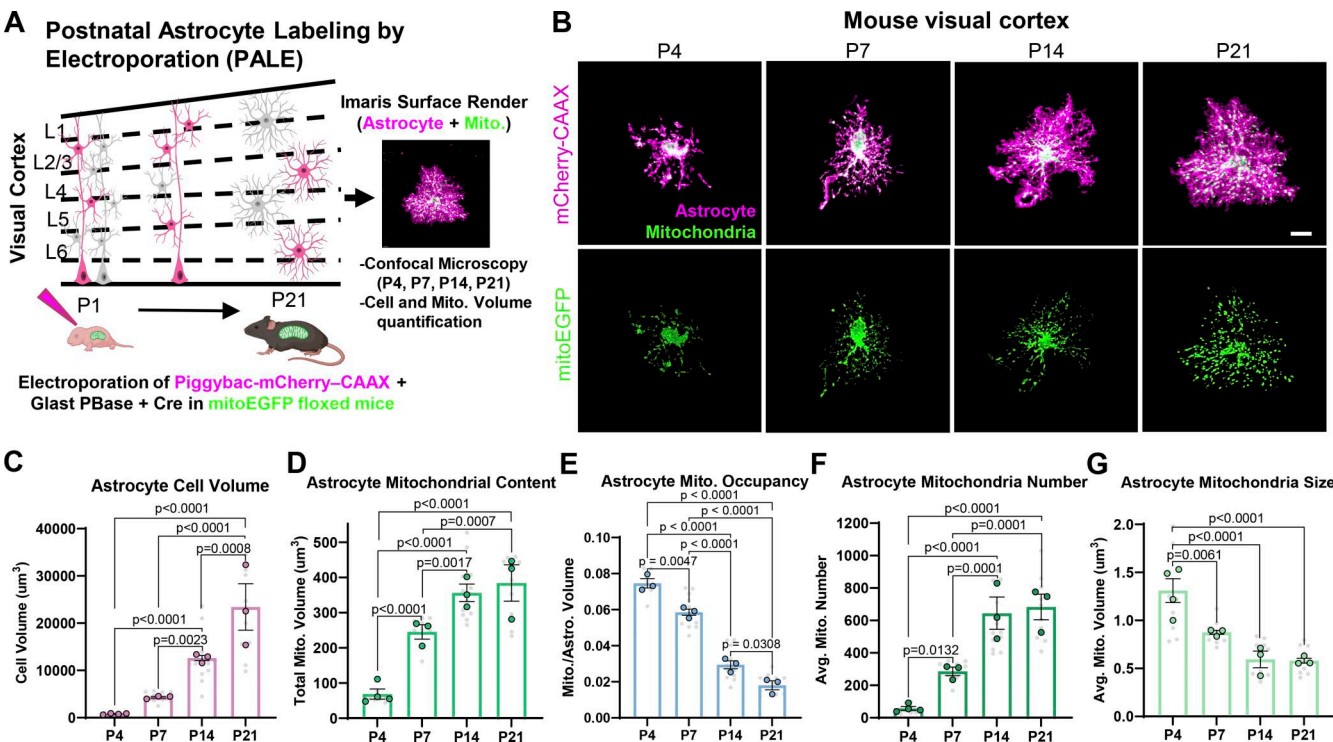

Figure 1. **Mitochondria increase in number and decrease in size during cortical astrocyte morphogenesis *in vivo*. (A)** Schematic of PALE. **(B)** Representative images of V1 astrocytes expressing mCherry-CAAX (magenta, top panels) and their EGFP mitochondria (green, bottom panels) at P4, P7, P14, and P21. **(C–G)** Scale bar: 10 μm. Super-plots for the quantification of (C) astrocyte cell volume, (D) total mitochondrial volume per cell, (E) mitochondrial volume normalized to cell volume, (F) average mitochondrial number per cell, and (G) average mitochondrial size per cell from P4 to P21. *N* = 3–4 male and female mice/time point (large circled data points), *n* = 2–6 cells/mouse, 10–13 cells total/time point (small gray data points). Data are presented as mean ± SEM. Nested one-way ANOVA with Tukey post hoc test.

and mitochondrial size across astrocyte arbors. Seg_Astro uses branch width and branchpoints to determine the astrocyte branch hierarchy and assign four branch types: (1) soma and primary processes, (2) secondary processes, (3) fine processes, and (4) the terminal tips of processes (Fig. 2 C). Seg_Astro then bins mitochondria into these four types of branches and outputs branch number, mitochondrial number, and mitochondrial size per branch type. We excluded mitochondrial measurements from the soma and primary branches of astrocytes, as mitochondria form a dense network in these compartments that cannot be distinguished as discrete mitochondria for number and size quantification.

Seg_astro quantification showed a significant increase across all astrocyte branch types (primary, secondary, fine, and terminal) between 4 and 48 h in culture. Strikingly, astrocyte fine branches had the largest increase in number, approximately fivefold from 4 to 48 h in culture (Fig. 2 D). Intriguingly, mitochondria were present across all astrocyte branch types, including fine and terminal, as early as 4 h and through 48 h in co-culture (Fig. 2 B inset and Fig. 2 E). These results show that mitochondria occupy distal processes during astrocyte branch development *in vitro* and suggest that mitochondrial recruitment to these distal processes may be required for sustained growth of astrocyte arbors.

Our analyses also revealed astrocyte mitochondrial numbers increased across branch types throughout time in culture,

with fine branches housing the largest increase in number of mitochondria from 4 to 48 h in culture compared with other branch types (secondary and terminal) (Fig. 2 E). This unique fine branch mitochondrial number increase correlates with the high increase in fine branch numbers (both approximately fivefold) compared with all other branch types (Fig. 2 D), indicating that the robust increase in astrocyte fine branch number during astrocyte morphogenesis is linearly scaled with increased occupancy of proportionally high mitochondrial numbers. While mitochondrial numbers increased, average mitochondrial size decreased across all astrocyte branch types throughout time in culture except in fine processes, which housed consistently small mitochondria throughout morphogenesis (Fig. 2 F). This result suggested that mitochondrial fission may be required for mitochondrial recruitment to the growing fine/distal processes, as there may be a maximum size threshold for mitochondria to occupy fine astrocyte branches. These data echo and expand upon our *in vivo* findings and implicate a role for mitochondrial division in distal astrocyte process morphogenesis.

### Drp1-induced mitochondrial fission is required for distal astrocyte process formation and mitochondrial localization *in vitro*

Because cortical astrocyte morphogenesis occurred concurrently with extensive mitochondrial fragmentation *in vivo* and *in vitro*, we next investigated the role of mitochondrial fission in

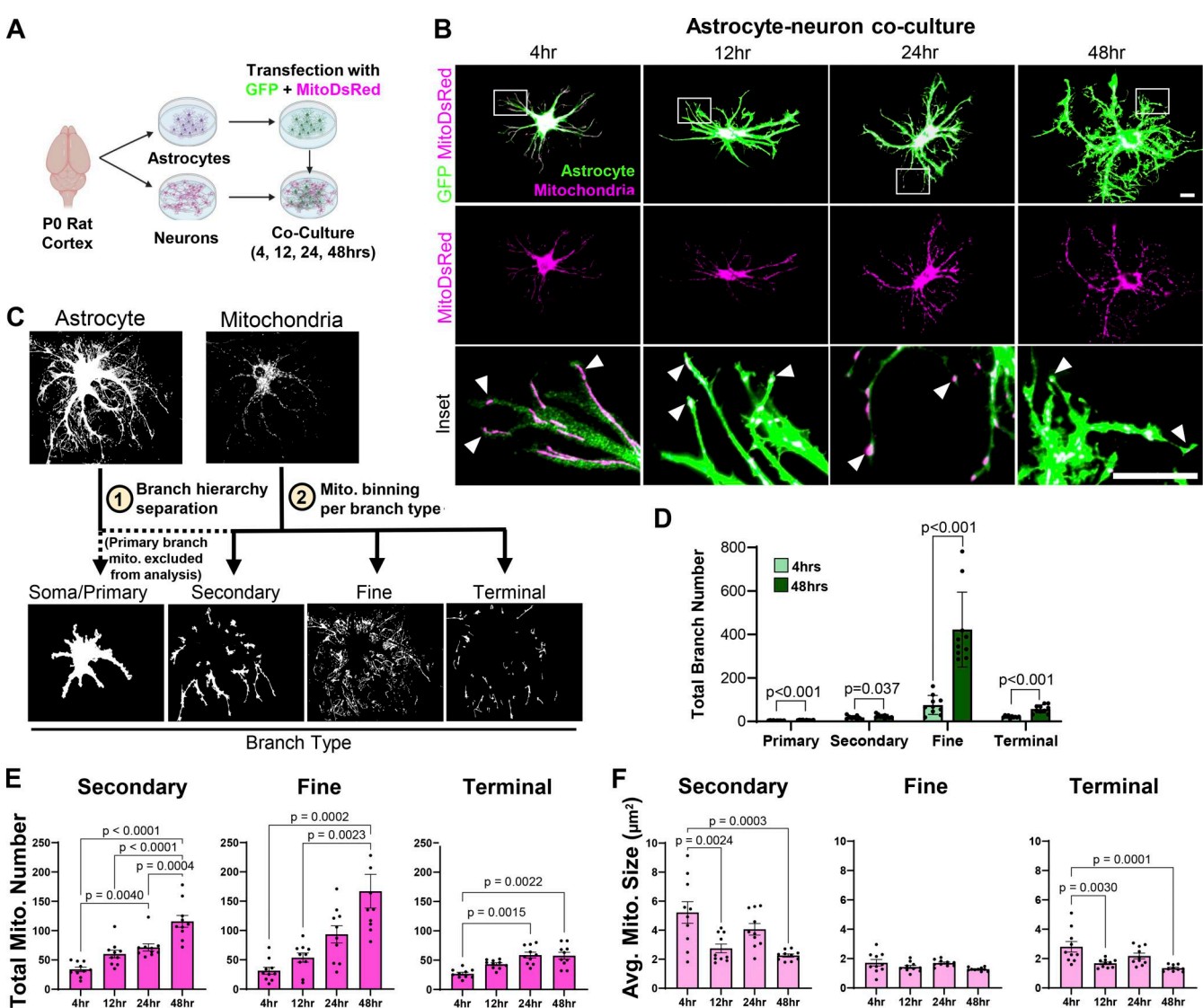

Figure 2. **Mitochondria occupy fine astrocyte processes concurrently with astrocyte morphological arborization *in vitro*. (A)** Schematic of astrocyte–neuron co-culture assay. **(B)** Representative images of rat astrocytes transfected with EGFP (green, top panels) and MitoDsRed (magenta, middle panels) from 4 to 48 h in co-culture with cortical neurons. Inset (bottom panels) of distal astrocyte processes (green) housing mitochondria (magenta) at the leading edge of growing processes (white arrowheads) from 4 to 48 h in culture. Scale bars: 20 μm. **(C)** Overview of the Seg_Astro image analysis pipeline. **(D)** Astrocyte secondary, fine/distal, and terminal branch number at 4 vs. 48 h in culture. n = 10–12 cells per time point from one experiment. Unpaired two-tailed *t* test. **(E and F)** Total astrocyte mitochondria number and (F) average mitochondrial size in secondary, fine/distal, and terminal astrocyte branches from 4 to 48 h in culture. n = 10–12 cells per time point from one experiment. Data are presented as mean ± SEM. One-way ANOVA with Tukey post hoc test.

astrocyte development. We knocked down Drp1, the key GTPase regulator of mitochondrial fission (Smirnova et al., 2001) (Fig. 3, A–C), in astrocytes using a shRNA targeting both the rat and mouse mRNAs (Fig. S2, A and B) and assessed how it affected astrocyte morphological complexity in co-culture with neurons. We co-transfected GFP-expressing shDrp1 or a scrambled control shRNA (shControl) and MitoDsRed into astrocytes and co-cultured them with neurons for 48 h (Fig. 3 C). shDrp1 astrocytes displayed robustly hyperfused and elongated mitochondria, as we would expect when inhibiting mitochondrial fission (Fig. 3 C). We used Seg_Astro to quantify mitochondrial and branch number per branch type in shControl and shDrp1 astrocytes (Fig. 3 D). We found that Drp1 knockdown significantly decreased mitochondria numbers in all branch types measured—secondary, fine, and terminal astrocyte processes—compared with control. The largest decrease in mitochondria numbers was in fine/distal branches, to <40% of shControl (Fig. 3 E). Interestingly, shDrp1 astrocytes only had a significant loss of fine and terminal branch numbers compared with shControl, with the greatest decrease quantified in fine branch numbers (∼40% of control, Fig. 2 F). Therefore, both the fine branch number and fine the branch mitochondrial number in shDrp1 astrocytes decreased to the same degree compared with the control (Fig. 2, E and F). Drp1 loss did not affect primary or secondary branch numbers, despite secondary branches also displaying a significant decrease in mitochondrial numbers in shDrp1 conditions (Fig. 3 F). These data demonstrate that Drp1 is specifically required for fine and distal astrocyte process formation, likely

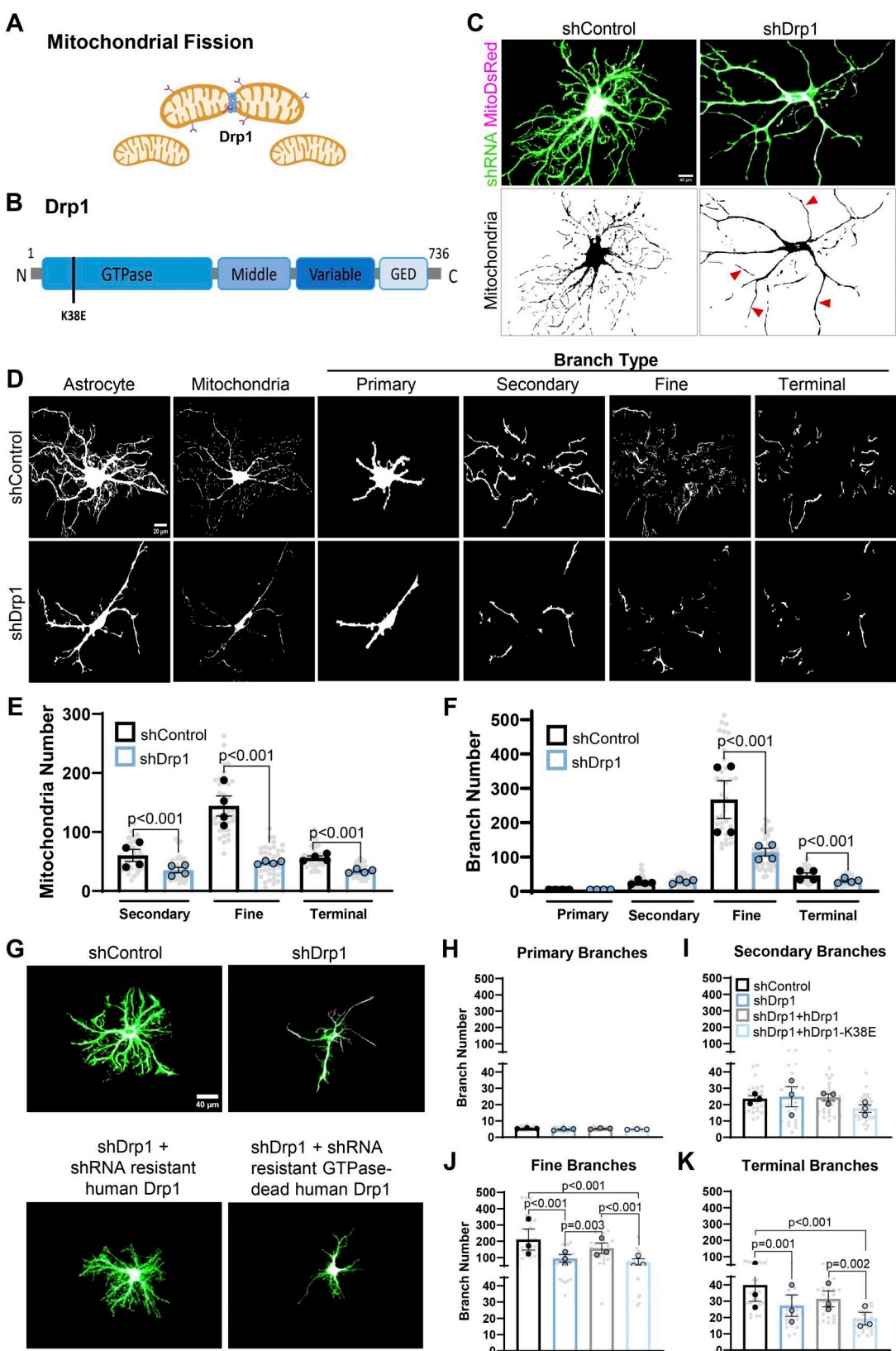

Figure 3. **Drp1-induced mitochondrial fission is required for fine astrocyte process formation and mitochondrial localization *in vitro*. (A)** Schematic of Drp1 in mitochondrial fission. **(B)** Drp1 domain structure, noting the K38E mutation in the GTPase domain. **(C)** Representative images of rat astrocytes transfected with shControl or shDrp1 (green) and MitoDsRed (magenta) (top panels) co-cultured on neurons (unlabeled), with a MitoDsRed mask (lower panels), noting hyperfused/elongated mitochondria in shDrp1 astrocytes (red arrowheads). Scale bars: 40 µm. **(D)** Representative images of rat astrocytes transfected with shRNA targeting Drp1 (shDrp1) or a scrambled control (shControl) (first column) co-cultured on neurons (unlabeled) and their mitochondria

(second column) across primary, secondary, fine, and terminal astrocyte process types. Scale bar: 20 µm. **(E)** Super-plotted quantification of average mitochondrial number in secondary, fine, and terminal branches from shControl vs. shDrp1 astrocytes. **(F)** Quantification of primary, secondary, fine, and terminal branch numbers from shControl and shDrp1 astrocytes. Data are mean ± SEM *n* = 4 independent experiments (large circles), 10 cells/condition/experiment (small gray dots). Nested *t* test. **(G)** Representative images of rat astrocytes transfected with shControl (green), shDrp1 (green), shDrp1 (green)+ hDrp1-YFP (magenta), or shDrp1 (green)+ hDrp1-K38E-CFP (magenta). Scale bar: 40 µm. **(H–K)** Quantification of total primary, (I) secondary, (J) fine, and (K) terminal branch number in astrocytes across the 4 conditions from G. Data are mean ± SEM *n* = 3 independent experiments (large circles), 10 cells/condition/experiment (small gray dots). Nested one-way ANOVA with Tukey post hoc test.

through regulation of mitochondrial size and recruitment to these fine processes.

To determine if Drp1 GTPase activity-driven scission is necessary for its role in astrocyte distal process morphogenesis, we implemented human Drp1 constructs that are resistant to shDrp1 (Fig. S2 A): WT human Drp1 (hDrp1) and GTPase-dead point mutant human Drp1 (hDrp1-K38E) (Fig. 3 B) (König et al., 2021). The co-transfection of shDrp1 with hDrp1 rescued astrocyte distal process formation compared with shDrp1 alone, whereas the GTPase-dead counterpart could not (Fig. 3, G–K). Together, these data reveal that Drp1-mediated mitochondrial fission is necessary for the formation of fine/distal astrocyte branches *in vitro*. These data also suggest that mitochondrial recruitment to distal processes is facilitated by mitochondrial fission, and this recruitment is required for astrocyte fine branch formation and/or stabilization during astrocyte morphological growth.

### Drp1 controls mitochondrial occupancy and distal astrocyte process morphogenesis *in vivo*

Because Drp1 is required for distal astrocyte branch formation *in vitro*, we next tested whether Drp1-mediated mitochondrial fission (Fig. 4 A) would play a similar role in cortical astrocyte morphogenesis *in vivo*. To address this question, we used PALE to electroporate plasmids with shDrp1 or shControl into the ventricle of P0 mito-EGFP–floxed mice. These plasmids also express a membrane-tagged mCherry reporter, which we used to quantify astrocyte morphology. They were delivered together with a Cre-expressing plasmid, which drives mito-EGFP expression to determine mitochondrial morphology at P21. We used Imaris to render 3D surfaces of astrocytes and their mitochondria and measure overall astrocyte territory size; the volume of distal neuropil-infiltrating astrocyte processes (neuropil infiltration volume; NIV); and total mitochondrial volume, size, and number in the whole astrocyte or the distal processes. Using these methods, we investigated how loss of Drp1 in cortical astrocytes *in vivo* impacted astrocyte territory size, distal process formation, and mitochondrial localization within the astrocyte arbors.

Importantly, Drp1 knockdown robustly modified astrocyte mitochondria, significantly increasing average mitochondrial size and decreasing mitochondrial number compared with shControl astrocytes (Fig. S2, A and B). These data validated that Drp1 loss inhibited mitochondrial fission in astrocytes, resulting in longer and fewer mitochondria *in vivo*, as we had observed *in vitro* (Fig. 3 C). Interestingly, shDrp1 had no effect on astrocyte territory size (Fig. 4 B), which is determined by larger astrocyte process outgrowth. However, Drp1 knockdown significantly decreased mitochondrial localization to distal astrocyte regions, which was accompanied by a significant decrease in the volume

of distal neuropil-infiltrating astrocyte processes (Fig. 4, C and D). These results suggest that mitochondrial fission controls mitochondrial recruitment to distal/perisynaptic astrocyte processes (PAPs), which is required for proper neuropil infiltration by astrocytes *in vivo*.

To determine if distal astrocyte process formation was specifically controlled by Drp1-induced mitochondrial fission or a byproduct of modifying mitochondrial dynamics in general, we tested how disrupting mitochondrial fusion by knocking down Mitofusin1 (Mfn1) or inhibiting mitochondrial transport by knocking down Miro1 modified astrocyte morphogenesis and mitochondrial occupancy. Mfn1 is a GTPase on the outer membrane of mitochondria that facilitates fusion (Chen et al., 2003) (Fig. 4 E). Miro1 is a GTPase adaptor that tethers the outer mitochondrial membrane to cytoskeletal motors for mitochondrial transport (Fransson et al., 2006) (Fig. 4 I). We used PALE to electroporate shMfn1 or shMiro1 constructs (Fig. S1, C–E). First, we found that Mfn1 knockdown in astrocytes significantly increased mitochondrial number and decreased average mitochondrial size compared with shControl, indicating inhibited mitochondrial fusion in shMfn1 astrocytes (Fig. S2, A and C). shMiro1 had no effect on astrocyte mitochondrial numbers but had a slight yet significant increase in mitochondrial size (Fig. S2, A and D). Interestingly, silencing Mfn1 caused a reduction in astrocyte territory size (Fig. 4 F) but did not alter the volume of distal neuropil-infiltrating processes (Fig. 4, G and H), indicating mitochondrial fusion may play a role in astrocyte territory outgrowth independently of distal process complexity. Despite increased mitochondrial numbers, significantly decreased mitochondrial size in Mfn1 KD astrocytes may stunt astrocyte territory outgrowth. Miro1 knockdown did not impact astrocyte morphology (Fig. 4, J and K) but caused an increase in the volume of mitochondria localized at the distal astrocyte processes (Fig. 4 L), suggesting Miro1-mediated mitochondrial trafficking may regulate retrograde mitochondrial transport in developing astrocytes. Lastly, total mitochondrial content (total mitochondrial volume normalized to total cell volume) was not altered in any of the genetic manipulations of mitochondrial dynamics (Fig. S2, E–H). This result indicates that dysfunctional mitochondrial fission, fusion, or transport impacts astrocyte morphogenesis proportionally to changes in the mitochondrial network. Taken together, we found that mitochondrial dynamics are key regulators of astrocyte morphology, likely through the control of mitochondrial localization to nascent processes. Our findings show that astrocyte territory growth (large primary/secondary processes) does not require Drp1 function. However, Drp1-induced mitochondrial fission, but not mitochondrial fusion or transport, is necessary for the formation of distal neuropil-infiltrating astrocyte processes *in vivo*.

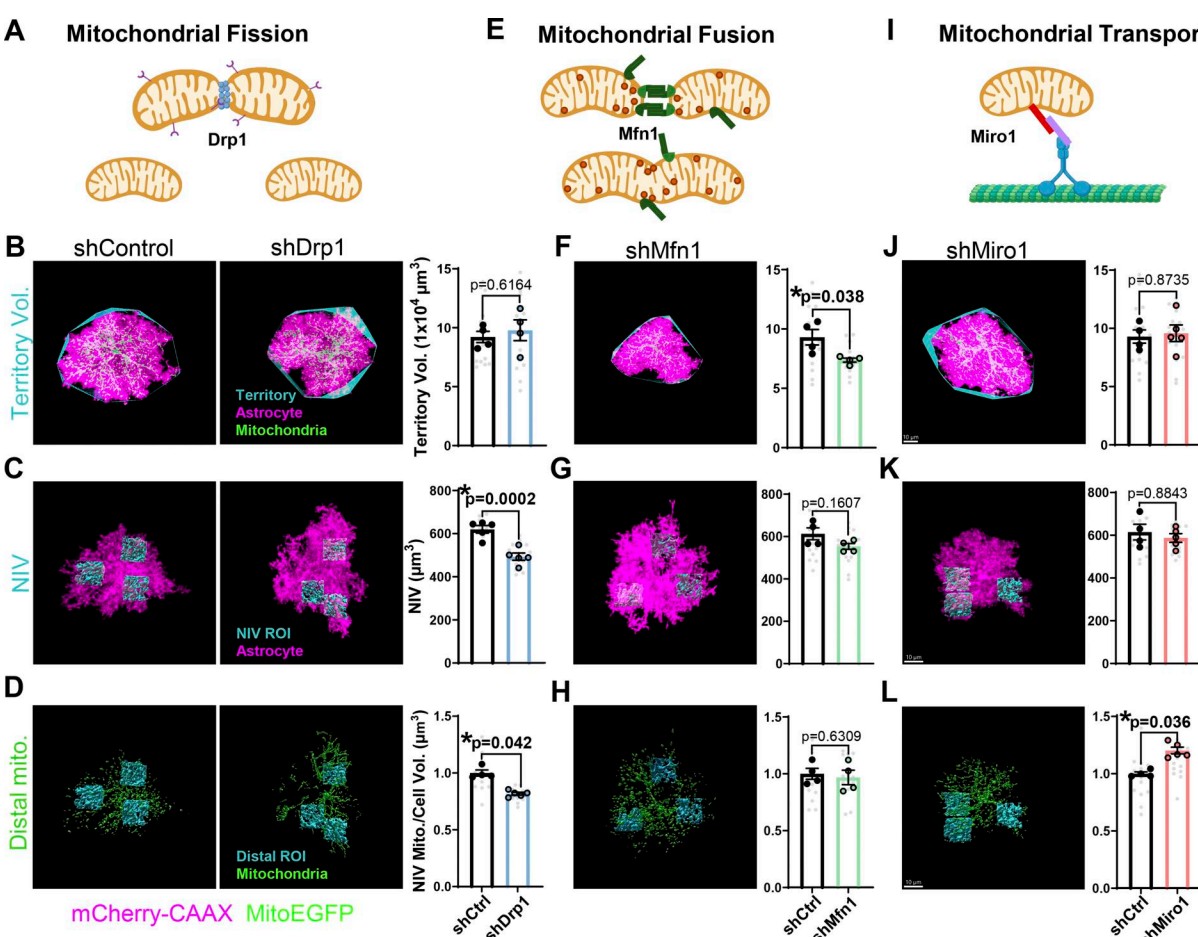

Figure 4. **Mitochondrial fission, not fusion nor transport, is required for distal astrocyte process morphogenesis in vivo. (A, E, and I)** Schematic of mitochondrial fission, fusion, and transport mediated by Drp1, Mfn1, and Miro1, respectively. **(B–D, F–H, and J–L)** Representative images of V1 P21 astrocytes expressing mCherry-CAAX–tagged shRNA (magenta) against Drp1 (shDrp1), Mfn1 (shMfn1), Miro1 (shMiro1), or scrambled control (shControl) and their EGFP mitochondria (green). Scale bar, 10 µm. **(B, F, and J)** Astrocyte territory (cyan) and quantification, **(D, G, and K)** NIV reconstructions (cyan) and quantification, and (D, H, and L) distal mitochondrial volume (green within cyan ROI) and quantification in shControl, shDrp1, shMfn1, and shMiro1 conditions, respectively. N = 4–5 male and female mice/condition (large circles), n = 2–4 cells/mouse, 10–15 cells total/condition (small gray dots). Data are mean ± SEM. Nested t test.

## Drp1 is required for astrocyte organization during postnatal cortical development

During our PALE studies investigating the role of Drp1 in cortical astrocyte morphogenesis, we observed an unusual clustering phenotype in shDrp1-transfected astrocytes compared with shControl at P21 (Fig. 5 A). PALE time course experiments from P7 to P21 revealed that shDrp1 astrocytes clustered as early as P7, and the clustering effect was cumulative across time points, reaching a nearly fourfold increase in cluster area compared with control by P21 (Fig. 5 B). To determine if this phenotype was due to a prolonged or exuberant astrocyte proliferation, we used Click-IT EdU chemistry to label proliferating cells in the cortex of PALE mice. shControl and shDrp1 plasmids were electroporated at P0, then EdU was injected intraperitoneally every 2 days from P3 to P13, and brains were collected 15 h after EdU injection from P4 through P14 (Fig. S3 A). We did not find any evidence of increased or prolonged cell proliferation in shDrp1-transfected astrocytes compared with control (Fig. S3, B and C), indicating astrocyte clustering in shDrp1-transfected astrocytes is not caused by overproliferation. These shDrp1 astrocyte clusters remained into

adulthood (in 2- and 6-mo-old mice, Fig. S3, D and E), demonstrating astrocyte clustering due to Drp1 loss is not a transient developmental phenomenon that resolves after maturation.

Cortical astrocytes organize themselves into a network of evenly dispersed, nonoverlapping domains, which is critical for proper brain function (Bushong et al., 2004). Because early postnatal loss of Drp1 led to cortical astrocyte clustering, we next sought to understand whether these astrocyte clusters displayed disrupted astrocyte organization. To do so, we measured the nearest neighbor distance between Sox9+ astrocyte nuclei to investigate whether shDrp1 astrocyte clusters consisted of adjacent astrocytes that maintained their evenly dispersed astrocyte organization (equidistant nuclei) or disorganized astrocytes with inconsistent nuclei distances (Fig. 5 C). First, we found that Drp1 knockdown clusters contained ~4 nuclei on average compared with control astrocytes, which only contain 1 nucleus (Fig. 5 D), confirming that shDrp1-transfected astrocyte clusters comprised multiple astrocytes. Importantly, shDrp1 astrocyte clusters had a threefold decrease in nearest neighbor distance compared with control astrocytes (Fig. 5 E), indicating a

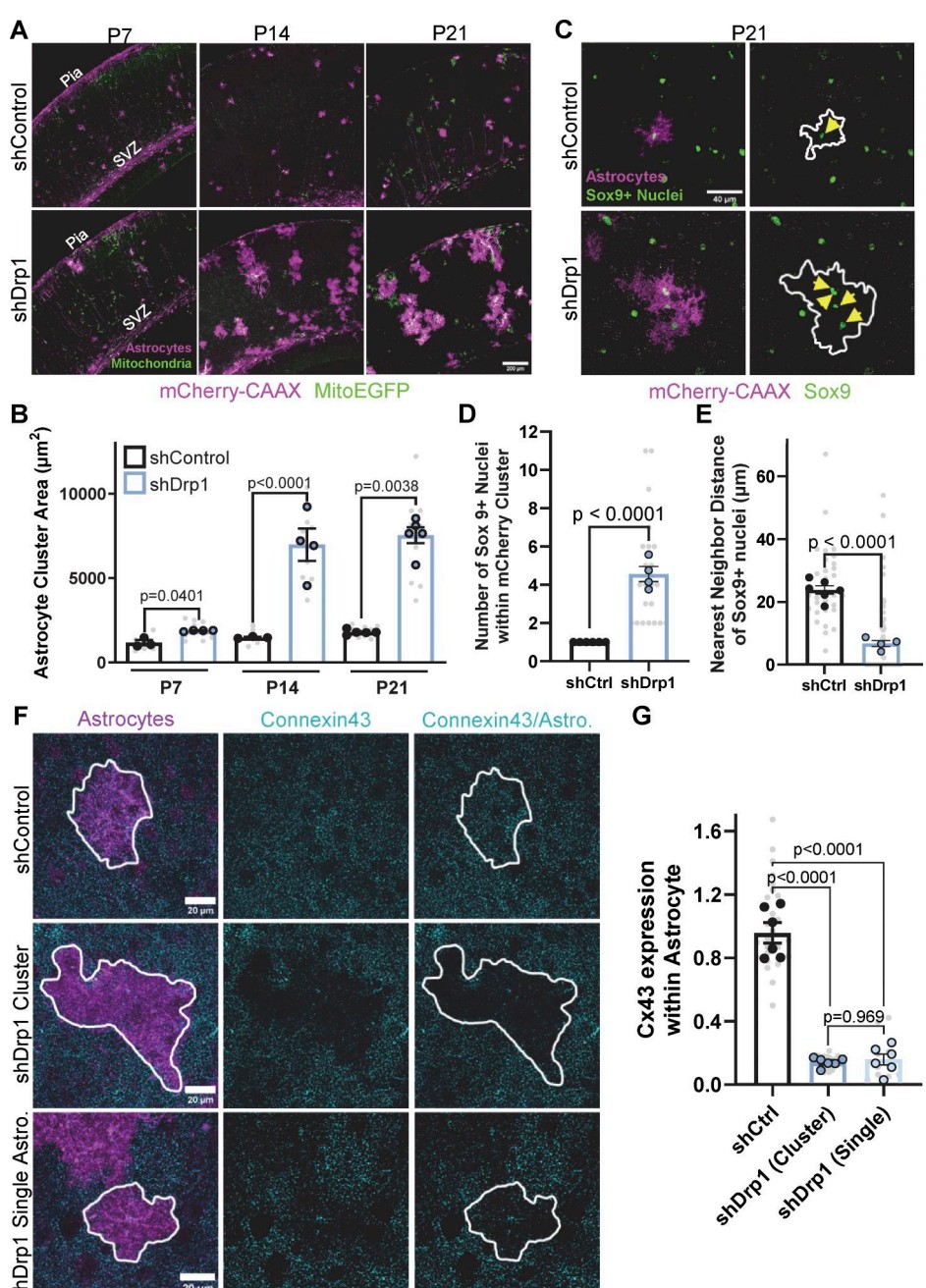

Figure 5. **Drp1 controls cortical astrocyte organization during postnatal development. (A)** Representative images of V1 mouse cortices with astrocytes expressing mCherry-CAAX–tagged shRNA (magenta) against Drp1 (shDrp1) or scrambled control (shControl) and their EGFP mitochondria (green) at P7, P14, and P21. Scale bar, 200 μm. **(B)** Quantification of astrocyte individual or cluster area per field of view at P7, P14, and P21. N = 3–4 male and female mice/ condition (big circles), three cortical sections/mouse (small gray dots). Data points are mean ± SEM. Nested *t* test. **(C)** Representative images of shControl (top) and shDrp1 (bottom) P21 astrocytes expressing mCherry-CAAX shRNAs (magenta) and stained with Sox9 (green). Yellow arrows note Sox9+ astrocytic nuclei within mCherry+ astrocytes. Scale bar, 40 μm. **(D and E)** Quantification of the number of Sox9+ astrocyte nuclei and (E) nearest neighbor distance between astrocytes per astrocyte cluster in P21 shControl and shDrp1 astrocytes. N = 4–5 male and female mice/condition (large circles), n = 2–6 cells or clusters/ condition, 20 cells or clusters total/condition (small gray dots). Data are mean ± SEM. Nested *t* test. **(F)** Representative images of V1 astrocytes expressing mCherry-CAAX (magenta) and shControl (top row) or shDrp1 (bottom two rows, shDrp1 clusters or single astrocytes) stained with Cx43 (cyan) at P21. Scale bars, 20 μm. **(G)** Quantification of Cx43 expression within mCherry-CAAX astrocytes. N = 6 male and female animals/condition (large circles), n = 1–8 cells or clusters/animal, 10–25 cells total/condition (small gray dots). Data are mean ± SEM. Nested one-way ANOVA.

disorganization among cortical astrocytes when Drp1 is knocked down. These data show that loss of Drp1 impairs the astrocyte organization in the cortex by inducing clustering, suggesting Drp1 may play a role in the establishment of astrocyte tiling during development. Taken together, these data demonstrate that Drp1-induced mitochondrial fission promotes both astrocyte morphogenesis and organization during postnatal development in the mouse cortex.

Astrocyte organization in the cortex is mechanistically linked to the formation of a gap junction–coupled astrocyte network (Baldwin et al., 2021). In particular, proper expression and localization of Cx43, the most abundant astrocytic gap junction protein (Dermietzel et al., 1991), is required for the establishment of nonoverlapping astrocyte territories (Baldwin et al., 2021; Lagos-Cabré et al., 2019). Furthermore, Cx43 has non-gap junction functions such as cell migration and cell–cell interactions, which control astrocyte migration and localization in the cortex (Homkajorn et al., 2010). Because we observed astrocyte disorganization and clustering in PALE shDrp1 astrocytes, we next wondered if loss of Drp1 caused concurrent disruptions of gap junctions. To test this possibility, we quantified Cx43 via immunohistochemistry within shDrp1 and shControl PALE astrocytes. Strikingly, shDrp1 astrocyte clusters had a fivefold decrease in Cx43, with visible "holes" of Cx43 staining coinciding with the shDrp1 astrocyte clusters (Fig. 5, F and G). Interestingly, Cx43 levels in shDrp1 astrocytes that were not part of shDrp1 astrocyte clusters or on the periphery of these clusters also exhibited significantly decreased Cx43 protein levels (Fig. 5, F and G), suggesting that inhibited mitochondrial fission leads to decreases of Cx43 protein in both single and clustered shDrp1 astrocytes. This finding also indicates that astrocyte clustering requires multiple mitochondrial fission–deficient astrocytes to be adjacent to occur, as single shDrp1 astrocytes with decreased Cx43 expression did not cluster with the neighboring WT astrocytes. These data reveal that Drp1 knockdown results in a profound reduction of Cx43 protein expression in astrocytes. Importantly, this effect of shDrp1 is Cx43 specific, as we did not observe a change in the levels of Cx30, another gap junction protein that is also expressed by astrocytes (Fig. S3 F). These results also suggest that Drp1-mediated Cx43 expression and/or localization could be linked to the observed clustering phenotype of Drp1 knockdown astrocytes.

## Astrocyte-specific Drp1 knockout induces astrocyte reactivity

To understand the impact of Drp1 loss in all astrocytes during cortical development, we utilized a conditional (floxed) allele of mouse Drp1 (Wakabayashi et al., 2009a), which we crossed with the tamoxifen-inducible astrocytic Cre-driver mouse Aldh1l1-CreERT2 (Srinivasan et al., 2016). A tdTomato-floxed reporter (Ai14) was also used to identify cells expressing Cre (Fig. 6 A). To delete Drp1 from developing astrocytes, tamoxifen was injected into the milk spot of pups at P1, P2, and P3, and mice were collected for analyses at P21 (Fig. 6 A). This approach resulted in a significant reduction of Drp1 protein in Drp1 conditional knockout (cKO) astrocytes compared with WT control (cWT) by immunoblotting of isolated cortical astrocytes (Fig. 6, B and C). Further, we found Drp1 protein reduced within Cre-positive (tdTomato⁺) astrocytes across all cortical layers by immunohistochemistry, with an overall significant decrease in V1 cortex in Drp1 cKO astrocytes compared with control (Fig. S4, A–C). To assess the recombination efficiency in this model, we stained for the astrocyte-specific nuclear marker Sox9 and quantified the number of tdTomato⁺/Sox9⁺ double-positive cells in proportion to the total number of Sox9⁺ cells in Drp1 cKO and cWT mice (Fig.

S4 E). We found that Cre recombination was not significantly different between Drp1 cWT and cKO conditions; both were ~80% efficient in recombining cells (Fig. S4 F). These results show that Aldh1L1-CreERT2 with daily tamoxifen injections between P1 and P3 can be used to significantly reduce astrocytic Drp1 expression in mice *in vivo*.

Because we found that shDrp1 decreased astrocyte distal process formation and caused astrocyte clustering in our sparse PALE manipulation model, we wondered how global loss of Drp1 in astrocytes via conditional deletion would affect astrocyte development and cortical homeostasis. We found an overall decrease in astrocyte density in the cortex by tdTomato⁺ soma count that was primarily driven by decreases in L1 and L6 astrocytes in Drp1 cKO mice compared with control (Fig. 6, D and E). Whole cortical quantification of the astrocyte-specific nuclear marker Sox9 also confirmed a slight but significant decrease in overall astrocyte density (Fig. S4, E and G). The decrease in astrocyte density did not impact overall cortical thickness (Fig. S4 D). Next, we quantified tdTomato⁺ signal by immunohistochemistry as a proxy for astrocyte coverage and found that loss of Drp1 in cKO astrocytes significantly decreased their process coverage across the cortex compared with cWT, with the greatest coverage loss found in L5 and L6 astrocytes (Fig. 6 F). Given that our astrocyte coverage analysis was normalized by astrocyte density, these data suggest that this significant reduction in cortical astrocyte coverage is driven by the loss of astrocyte process formation in Drp1 cKO.

Astrocytes modify their morphology and coverage of the cortex under disease and injury conditions (Fiebig et al., 2019; Oberheim et al., 2008). Further, mature cortical astrocytes express low GFAP protein under homeostatic conditions but increase their GFAP expression upon entering a reactive state (Middeldorp and Hol, 2011). Importantly, astrocyte reactivity is accompanied by shifts in both metabolic state and mitochondrial dynamics (Cassina et al., 2025; Rahman and Suk, 2020). Because we found morphology and coverage abnormalities in astrocyte-specific Drp1 cKO mice, we next tested whether the astrocytes in these brains became reactive. To do so, we performed immunohistochemistry for GFAP in Drp1 cKO and cWT cortices and quantified changes in their GFAP density and coverage (Fig. 6 D). We found that Drp1 cKO mice had significantly increased cortical GFAP expression predominantly in the mid layers (L2/3, L4, and L5) compared with control (Fig. 6, G and H). Next, we measured mRNA levels of six markers of astrocyte reactivity in isolated astrocytes from Drp1 cKO and cWT mice. We found a significant upregulation of four out of six markers—GFAP, vimentin, LCN2, and Cxcl10 (Fig. 6, I–L), a trending increase of Serpina3n, and no significant difference in Cp mRNA levels (Fig. S4, H and I). These findings indicate that mitochondrial fission maintains astrocyte homeostasis during development and suggest that developmental conditional deletion of Drp1 in astrocytes induces cortical astrocyte reactivity *in vivo*.

## Astrocytic Drp1 loss disrupts astrocyte organization and dysregulates PAP protein expression *in vivo*

To determine if conditional deletion of Drp1 in cortical astrocytes disrupted astrocyte organization as we previously observed in

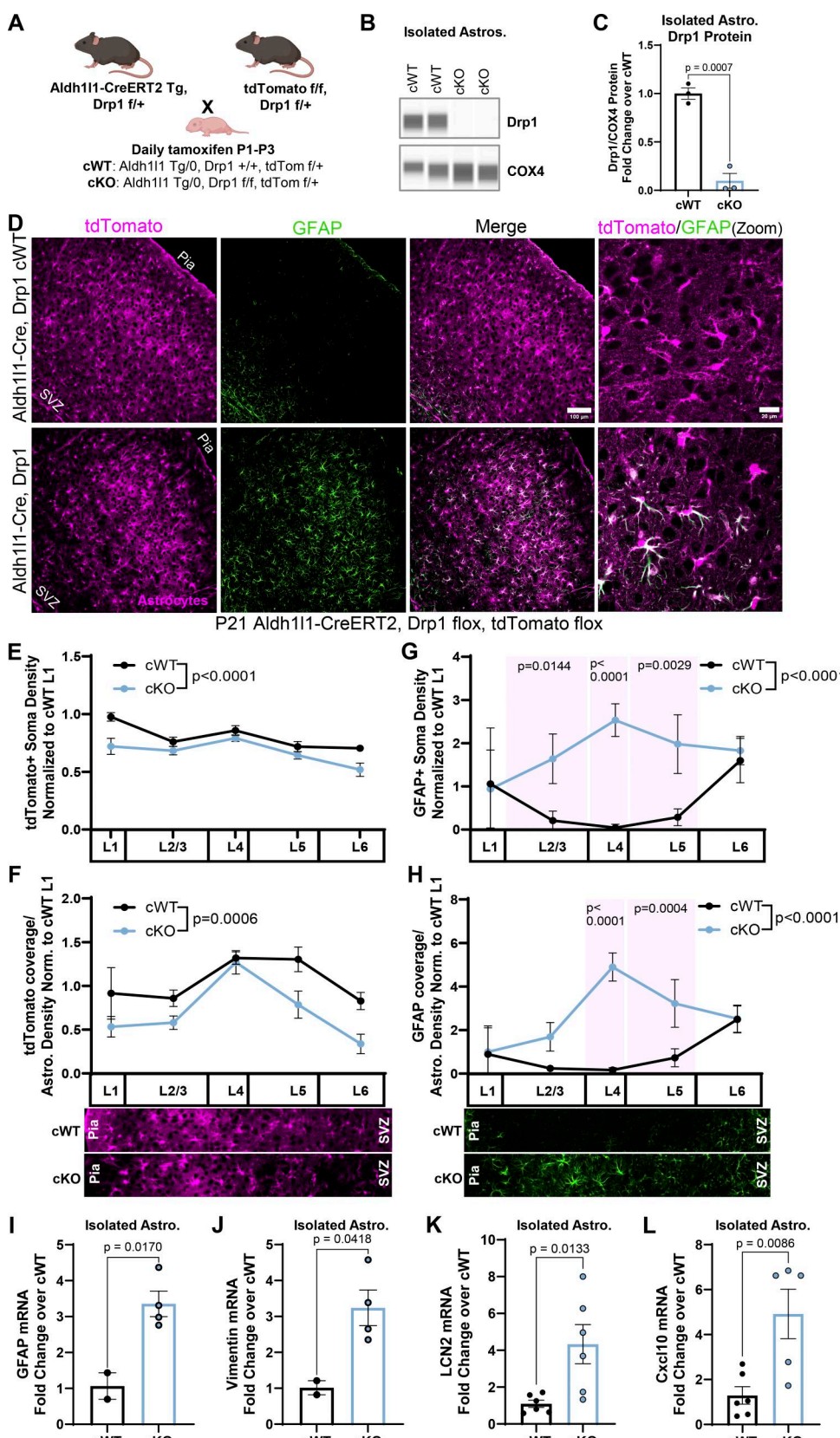

Figure 6. **Astrocyte-specific Drp1 cKO induces cortical astrocyte reactivity. (A)** Overview of astrocyte-specific conditional Drp1 knockout mouse breeding strategy and timeline of tamoxifen administration. **(B)** Representative immunoblot for Drp1 from immunopurified astrocytes. COX4, a mitochondrial protein, serves as a loading control. Immunoblot ran in the same experiment as Fig. 7 I, therefore COX4 loading control image is the same as Fig. 7 I. **(C)** Quantification of Drp1 protein in isolated cKO and cWT astrocytes. N = 3 mice/condition. Data are mean ± SEM. Unpaired, two-tailed t test. **(D)** Representative images of Drp1

cWT and cKO V1 cortices at P21 with tdTomato$^+$ astrocytes (magenta, first column), stained for GFAP (green, second column), and merged (third column). Scale bar, 100 µm. Zoom merge (last column). Scale bar, 20 µm. **(E–H)** Quantification of cortical tdTomato$^+$ soma count of astrocyte density, (F) tdTomato$^+$ astrocyte coverage normalized to astrocyte density, (G) GFAP soma count, and (H) GFAP coverage normalized to astrocyte density per layer in Drp1 cKO compared with cWT. $N$ = 4 male and female mice/condition, 2–3 cortical images/mouse. Data are mean ± SEM. Two-way ANOVA with Sidak's multiple comparisons test. **(I–L)** Quantification of (I) GFAP, (J) vimentin, (K) LCN2, and (L) Cxcl10 mRNA from isolated Drp1 cKO and cWT astrocytes. $N$ = 2–6 male and female mice/condition. Data are mean ± SEM. Unpaired two-tailed $t$ test. Source data are available for this figure: SourceData F6.

our sparse PALE knockdown model (Fig. 5), we quantified both nearest neighbor and multiple neighbor distances between tdTomato$^+$ somas in the cortices of cKO and cWT mice (Fig. 7 A). We found that in Drp1 cKO mice, astrocytes displayed significant decreases in nearest neighbor distance (Fig. 7 B), similar to the astrocyte clustering observed in our sparse Drp1 KD model. Further, quantification of distances between multiple astrocyte neighbors revealed that Drp1 cKO astrocytes have a greater variability in distances to their neighbors compared with cWT astrocytes (Fig. 7 C), indicating Drp1 cKO astrocytes have an unevenly dispersed distribution of astrocyte somas compared with cWT. These data identify a disorganization in cortical astrocytes in the absence of Drp1. Taken together, these results are in line with our findings using shRNA constructs and PALE (Fig. 5) and show that loss of Drp1 in developing astrocytes disrupts cortical astrocyte organization *in vivo*.

Given the robust disruption to astrocyte organization in Drp1 cKO mice, we next tested how Cx43 expression was impacted in Drp1 cKO astrocytes compared with cWT. To do so, we stained for Cx43 in the cortex and found that Drp1 cKO astrocytes displayed a heterogeneous and uneven expression of Cx43, with some astrocytes apparently expressing almost no Cx43 while others expressed WT or higher levels of the protein compared with the more even distribution of Cx43 in cWT cortices (Fig. 7 D). Indeed, quantification of the Cx43 signal showed that while there was no change in average Cx43 expression between genotypes (Fig. 7 E), there was a significant increase in the variance of Cx43 expression in Drp1 cKO astrocytes compared with cWT (Fig. 7 F). We quantified this heterogeneity by comparing the histogram of Cx43 expression per astrocyte and found that Drp1 cKO astrocytes had a significantly altered distribution of Cx43 expression (Fig. 7 G). This heterogeneous expression of cortical Cx43 protein *in situ* led us to investigate how Cx43 mRNA and protein expression may be disrupted in Drp1 cKO astrocytes. To do so, we immunopurified astrocytes from the cortex and measured mRNA and protein levels of Cx43. Cx43 mRNA levels were unchanged in Drp1 cKO isolated astrocytes compared with cWT (Fig. 7 H). Interestingly, immunoblotting for Cx43 in isolated astrocytes demonstrated a robust decrease of Cx43 full-length (FL) protein as well as its internally translated smaller isoforms in Drp1 cKO astrocytes compared with cWT (Fig. 7 I). Importantly, the trafficking of Cx43-FL to the cell membrane to carry out its gap junction and cell adhesion functions is regulated by its C-terminal isoform, Cx43-20k (Smyth and Shaw, 2013). Both Cx43-FL and Cx43-20k protein levels were significantly decreased in Drp1 cKO astrocytes compared with cWT (Fig. 7, J and K). These findings indicate that loss of Drp1 dysregulates Cx43 protein, but not mRNA, expression in astrocytes *in vivo*.

Lastly, we tested how Drp1 loss impacts the abundance of other astrocyte membrane proteins known to be localized to PAPs. To do so, we immunopurified astrocytes from the cortices of cWT and cKO mice and measured protein levels of glutamate-aspartate transporter (GLAST), glutamate transporter-1 (Glut1), and the inwardly rectifying potassium channel Kir4.1 (Pajarillo et al., 2019; Takumi et al., 1995) by immunoblotting (Fig. 7, L, N, and P). Intriguingly, all three PAP proteins were significantly decreased in isolated Drp1 cKO astrocytes compared with control (Fig. 7, M, O, and Q), potentially due to the loss of distal astrocyte processes that house these PAP proteins (Fig. 4 C and Fig. 6 F). These data indicate that mitochondrial fission regulates PAP protein expression, including but not limited to Cx43, in developing astrocytes. However, Drp1 cKO astrocytes lose Cx43 protein levels to a much greater degree (reduced to <10% of cWT) than other PAP proteins (∼70–50% of cWT), suggesting Cx43 homeostasis is more vulnerable to aberrant mitochondrial fission in astrocytes. Taken together, these data suggest that Drp1-induced mitochondrial fission is required for proper PAP formation, PAP protein expression, and the establishment of astrocyte organization in the cortex.

## Discussion

Astrocytes require a highly complex morphology to perform essential roles in brain development. They ramify into hundreds of thousands of fine processes, enabling their functional interaction with synapses, neighboring astrocytes, and other brain cells. Distal mitochondria locally interact with the cytoskeleton to regulate branch morphology in compartmentalized cells such as neurons (Courchet et al., 2013; Smith and Gallo, 2018), but the roles of mitochondria in fine astrocyte processes are largely unknown. Mature astrocytes were historically believed to be predominantly glycolytic (Pellerin and Magistretti, 1994), not depending on mitochondrial respiration for their own survival (Supplie et al., 2017), and instead using their mitochondria to provide metabolic support for neurons (Pellerin et al., 1998). Indeed, astrocyte mitochondria have recently emerged as gatekeepers of several biological processes that maintain brain homeostasis, including regulation of blood–brain barrier remodeling, neuroinflammation, and neurodegeneration (Göbel et al., 2019, *Preprint*; Ignatenko et al., 2018; Motori et al., 2013; Popov et al., 2023). While these studies introduce astrocytic mitochondria as key regulators of brain function, the cell-intrinsic roles of mitochondria in astrocyte and brain development are not well understood.

Our findings elucidate a previously unknown role for mitochondrial fission in regulating distal astrocyte process morphogenesis and astrocyte organization in the developing mouse

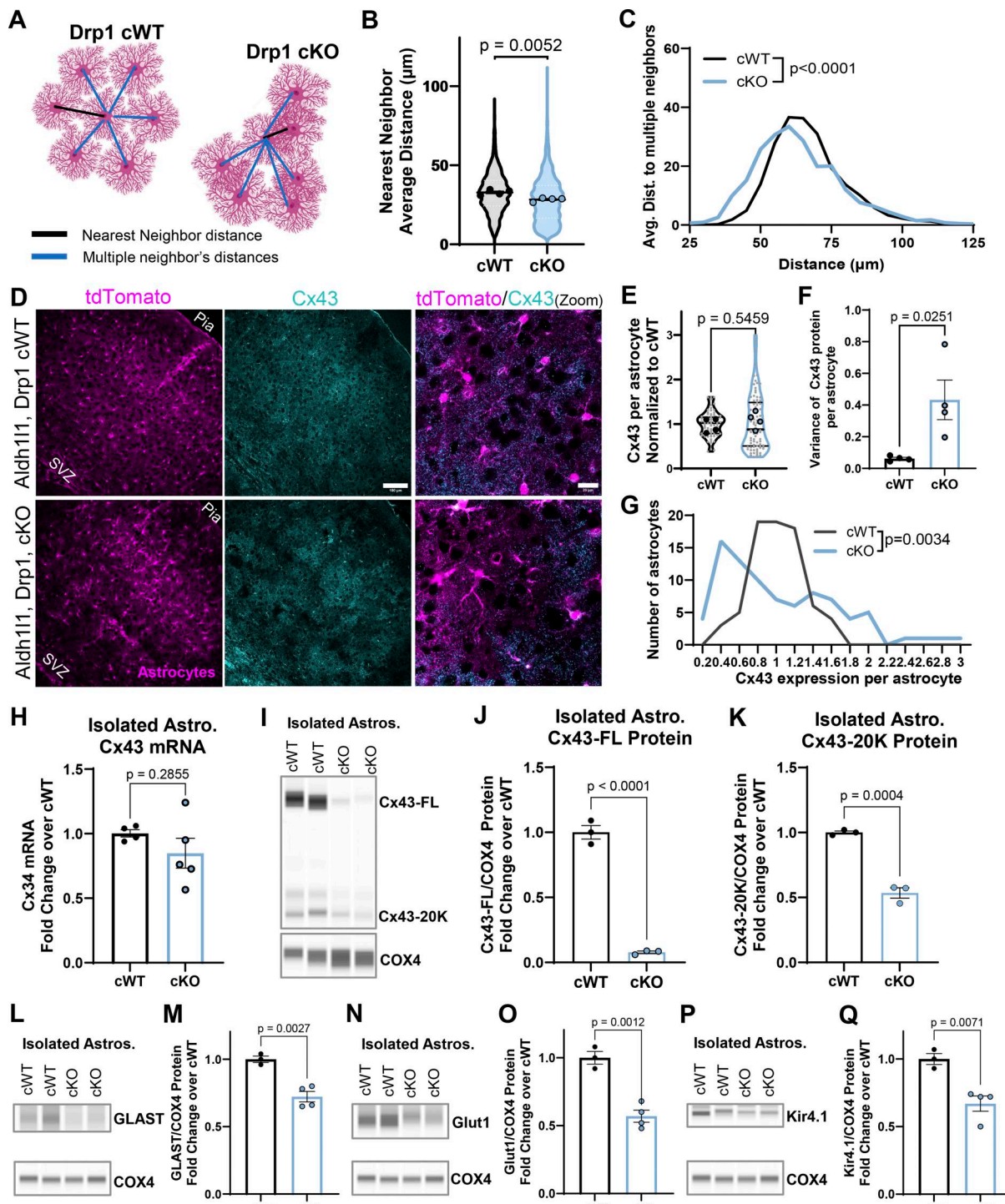

Figure 7. **Astrocytic Drp1 loss dysregulates astrocyte organization and PAP protein expression in the mouse cortex. (A)** Diagram of astrocyte organization analysis by nearest and multiple neighbor distance quantification. **(B)** Violin plot of nearest neighbor average distance between tdTomato⁺ somas across V1 cortex of Drp1 cWT and cKO mice. $N$ = 4 male and female mice/condition (large circles), ~200 distances per image, 2–3 cortical images/mouse. Data are mean ± SEM. Nested $t$ test. **(C)** Distribution of average distances to multiple (9) neighbors. $N$ = 4 male and female mice/condition, ~200 distances per image, 2–3 images/mouse. Kolmogorov–Smirnov test. **(D)** Representative images of Drp1 cWT and cKO V1 cortices with tdTomato⁺ astrocytes (magenta) stained with Cx43 (cyan) at P21. Scale bar, 100 µm. Zoom merge (last column). Scale bar, 20 µm. **(E–G)** Quantification of (E) individual, (F) variance, and (G) distribution of Cx43 mean gray value per astrocyte in Drp1 cKO and cWT mice. $N$ = 4 male and female mice/condition, $n$ = 15–30 mid-layer astrocytes (L2/3, L4, and L5) per animal from 2 to 3 section images/animal. Nested $t$ test for (E), unpaired, two-tailed $t$ test for (F), and Kolmogorov–Smirnov test for G. **(H)** Quantification of Cx43 mRNA from isolated cKO and cWT astrocytes. $N$ = 4–6 mice/condition. Data are mean ± SEM. Unpaired two-tailed $t$ test. **(I)** Representative immunoblot of isolated astrocytes for Cx43 showing its FL form at 40 kDa and its 20 kDa isoform using COX4 as loading control. Immunoblot ran in the same experiment as Fig. 6 B, therefore COX4 loading control image is the same as Fig. 6 B. **(J and K)** Quantification of FL Cx43 protein and (K) Cx43-20 kDa isoform protein in isolated cKO and cWT astrocytes. $N$ = 3 mice/condition. Data are mean ± SEM. Unpaired, two-tailed $t$ test. **(L, N, and P)** Representative immunoblots for (L) GLAST, (N)

Glut1, and (P) Kir4.1 from immunopurified astrocytes from Drp1 cWT and cKO mice. COX4, a mitochondrial protein, serves as a loading control. **(M, O, and Q)** Quantification of (M) GLAST, (O) Glut1, and (Q) Kir4.1 protein in isolated cKO and cWT astrocytes. $N$ = 3–4 mice/condition. Data are mean ± SEM. Unpaired, two-tailed $t$ test. Source data are available for this figure: SourceData F7.

cortex. We show that astrocytes undergo robust mitochondrial division to predominantly occupy distal astrocyte processes during postnatal cortical development. Inhibiting mitochondrial fission by silencing Drp1 reduces mitochondrial occupancy of distal astrocyte processes and diminishes peripheral astrocyte branch formation specifically, without affecting overall astrocyte territory size or growth. Functionally, Drp1-induced mitochondrial fission is essential for the establishment of an organized astrocyte network and proper expression of PAP proteins, including Cx43. Finally, global Drp1 loss in developing astrocytes induces cortical astrocyte reactivity, indicating that mitochondrial fission, distal mitochondrial localization, and astrocyte fine process formation are required for proper astrocyte development.

Several recent studies have elucidated astrocyte–neuron and astrocyte–astrocyte cell adhesion–based mechanisms that instruct astrocyte morphogenesis (Baldwin et al., 2021; Stogsdill et al., 2017; Takano et al., 2020; Tan et al., 2023; Zipursky et al., 2024). Curiously, loss of these cell adhesions often concurrently disrupts both astrocyte territory formation and complexity of distal astrocyte processes. Here, we identify an organelle-driven mechanism that regulates peripheral astrocyte process morphogenesis independently from the development of higher-order astrocyte processes and overall astrocyte size. The specificity of mitochondrial fission in regulating distal astrocyte process development could be attributed to their actin-rich cytoskeletal make-up, which is distinct from larger processes. Primary astrocyte processes, which determine territory size, contain mainly intermediate filaments and microtubules (Haseleu et al., 2013). In other cell types, local mitochondrial ATP has been shown to fuel actin polymerization (Cunniff et al., 2016). Thus, it is possible that mitochondrial fission facilitates the recruitment of mitochondria into actin-rich distal astrocyte processes to supply local ATP required for actin polymerization during morphogenesis. In the absence of mitochondrial fission, the mislocalization of mitochondria away from peripheral, fine astrocyte branchlets could impair actin-cytoskeleton dynamics during morphogenesis. Because these peripheral astrocyte processes are critical components of the tripartite synapse, future work is needed to elucidate the role of astrocyte mitochondrial fission in proper synapse formation and function.

Our study establishes the role of Drp1 and mitochondrial fission in astrocyte organization in the cortex, and this disruption is likely accompanied by tiling defects. While the molecular mechanisms that drive astrocyte domain formation and tiling remain largely unknown, we know that proper organization of astrocyte networks is functionally necessary for brain development and health. A recent study identified an astrocyte–astrocyte cell adhesion molecule, HepaCAM, that controls the establishment of nonoverlapping astrocyte domains through cis interactions with neighboring astrocytes and regulation of Cx43 localization (Baldwin et al., 2021). In our study, both sparse knockdown and conditional deletion of Drp1 in astrocytes led to

dysregulation of Cx43 protein expression and concurrent disruption of astrocyte organization in the cortex. These results support previous findings that both Cx43 abundance and stability are critical for the proper migration of astrocytes and even dispersion of the astrocyte network (Lagos-Cabré et al., 2019; Wiencken-Barger et al., 2007) and reveal that this process is regulated at least in part by astrocyte mitochondrial fission during cortical development. Additionally, mitochondrial fission itself has been shown to regulate cell migration in immune cells, cancer cells, and neural progenitor cells (da Silva et al., 2014; Kim et al., 2015; Paupe and Prudent, 2018). Thus, it is possible that Drp1 loss could directly, or synergistically with Cx43 reduction, disrupt astrocyte migration and organization in our model. Future work will need to investigate whether the relationship between Drp1, Cx43, and astrocyte organization is direct or indirect, particularly because we found other PAP proteins were also reduced in Drp1 cKO astrocytes. Lastly, while our study establishes that mitochondrial fission is necessary for proper Cx43 protein expression in astrocytes, functional gap junction–coupling studies will reveal whether Drp1 regulates astrocyte-network communication via gap junctions.

How could Drp1-induced mitochondrial fission modulate Cx43 protein levels in astrocytes? Previous studies in cardiac cells found that the C terminus of Cx43 is internally translated to generate a small 20-kDa Cx43 isoform (Cx43-20 kDa), which is required for the trafficking of Cx43-FL to the cell membrane (Smyth and Shaw, 2013). Loss of Cx43-20 kDa leads to degradation of mistrafficked Cx43-FL (Xiao et al., 2020). Importantly, Cx43-20 kDa can abandon its canonical Cx43-FL trafficking roles and instead employ a Drp1-independent, actin-based mechanism to induce mitochondrial fission in conditions of oxidative stress and mitochondrial elongation (Shimura et al., 2021). Therefore, it is plausible that Drp1 loss in astrocytes redirects Cx43-20 kDa away from its Cx43-FL trafficking role in an attempt to rescue dysfunctional mitochondrial fission, resulting in the degradation of mistrafficked Cx43-FL. Furthermore, Cx43-20 kDa has been previously detected in astrocytes, and overexpression of astrocytic Cx43-20 kDa promotes neuronal survival and recovery in a rat model of traumatic brain injury (Ren et al., 2020). These studies, coupled with our findings, suggest the Cx43/Drp1 axis may be critical for brain development, brain disease, and neurotherapeutics.

One of the existing dogmas of astrocyte biology is that astrocyte reactivity is often accompanied by increased mitochondrial fragmentation and decreased oxidative phosphorylation (Cassina et al., 2025; Motori et al., 2013; Rahman and Suk, 2020). In fact, inhibition of mitochondrial fission has been studied as a therapeutic target for reducing astrocyte reactivity and neurotoxicity in models of neurodegenerative disorders such as ALS and Huntington's disease (Gollihue and Norris, 2020; Gu et al., 2025; Reddy, 2014). Intriguingly, our study demonstrates an opposite relationship between mitochondrial fission and astrocyte

reactivity during postnatal cortical development—inhibition of mitochondrial fission in astrocytes upregulates multiple markers of astrocyte reactivity. This finding indicates active mitochondrial fission is temporally necessary for maintaining astrocyte homeostasis, specifically during brain development. Indeed, developing and mature astrocytes have distinct metabolic profiles (Zehnder et al., 2021); thus, it is plausible that mitochondrial fission may have differential effects on astrocyte metabolism, mitochondrial function, and astrocyte reactivity during astrocyte development, maturation, or aging. Lastly, while Drp1-null or neuron-specific conditional deletion of Drp1 during development is embryonically or neonatally lethal in mice (Ishihara et al., 2009; Wakabayashi et al., 2009b), our astrocyte-specific model of Drp1 loss did not affect mouse viability during the developmental timeframe of this study. Future investigations extending beyond the postnatal developmental window will reveal how astrocyte-specific Drp1 loss and astrocyte reactivity affect mouse behavior or physiology long-term.

Finally, this study has important implications for understanding the pathology of neurological diseases, particularly neurodevelopmental diseases with Drp1 dysfunction. Indeed, point mutations in human Drp1 cause encephalopathy due to defective mitochondrial and peroxisomal fission-1, a devastating disorder that presents with microcephaly, developmental delay, refractory epilepsy, or lethal infantile encephalopathy (Fahrner et al., 2016; Lhuissier et al., 2022; Liu et al., 2021; Vanstone et al., 2016). While much research has focused on the disruption of excitation/inhibition (E/I) balance and neuronal pathogenesis of encephalopathy due to defective mitochondrial and peroxisomal fission-1 (Casillas-Espinosa et al., 2012; Ke et al., 2023), our results reveal how Drp1 dysfunction in astrocytes alone can fundamentally disrupt cortical development through astrocyte morphogenesis and organization deficits. Decreased distal astrocyte process development could contribute to diminished tripartite synapse formation (Stogsdill et al., 2017), subsequently disrupting the E/I balance. Furthermore, while loss of astrocyte tiling has been reported in models of epilepsy (Oberheim et al., 2008), it has been unclear whether the phenotype is causative or a consequence of disease. Our findings suggest endogenous loss of astrocytic Drp1 is sufficient to cause astrocyte reactivity, which is a hallmark of many neurological disorders, underscoring the translational importance of astrocyte mitochondrial fission in neuropathology. In future studies, it will be interesting to investigate whether astrocyte-specific Drp1 loss causes synaptic E/I imbalance deficits associated with human *DNM1L* mutations. In conclusion, we show a novel role of Drp1-induced mitochondrial fission in astrocyte peripheral process morphogenesis and in regulating the establishment of an organized astrocyte network, demonstrating astrocyte mitochondria are essential instructors of proper brain development.

# Materials and methods
## Resource availability
Further information and requests for resources and reagents should be directed to and will be fulfilled by Cagla Eroglu (cagla.eroglu@duke.edu).

## Materials availability
The reagents generated in this study are available without restriction.

## Animal studies
All mice and rats were used in accordance with the Institutional Animal Care and Use Committee and with oversight by the Duke Division of Laboratory Animal Resources (Institutional Animal Care and Use Committee Protocol Numbers A117-20-05 and A103-23-04). All animals were housed under typical 12-h light/dark cycles. Aldh1l1-Cre/ERT2 BAC transgenic (RRID:IMSR_JAX: 029655) and ROSA-td-Tomato Ai14 (RTM) (RRID:IMSR_JAX: 007914) lines were obtained through Jackson Laboratory. WT CD1 mice (RRID: IMSR_CRL:022) used for PALE and CD1 (Sprague–Dawley) IGS rats (RRID: IMSR_CRL:400) used for primary culture preps were purchased from Charles River Laboratories. Rosa26-lsl-mito-EGFP (RRID:IMSR_JAX:021429) were previously described (Agarwal et al., 2017). Drp1 cKO mice were previously described and were a gift from the Sesaki Laboratory at John Hopkins University School of Medicine, Baltimore, MD, USA (Wakabayashi et al., 2009a). Mice and rats of both sexes were included in all experiments. Sex was not an influence on any of the experimental outcomes.

## Primary cultures
### Cortical neuron isolation and culture
Purified (glia-free) rat cortical neurons were prepared as described previously (Stogsdill et al., 2017). In brief, cortices from P1 rat pups of both sexes (Sprague–Dawley, SD-001; Charles River Laboratories) were microdissected, digested in papain (~7.5 U/ml) at 33°C for 45 min, triturated in low and high ovomucoid solutions, resuspended in panning buffer (DPBS [14287; Gibco] supplemented with BSA and insulin), and passed through a 20-μm mesh filter (03-20/14; Elko Filtering). Filtered cells were incubated on negative panning dishes coated with *Bandeiraea simplicifolia* lectin 1 (x2), followed by goat anti-mouse IgG+IgM (H+L) (RRID: AB_2338451) and goat anti-rat IgG+IgM (H+L) (RRID:AB_2338094) antibodies, then incubated on positive panning dishes coated with mouse anti-L1 (RRID:AB_528349) to bind cortical neurons. Adherent cells were collected by forceful pipetting with a P1000 pipette. Isolated neurons were pelleted (11 min at 200 *g*) and resuspended in serum-free neuron growth media (NGM; neurobasal, B27 supplement, 2 mM L-glutamine, 100 U/ml Pen/Strep, 1 mM sodium pyruvate, 4.2 μg/ml forskolin, 50 ng/ml BDNF, and 10 ng/ml CNTF). 100,000 neurons were plated onto 12-mm glass coverslips coated with 10 μg/ml poly-D-lysine (P6407; Sigma-Aldrich) and 2 μg/ml laminin and incubated at 37°C in 10% $CO_2$. On day *in vitro* (DIV) 2, half of the media was replaced with NGM Plus (Neurobasal Plus, B27 Plus, 100 U/ml Pen/Strep, 1 mM sodium pyruvate, 4.2 μg/ml forskolin, 50 ng/ml, BDNF, and 10 ng/ml CNTF), and AraC (10 μM, C1768; Sigma-Aldrich) was added to stop the growth of proliferating contaminating cells. On DIV 3, all of the media was replaced with NGM Plus. Neurons were fed on DIV 6 and DIV 9 by replacing half of the media with NGM Plus. Detailed protocol can be found at (https://www.protocols.io/view/glia-free-cortical-neuron-culture-36wgq3r35lk5/v1).

## Cortical astrocyte isolation and culture

Rat cortical astrocytes were prepared as described previously (Stogsdill et al., 2017). Briefly, P1 rat cortices from both sexes were microdissected, papain digested, triturated in low and high ovomucoid solutions, and resuspended in astrocyte growth media (AGM: DMEM [11960; Gibco], 10% FBS, 10 µM, hydrocortisone, 100 U/ml Pen/Strep, 2 mM L-glutamine, 5 µg/ml insulin, 1 mM Na pyruvate, and 5 µg/ml N-acetyl-L-cysteine). Between 15 and 20 million cells were plated on 75-mm² flasks (non-ventilated cap) coated with poly-D-lysine and incubated at 37°C in 10% $CO_2$. On DIV 3, removal of non-astrocyte cells was performed by forcefully shaking closed flasks by hand for 10–15 s until only an adherent monolayer of astrocytes remained. AraC was added to the media from DIV 5 to DIV 7 to eliminate contaminating fibroblasts. On DIV 7, astrocytes were trypsinized (0.05% trypsin-EDTA) and plated into 12-well or 6-well dishes. On DIV 8, cultured rat astrocytes were transfected with shRNA and/or expression plasmids using Lipofectamine LTX with Plus Reagent (15338100; Thermo Fisher Scientific) as per the manufacturer's protocol. Briefly, 1 µg (12-well) or 2 µg (6-well) total DNA was diluted in Opti-MEM containing Plus Reagent, mixed with Opti-MEM containing LTX (1:2 DNA to LTX), and incubated for 30 min. The transfection solution was added to astrocyte cultures and incubated at 37°C for 3 h. On DIV 10, astrocytes were trypsinized, resuspended in NGM plus, plated (20,000 cells per well) onto DIV 10 neurons, and co-cultured for 48 h. Detailed protocol can be found at (https://www.protocols.io/view/astrocyte-isolation-and-acm-production-261gedp57v47/v1).

## Cell lines

*3T3* 3T3 cells (RRID:CVCL_0594) used for shRNA validation experiments were cultured in DMEM supplemented with 10% FBS, 100 U/ml Pen/Strep, 2 mM L-glutamine, and 1 mM sodium pyruvate. Cells were incubated at 37°C in 5% $CO_2$ and passaged every 2–3 days.

## Plasmids
### shRNA plasmids

pLKO.1 puro plasmids containing shRNA against mouse/rat *Drp1* (TRCN0000321170; 5′-CTATAATGCATGCACTATTTA-3′) were generated in this study. The shRNA sequence was obtained from the RNAi Consortium (TRC) via Dharmacon. A scrambled shRNA sequence was generated (5′-GTTCTAAGTTCCGTGTTCAGG-3′) and cloned into the pLKO.1 TRC cloning vector (Moffat et al., 2006) according to Addgene protocols (https://www.addgene.org/protocols/plko/). To generate pLKO.1 shRNA plasmids that express EGFP, CAG-EGFP was removed from pLenLox-shNL1-CAG-EGFP (Chih et al., 2006) and inserted between Kpn1 and Spel sites in pLKO.1 Puro, replacing the puromycin resistance gene in pLKO.1. Final constructs generated were pLKO.1-scrambled-GFP (RRID: Addgene_228736) and pLKO.1-shDrp1-GFP (RRID: Addgene_228737).

### PiggyBac plasmids

pGLAST-PBase was a gift from Dr. Joseph Loturco (Chen and LoTurco, 2012). shRNA sequences against mouse/rat *Miro1*

(TRCN0000330651; 5′-TGGACTGTGCTTCGACGATTT-3′), *Mfn1* (TRCN0000081400; 5′-GCAAGATTACAGGAGTTTCAA-3′), and *Drp1* (see above *shRNA plasmids*) were obtained from the RNAi Consortium (TRC) via Dharmacon. A scrambled shRNA sequence was generated (see above shRNA plasmids). To insert the hU6 promoter and shRNA in pPBCAG-mCherry-CAAX, a DNA fragment containing hU6 and shRNA was amplified from pLKO.1-shRNA using Phusion High-Fidelity DNA Polymerase (NEB) with primers that introduced SpeI restriction sites (forward primer: 5′-GGACTAGTCAGGCCCGAAGGAATAGAAG-3′; reverse primer: 5′-GGACTAGTGCCAAAGTGGATCTCTGCTG-3′). PCR products were purified, digested with SpeI, and ligated into pPBCAG-mCherry-CAAX at the SpeI restriction site. An analytical digest with EcoRI followed by sequencing was used to confirm the orientation of the inserted DNA fragment. The final plasmids generated were pPB-scrambled-mCherry-CAAX (RRID: Addgene_228738), pPB-sDrp1-mCherry-CAAX (RRID: Addgene_228739), pPB-shMiro1-mCherry-CAAX (RRID: Addgene_228741), and pPB-shMfn1-mCherry-CAAX (RRID: Addgene_228740).

## Other plasmids

hDrp1-YFP and hDrp1-K38E-CFP were gifts from Heidi McBride (König et al., 2021). MitoDsRed was acquired from Addgene (RRID: Addgene_55838).

## Biochemical assays
### mRNA extraction and cDNA preparation

Cells stored in TRIzol (#15596026; Invitrogen) were brought to room temperature and resuspended in a final volume of 1 ml of TRIzol. 200 µl of chloroform was added to each sample and mixed thoroughly. Samples were centrifuged at 12,000 *g* for 15 min at 4°C for phase separation, and the clear aqueous phase was collected. 2 µl of GlycoBlue Coprecipitant (15 mg/ml, #AM9515; Invitrogen) and 500 µl of isopropanol were added to each sample, centrifuged at 12,000 *g* for 10 min at 4°C, precipitating RNA as a blue pellet. The RNA pellet was rinsed in 75% ethanol, air-dried, and resuspended in 20 µl of nuclease-free water. mRNA concentration in each sample was quantified via Qubit RNA HS Assay Kit (#Q32852; Invitrogen). RNA samples were then diluted with nuclease-free water to match concentrations across all samples. cDNA libraries were then generated by incubating the samples with qScript cDNA SuperMix (#101414-102; VWR) and nuclease-free water for 5 min at 25°C, 30 min at 42°C, and 5 min at 85°C. The resulting cDNA was then diluted appropriately with nuclease-free water and stored at −80°C (https://www.protocols.io/view/mrna-extraction-and-cdna-preparation-bp2l623n5gqe/v1).

### Real-time qPCR

cDNA samples were plated in duplicate on a 96-well qPCR plate and incubated with Fast SYBR Green Master Mix (#4385616; Applied Biosystems), nuclease-free water, and the forward and reverse primers of interest at a ratio of 5 µl SYBR: 3 µl water: 0.5 µl forward primer: 0.5 µl reverse primer: 1 µl sample. Each sample was plated twice to ensure technical replicates. A no-cDNA sample (water with primers and Master Mix) served as

a negative control. Cycle threshold values were collected for each well and normalized to PPIA as a housekeeping gene (https://www.protocols.io/view/real-time-qpcr-j8nlk8mexl5r/v1). The sequences of forward (F) and reverse (R) primers used (5′→3′) are:

Drp1: (F) 5′-TTACGGTTCCCTAAACTTCACG-3′ and (R) 5′-GTCACGGGCAACCTTTTACGA-3′

Miro1: (F) 5′-TGGGCAGCACTGATAGAATAGA-3′ and (R) 5′-GCAAAGACCGTAGCACCAAAG-3′

Mfn1: (F) 5′-CCTACTGCTCCTTCTAACCCA-3′ and (R) 5′-AGGGACGCCAATCCTGTGA-3′

Cx43: (F) 5′-ACAGCGGTTGAGTCAGCTTG-3′ and (R) 5′-GAGAGATGGGGAAGGACTTGT-3′

GFAP: (F) 5′-GGGGCAAAAGCACCAAAGAAG-3′ and (R) 5′-GGGACAACTTGTATTGTGAGCC-3′

Vimentin: (F) 5′-CGTCCACACGCACCTACAG-3′ and (R) 5′-GGGGGATGAGGAATAGAGGCT-3′

PPIA: (F) 5′-GAGCTGTTTGCAGACAAAGTTC-3′ and (R) 5′-CCCTGGCACATGAATCCTGG-3′ LCN2: (F) 5′-TGGCCCTGAGTGTCATGTG-3′ and (R) 5′-CTCTTGTAGCTCATAGATGGTGC-3′

Cxcl10: (F) 5′-CCAAGTGCTGCCGTCATTTTC-3′ and (R) 5′-GGCTCGCAGGGATGATTTCAA-3′

Serpina3: (F) 5′-ATTTGTCCCAATGTCTGCGAA-3′ and (R) 5′-TGGCTATCTTGGCTATAAAGGGG-3′

Cp: (F) 5′-CTTAGCCTTGGCAAGAGATAAGC-3′ and (R) 5′-GGCCTAAAAACCCTAGCCAGG-3′

### Protein extraction and western blotting

Protein was extracted from cultured cells using RIPA lysis buffer (89900; Thermo Fisher Scientific). Cells were washed twice with ice-cold DPBS and incubated on ice in RIPA buffer for 5 min with occasional agitation. Cell lysates were collected, vortexed briefly, and centrifuged at 4°C at high speed for 10 min to pellet non-solubilized material. The supernatant was collected and stored at −80°C. To collect protein from astrocytes from Drp1 cWT and cKO mice at P21, astrocytes were isolated following a modified version of the Magnetic Activated Cell Sorting (MACS) protocol previously described (Holt et al., 2019). In brief, mice were anesthetized with 200 mg/kg tribromoethanol (avertin) and perfused with TBS/heparin, cortices rapidly dissected, and digested in papain (~7.5 U/ml) at 33°C for 45 min with pipette trituration every 15 min. Then homogenates were resuspended in MACS buffer (0.002 M EDTA, 0.2% milk peptone, 0.01 M HEPES, pH 7, 0.5% glucose, and 1X HBSS) and passed through a CellTrics 30-μm filter (04-004-2326; Sysmex). Cortical cells were then cleared from microglia by incubating with anti-CD11b human/mouse magnetic microbeads (130-093-634; Miltenyi Biotec) for 10 min at 4°C with rotation and passed through an LS separation column (130-042-401; Miltenyi Biotec) inserted into a Quadro MACS Multi Stand magnetic stand (Miltenyi Biotec), followed by three washes with MACS buffer. Flow-through was then spun down and resuspended in MACS buffer and incubated with blocking buffer and anti-ACSA2 mouse magnetic microbeads (130-097-678; Miltenyi Biotec, RRID: AB_2894998) for 30 min at 4°C with rotation to isolate astrocytes. Cell suspension was again passed through a separation column inserted into a

magnetic stand, rinsed three times with MACS buffer, and column retentate (ACSA2+ population, astrocytes) was pelleted. Supernatant was discarded, and the astrocyte pellet was resuspended in RIPA buffer containing protease inhibitors (4693132001; Roche) and incubated with rotation at 4°C for 20 min. Lysate was centrifuged at max speed at 4°C for 10 min, and the supernatant was collected and stored at −80°C.

Pierce Micro BCA Protein Assay Kit (23235; Thermo Fisher Scientific) was used to determine protein concentration, and lysates were subjected to western blot analysis and quantification using a SimpleWestern Jess (ProteinSimple) automated immunoassay system with a 12–230 kDa Chemiluminescence Separation module (SM-W001) and the associated manufacturer's protocol. Primary antibodies used were anti-Drp1 (rabbit, 1:10, RRID: AB_2895215), Connexin43 (rabbit, 1:10, RRID:AB_2294590), COX4 (rabbit, 1:10, RRID:AB_443304), GLAST (rabbit, 1:150, RRID: AB_2190597), GLUT1 (rabbit, 1:30, RRID:AB_2809254), and Kir4.1 (rabbit, 1:10, RRID:AB_2040120).

### Immunocytochemistry

Astrocyte–neuron co-cultures on glass coverslips were fixed on DIV 12 with warm 4% PFA for 7 min, washed three times with PBS, blocked in a blocking buffer containing 50% normal goat serum (NGS) and 0.4% Triton X-100 for 30 min, and washed in PBS. Samples were then incubated overnight at 4°C in primary antibodies diluted in blocking buffer containing 10% NGS. Primary antibodies used were: anti-GFP (chicken, 1:1,000, RRID:AB_10000240) and anti-RFP (rabbit, 1:1,000, RRID: AB_2209751). Cells were then washed with PBS, incubated in Alexa-Fluor–conjugated secondary antibodies (Ck 594 RRID: AB_2534099; Rb 594, RRID:AB_2534079; Ck 488 RRID:AB_2534096; Rb 488, RRID:AB_2925776; Rb 405, RRID:AB_2890548; Ms IgG1 647, RRID:AB_2535809) for 2 h at room temperature, and washed again in PBS. Coverslips were mounted onto glass slides with Vectashield mounting media containing DAPI (Vector Labs, H-1200-10), sealed with nail polish, and imaged on a BZ-X800 microscope (Keyence) with a Nikon S Plan Fluor ELWD ADM 40XC/0.60NA dry/oil objective and a 2/3-inch, 2.83 million pixel Peltier-cooled monochrome CCD (colorized with LC filter) camera, supporter by the BZ-X800 analyzer software. Images of astrocytes were acquired in red, green, and/or DAPI channels using a CCD camera. Astrocytes that contained a single nucleus, as revealed by DAPI stain, strongly expressed fluorescent markers, and did not overlap with other labeled astrocytes were selected for imaging (https://www.protocols.io/view/immunocytochemistry-14egn2556g5d/v1).

Astrocyte branch number and mitochondrial localization were analyzed using Seg_Astro MATLAB (RRID: SCR_001622, version R2022b, http://www.mathworks.com/products/matlab/) plugin. To quantify branch and mitochondrial number and size across the astrocyte arbor, Seg_Astro (see Data Availability to access plugin) first detects the astrocyte as well as the mitochondria by identifying locally significant regions based on order statistics (Wang et al., 2020). Specifically, for the detection of the astrocyte, due to the nonuniform intensity inside the cell, the initial detection contained many fragments. We further refined the detection by searching for the globally

optimal linkage, which is based on the graph design in which each node represents a fragment, and the edge weight between nodes represents the mean intensity of the brightest path between each pair of fragments. Then, the optimal linkage was obtained by finding the min-spanning tree of the graph. Once the astrocyte is detected, Seg_Astro then uses branch width and branchpoints to determine the astrocyte branch hierarchy and assign four types of astrocyte branches per cell: (1) soma and primary processes, (2) secondary processes, (3) fine processes, and (4) the terminal tips of processes (Fig. 2 C). Seg_Astro then bins mitochondria into these 4 branch types and outputs branch number, mitochondrial number, and mitochondrial size per branch type. Importantly, we excluded mitochondrial measurements from the soma and primary branches of astrocytes, as mitochondria form a dense network in these compartments that cannot be distinguished as discrete mitochondria for number and size quantification. Image acquisition and analysis were performed blinded to experimental conditions. Each independent experiment consisted of primary neurons and astrocytes isolated from a unique litter of WT rats of mixed sexes. The exact number of independent experiments and the exact number of cells analyzed are indicated in the figure legend for each experiment. Branch and mitochondrial numbers per condition were statistically analyzed using a nested *t* test or nested one-way ANOVA on GraphPad Prism 9.

## PALE

Late P0/early P1 mice were sedated by hypothermia until anesthetized, and 1 µl of plasmid DNA mixed with Fast Green Dye was injected into the lateral ventricle of one hemisphere using a pulled glass pipette. For mCherryCAAX labeling and shRNA knockdown experiments in pups from mito-EGFP mice crossed with WT CD1 mice, the 1 µl of DNA contained 0.8 µg of pGLAST-PBase, 0.7 µg of pPB-shRNA-mCherryCAAX, and 0.5 µg of pCAG-Cre (RRID: Addgene_13775). Following DNA injection, electrodes were oriented with the positive terminal above the visual cortex and the negative terminal below the chin, and 5 discrete 50-ms pulses of 100 V spaced 950 ms apart were applied. Pups were recovered on a heating pad, returned to their home cage, and monitored until collection at P4, P7, P14, P21, P60, or P180. In the case of plasmid expression in CD1 mice, assignment to experimental groups was randomly determined for each litter. The number of replicates for each experiment is indicated in the figure legends. All animals that appeared healthy at the time of collection were processed for data collection. All brain sections were examined for the presence of electroporated cells before staining and downstream analyses (https://dx.doi.org/10.17504/protocols.io.8epv5op66g1b/v1).

## Tamoxifen administration

Tamoxifen (T5648; Sigma-Aldrich) was dissolved in corn oil (8267; Sigma-Aldrich) at a concentration of 10 mg/ml and further diluted in corn oil to 1.25 mg/ml. 40 µl of the tamoxifen solution was injected into the milk spot using an insulin syringe, for a dose of 0.05 mg at P1, P2, and P3. Two tamoxifen injections (P1 and P2) were sufficient to turn on tdTomato expression in all astrocytes; however, a third dose of tamoxifen (at P3) was needed for

achieving a significant decrease of Drp1 levels in astrocytes (https://dx.doi.org/10.17504/protocols.io.dm6gp21b5lzp/v1).

## EdU administration

EdU stock solution from the Click-iT EdU Imaging Kit (C10640; Thermo Fisher Scientific) was diluted in DPBS to a concentration of 10 µM (2.5 µg of EdU/µl). At time points noted in Fig. 4, pups were weighed and injected with 15 µg of EdU per gram of animal weight (6 µl of working 10 µM EdU stock solution per gram of animal weight). Animals were collected for staining 16 h after EdU injection, and tissue sections were stained following the Click-iT EdU Imaging Kit manufacturer's protocol (https://dx.doi.org/10.17504/protocols.io.8epv5m5bdl1b/v1).

## Immunohistochemistry
### Sample preparation

Mice used for immunohistochemistry were anesthetized with 200 mg/kg tribromoethanol (avertin) and perfused with TBS/heparin and 4% PFA. Brains were collected and postfixed in 4% PFA overnight, cryoprotected in 30% sucrose, frozen in a solution containing 2 parts 30% sucrose and 1 part TissueTek O.C.T. (cat#4583; Sakura Finetek), and stored at –80°C. Floating coronal tissue sections of 30, 40, and 100 µm thickness were collected and stored in a 1:1 mixture of TBS/glycerol at –20°C. For immunostaining, sections were washed in 1× TBS containing 0.2% Triton X-100 (TBST), blocked in 10% NGS diluted in TBST, and incubated in primary antibody for 2–3 nights at 4°C with shaking. Primary antibodies used were anti-RFP (rabbit, 1:2,000, RRID:AB_2209751), GFP (chicken, 1:2,000, RRID:AB_10000240), Sox9 (rabbit, 1:2,000, RRID:AB_2239761), Connexin43 (rabbit, 1:100, RRID:AB_2294590), Drp1 (mouse, 1:100, RRID:AB_398424), Connexin30 (rabbit, 1:500, RRID:AB_2533979), and GFAP (rabbit, 1:1,000, RID:AB_10013382). Following primary incubation, sections were washed in TBST, incubated in Alexa-Fluor–conjugated secondary antibodies diluted 1:200 (Life Technologies) for 2–3 h at room temperature, washed with TBST, and mounted onto glass slides using a homemade mounting media (90% glycerol, 20 mM Tris, pH 8.0, and 0.5% n-propyl gallate) and sealed with nail polish. For primary antibodies produced in mouse, isotype-specific secondary antibodies were always used (e.g., goat anti-mouse IgG1) to avoid excessive background staining. For DAPI staining, DAPI (1:50,000) was added to the secondary antibody solution for the final 10 min of incubation (https://dx.doi.org/10.17504/protocols.io.261ge876og47/v1).

### Astrocyte territory volume analysis

To assess the territory volume of individual astrocytes in the developing mouse cortex, 100-µm–thick floating sections containing V1 astrocytes labeled sparsely via PALE with mCherry-CAAX and mito-EGFP were collected. In shDrp1 conditions where astrocytes cluster together, only single astrocytes on the periphery of clusters were used for this analysis. Images containing an entire astrocyte (50–60-µm z-stack) were acquired at 21°C on an Olympus FV 3000 microscope with an Olympus Plan APO 60× 1.4 NA oil objective (Immoil-f30cc), supported by the Olympus FV315-SW software. Criteria for data inclusion required that the entirety of the astrocyte could be captured within

a single brain section and that the astrocyte was located in layers 2/3, 4, or 5 of the visual cortex. Astrocytes in which the entire astrocyte could not be captured within the section or were located in other layers or outside of the visual cortex were excluded from the study and not imaged. Imaged astrocytes were analyzed using Imaris Bitplane software (RRID: SCR_007370, version 9.9.9, https://imaris.oxinst.com/packages) as described previously (Stogsdill et al., 2017). Briefly, surface reconstructions were generated, and the Imaris Xtensions "Visualize Surface Spots" and "Convex Hull" were used to create an additional surface render representing the territory of the astrocyte. The volume of each territory was recorded, and astrocyte territory sizes from biological replicates were analyzed across experimental conditions using a nested *t* test. The number of mice and cells/mouse analyzed are specified in the figure legend for each experiment.

### NIV analysis

To measure the extent of astrocyte infiltration into the neuropil, astrocyte images were acquired from mCherry-CAAX and mito-EGFP–labeled astrocytes in 40-µm brain sections. Images were acquired at 21°C on an Olympus FV3000 microscope with an Olympus Plan APO 60× 1.4 NA oil objective (Immoil-f30cc) and a 2× zoom, supported by the Olympus FV315-SW software. Criteria for inclusion required the astrocyte to be located within layers 2/3, 4, or 5 of the visual cortex; express the fluorescent label; include their soma in the z-stack capture; and have at least 15 µm in the z-dimension contained within the section. Astrocytes that did not meet these criteria were excluded from the study and not imaged. For each astrocyte, three regions of interest (ROIs) (12.65 µm × 12.65 µm × 10 µm) containing the neuropil and devoid of cell soma, large branches, and end feet were chosen and reconstructed using the surface tool in Imaris. The surface volume of each ROI was calculated, and the three data points from each astrocyte were averaged. Data from biological replicates were analyzed using a nested *t* test. The number of mice and cells/mouse analyzed are specified in the figure legend for each experiment.

### Mitochondrial average volume, number, and total volume analysis

To measure mitochondrial size (average volume), number, and total volume within whole astrocytes or distal astrocyte processes, astrocyte images were captured as described above in Astrocyte territory volume analysis (for whole astrocyte mitochondrial volume) or NIV analysis (for distal NIV mitochondrial volume). In Imaris, the astrocyte volume rendered above was used to mask the mitochondrial EGFP channel. Either whole or distal mitochondria were reconstructed using the surface volume tool in Imaris (surface detail: 0.1, background subtraction: 0.5, volume filter: 0.01). Data from biological replicates were analyzed using a nested *t* test. The number of mice and cells/mouse analyzed are specified in the figure legend for each experiment.

### Quantification of astrocyte cell or cluster area

To analyze astrocyte cell or cluster areas of sparse Drp1 knockdown PALE experiments, three 40-µm sections per animal were imaged at 21°C on an Olympus FV 3000 microscope with an Olympus UPlanSApo 10×/0.40 dry objective, supported by the Olympus FV315-SW software, and loaded into ImageJ. The astrocyte mCherry channel, the RFP channel, was selected. The image was then stacked into a max projection z-stack, and a Gaussian blur of Sigma-Aldrich radius 1.00 was applied. A threshold was then applied to the image to include signal from astrocytes in the cortex above the subventricular zone. Then, using ImageJ analysis, the area of each cluster above 500 pixels per image was recorded. The average cluster area per image was calculated, and three technical replicates per animal were averaged for analysis. Data from biological replicates were analyzed using a nested *t* test. The number of mice and cells/mouse analyzed are specified in the figure legend for each experiment.

### Nearest and multiple neighbor analysis

To measure the distance between astrocyte neighbors in sparse Drp1 knockdown PALE experiments, images of astrocytes or astrocyte clusters from 40-µm sections were captured at 21°C on a Leica Stellaris 8 Confocal with a 63×/1.40 HC PL APO CS2 (11506350) oil objective (Leica F Immersion liquid [Oil] 11513859) and a Leica K5 sCMOS (14402610) camera, supported by the Leica Application Suite X (LASX, build 4.5.0.25531, RRID: SCR_013673, https://www.leica-microsystems.com/products/microscope-software/p/leica-las-x-ls/) software and loaded into ImageJ. Composite images merging the DAPI channel for all nuclei, the Sox9 channel for astrocytic nuclei, and the RFP channel for astrocytes were generated. Using the line tool, a line was manually drawn from the edge of one Sox9+ nucleus within mCherry to the closest Sox9+ nucleus, and the length of the line was measured. Each distance between neighbors was recorded from the same animal as technical replicates. Data from biological replicates were analyzed using a nested *t* test. To quantify the number of Sox9+ nuclei, the composite images from above were used, and the number of Sox9+ nuclei within the mCherry signal was manually counted. The number of Sox9+ nuclei per cell or cluster was recorded as technical replicates per mouse. The number of mice and cells/mouse analyzed are specified in the figure legend for each experiment. In the Drp1 conditional deletion mouse model, V1 cortical images of 30-µm sections were taken at 21°C on a Leica Stellaris 8 Confocal with a 20×/0.75 HC PL APO CS2 (11506343) immersion objective (Leica F Immersion liquid [Oil] 11513859) at 0.75 optical zoom and a Leica K5 sCMOS (14402610) camera, supported by the Leica Application Suite X (LASX, build 4.5.0.25531, RRID:SCR_013673, https://www.leica-microsystems.com/products/microscope-software/p/leica-las-x-ls/) software. Single optical section images were loaded onto Imaris with a z-size of 0.01 µm. Selecting the tdTomato channel, soma spots were generated with an estimated XY diameter of 7.5 µm and with background subtraction using the Imaris Spots tool. Object–Object statistics were collected for the spots generated. Once these soma spots were generated, the distance to the nearest neighbor and the average distance to 9 neighbors were analyzed. Data from biological replicates were analyzed using a nested *t* test and a Kolmogorov–Smirnov test. The number of mice and cells/mouse analyzed are specified in the figure legend for each experiment.

### Aldh1l1-Drp1 mouse model recombination efficiency analysis with Sox9

In the Drp1 conditional deletion mouse model, V1 cortical images of 30-μm sections were taken at 21°C on a Leica Stellaris 8 Confocal with a 20×/0.75 HC PL APO CS2 (11506343) immersion objective (Leica F Immersion liquid [Oil] 11513859) at 0.75 optical zoom and a Leica K5 sCMOS (14402610) camera, supported by the Leica Application Suite X (LASX, build 4.5.0.25531, RRID: SCR_013673, https://www.leica-microsystems.com/products/microscope-software/p/leica-las-x-ls/) software. A z-stack of 5 μm with a step size of 1 μm was captured. Images were loaded onto Imaris, and the surface tool was used to create a mask of the tdTomato channel. This tdTomato mask was used to sort the Sox9+ signal within tdTomato and outside tdTomato separately. Soma spots for Sox9+ nuclei within and outside of tdTomato were generated with an estimated XY diameter of 6 μm and with background subtraction using the Imaris Spots tool. Once these spots were generated, the number of Sox9+ spots was collected for both inside and outside tdTomato signal. The total number of Sox9+ spots was calculated as the sum of Sox9+ spots inside and outside tdTomato. The percent recombination efficiency was quantified as the number of Sox9+ spots within the tdTomato signal divided by the total number of Sox9+ spots per field of view. The number of mice and cells/mouse analyzed are specified in the figure legend for each experiment.

### Cx43 volume analysis

To quantify the amount of Cx43 within astrocytes in the sparse Drp1 knockdown PALE model, images (0.5-μm z-step, 10-μm z-stacks) from 40-μm sections of astrocytes or clusters were taken at 21°C on a Leica Stellaris 8 Confocal with a 93×/1.30 HC PL APO CS2 (11506417) glycerol objective (Leica G Immersion liquid [Glycerol] 11513910) and a Leica K5 sCMOS (14402610) camera, supported by the Leica Application Suite X (LASX, build 4.5.0.25531, RRID:SCR_013673, https://www.leica-microsystems.com/products/microscope-software/p/leica-las-x-ls/) software. Criteria for inclusion required the astrocyte to be located within layers 2/3, 4, and 5 of the visual cortex, express the fluorescent label, and have at least 10 μm in the z-dimension contained within the section. For each astrocyte or astrocyte cluster, the Imaris surface tool was used to render the astrocyte volume. This astrocyte volume was then used to mask the Cx43 channel, and then the Imaris surface tool was used to generate surface volumes of Cx43 puncta within the astrocyte volume territory only (surface detail: 0.1, background subtraction: 0.5, volume filter: 0.01). Data from biological replicates were analyzed using a nested one-way ANOVA. The number of mice and cells/mouse analyzed are specified in the figure legend for each experiment.

### Astrocyte density analysis

For quantifying the density of astrocytes in the Drp1 conditional deletion model, V1 cortical images of 30-μm sections were taken at 21°C on a Leica Stellaris 8 Confocal with a 20×/0.75 HC PL APO CS2 (11506343) immersion objective (Leica F Immersion liquid [Oil] 11513859) at 0.75 optical zoom and a Leica K5 sCMOS (14402610) camera, supported by the Leica Application Suite X

(LASX, build 4.5.0.25531, RRID:SCR_013673, https://www.leica-microsystems.com/products/microscope-software/p/leica-las-x-ls/) software. Using the tdTomato channel on ImageJ, a rectangular ROI that encompassed all cortical layers of the visual cortex was selected and saved to the ROI manager in ImageJ. Within the multilayer rectangular ROI, ROIs for each cortical layer were drawn and saved. The area of each ROI was recorded. These imaging parameters and ROIs were also used for the Cortical astrocyte coverage analysis and Drp1 protein expression analyses by immunohistochemistry below. The "Show-all" feature on the ROI manager was selected so that the boundaries between cortical layers were shown. Using the Cell Counter plug-in (RRID:SCR_025376, https://imagej.net/ij/plugins/cell-counter.html), tdTomato+ somas were manually counted. The number of somas across the cortex was calculated as the sum of tdTomato+ somas in all cortical layers. Density of tdTomato+ somas was calculated by dividing the number of tdTomato+ somas by their corresponding ROI area. GFAP+ cell density was calculated and analyzed in the same manner, using the GFAP channel instead of the tdTomato channel. Three sections were analyzed as technical replicates per mouse. Data from biological replicates were analyzed using a two-way ANOVA with Sidak's multiple comparisons test. The number of mice and cells/mouse analyzed are specified in the figure legend for each experiment.

### Cortical astrocyte coverage analysis

To quantify astrocyte coverage across the cortex in the Drp1 conditional deletion model, cortical images were captured and divided into multiple ROIs for analysis as described above in Astrocyte density analysis. Using the threshold tool in ImageJ, the tdTomato channel coverage was binarized and saved as a tdTomato mask. Then, selecting each cortical ROI, the area of tdTomato coverage from the mask generated across the cortex and per layer was collected. tdTomato coverage was defined as the percent of tdTomato+ area divided by the density of tdTomato+ cells. GFAP coverage was calculated and analyzed in the same manner, using the GFAP channel instead of the tdTomato channel. GFAP coverage was likewise normalized by tdTomato density. Three sections were analyzed as technical replicates per mouse. Data from biological replicates were analyzed using a two-way ANOVA with Sidak's multiple comparisons test. The number of mice and cells/mouse analyzed are specified in the figure legend for each experiment.

### Drp1 protein expression analyses by immunohistochemistry

For quantifying Drp1 protein levels in the Drp1 conditional deletion model, cortical images were captured and divided into multiple ROIs for analysis as described above in Astrocyte density analysis. Then a mask of the tdTomato channel was applied to the Drp1 channel, and the mean gray value of Drp1 within astrocytes was collected per layer and across the cortex using ImageJ. The collected values were normalized to tdTomato+ soma density. Three sections were analyzed as technical replicates per mouse. Data from biological replicates were analyzed using a two-way ANOVA with Sidak's multiple comparisons test. The number of mice and cells/mouse analyzed are specified in the figure legend for each experiment.

### Cx43 protein expression analysis by immunohistochemistry

To measure Cx43 protein levels in astrocytes in the Drp1 conditional deletion model, cortical images were captured as described above in Astrocyte density analysis. ImageJ was used to subtract background from the Cx43 channel with a rolling ball radius of 50 units. Following background subtraction, individual astrocytes within layers 2/3, 4, or 5 were outlined with the freehand tool using tdTomato to determine astrocyte territories, and astrocyte outlines were saved as ROIs. Astrocyte ROIs were applied to the Cx43 channel, and the mean gray value of the Cx43 signal was collected per astrocyte and normalized by the ROI area. Three sections were analyzed as technical replicates per mouse. Data from biological replicates were analyzed using a nested *t* test and a Kolmogorov–Smirnov test. The number of mice and cells/mouse analyzed are specified in the figure legend for each experiment.

### Cortical thickness analysis

To measure cortical thickness in Drp1 cWT and cKO mice, images of V1 cortices were taken at 21°C on a Leica Stellaris 8 Confocal with a 10×/0.40 HC PL APO CS2 (11506424) dry objective and a Leica K5 sCMOS (14402610) camera, supported by the Leica Application Suite X (LASX, build 4.5.0.25531, RRID:SCR_013673, https://www.leica-microsystems.com/products/microscope-software/p/leica-las-x-ls/) software. Images were loaded into ImageJ, and the line tool was used to record length. Three lines per section were drawn from the subventricular zone to the Pia. The lengths of these lines were averaged per section for 3 sections per mouse, and data from each section were plotted as a technical replicate per mouse. Data from biological replicates were analyzed using a nested *t* test.

### Quantification and statistical analysis

All statistical analyses were performed using GraphPad Prism (RRID: SCR_002798, version 9, https://www.graphpad.com/scientific-software/prism/). The exact number of replicates, specific statistical tests, and P values for each experiment are indicated in the figures and figure legends. All data are represented as mean ± SEM, and individual data points are shown for all data, where applicable. Where indicated, nested *t* tests were applied with an alpha threshold of 0.05 for adjusted P value. A Geisser-Greenhouse correction was used for both one-way and two-way ANOVA analyses. All experimental animals that appeared healthy at the time of tissue collection were processed for data collection. Specific details for inclusion, exclusion, and randomization are included in the specific subsections of the Materials and methods section.

### Online supplemental material

Fig. S1 contains the qPCR validation of shRNAs targeting mouse and rat Drp1, mouse Mfn1, and mouse Miro 1. Fig. S2 emphasizes how silencing astrocytic mitochondrial dynamics modifies mitochondrial numbers and size. Fig. S3 establishes that Drp1 KD astrocyte clusters are not a result of increased astrocyte proliferation, persist into adulthood, and express gap junction protein Cx30. Fig. S4 characterizes the astrocyte-specific Drp1 cKO model, demonstrating decreased Drp1 protein expression, Cre-mediated recombination efficiency, and additional astrocyte reactivity markers probed in isolated astrocytes from Drp1 cKO and cWT mice.

### Data availability

The raw data and analyses generated during this study can be accessed on Zenodo at DOI: https://doi.org/10.5281/zenodo.15328429. The Seg_Astro code can be found at https://zenodo.org/records/15311182. Any additional information can be requested from the lead contact.

## Acknowledgments

We thank Drs. Romain Cartoni and Chantell Evans for critical feedback on this manuscript. We thank Sarah Johnson and Donna Porter for their excellent technical help.

This work was supported by a Paul and Daisy Soros Fellowship (2020) and an HHMI Gilliam Fellowship (GT15076) to Maria Pia Rodriguez Salazar, a Chan Zuckerberg Initiative Neurodegeneration Challenge Network Collaborative grant (DAF2018-191999 and DAF2021-237435) to Cagla Eroglu, an National Institutes of Health grant U19-NS123719 to Cagla Eroglu and Guoqiang Yu, and by the joint efforts of Michael J. Fox Foundation for Parkinson's Research and the Aligning Science Across Parkinson's (ASAP) initiative. MJFF administers the grant ASAP-020607 to Cagla Eroglu for ASAP and the Michael J Fox Foundation. Dr. Cagla Eroglu is an HHMI investigator.

IP rights notice: This article is subject to HHMI's Open Access to Publications policy. HHMI lab heads have previously granted a nonexclusive CC BY 4.0 license to the public and a sublicensable license to HHMI in their research articles. Pursuant to those licenses, the author-accepted manuscript of this article can be made freely available under a CC BY 4.0 license immediately upon publication.

Author contributions: Maria Pia Rodriguez Salazar: conceptualization, data curation, formal analysis, funding acquisition, investigation, methodology, project administration, resources, software, supervision, validation, visualization, and writing—original draft, review, and editing. Sprihaa Kolanukuduru: formal analysis, investigation, methodology, validation, visualization, and writing—original draft, review, and editing. Valentina Ramirez: conceptualization, data curation, formal analysis, investigation, methodology, software, visualization, and writing—original draft. Boyu Lyu: software. Gracie SeiRise Manigault: conceptualization, formal analysis, investigation, validation, and visualization. Gabrielle Sejourne: investigation. Hiromi Sesaki: resources and writing—review and editing. Guoqiang Yu: formal analysis, funding acquisition, investigation, methodology, software, supervision, visualization, and writing—review and editing. Cagla Eroglu: conceptualization, funding acquisition, project administration, and writing—original draft, review, and editing.

Disclosures: The authors declare no competing interests exist.

Submitted: 22 October 2024

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

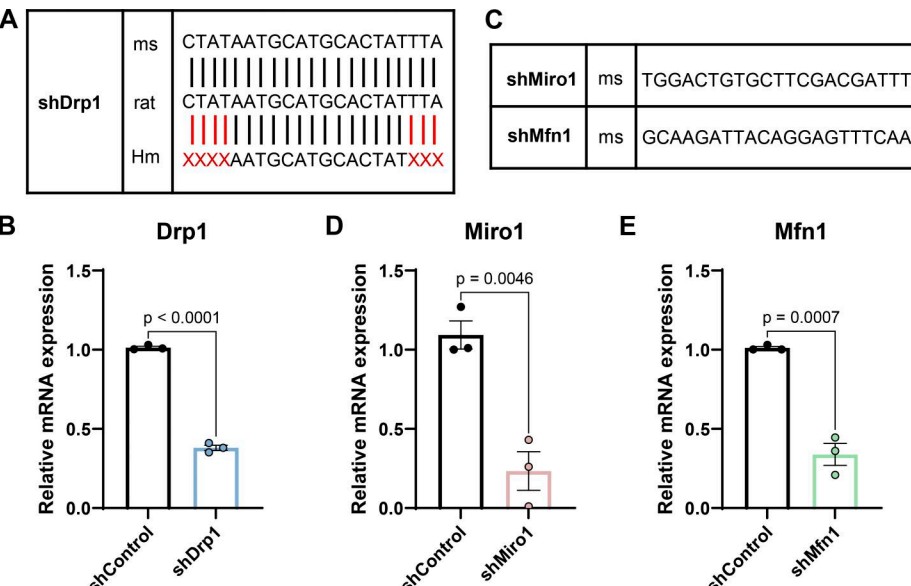

Figure S1. **Validation of shRNA tools. (A)** Schematic of shDrp1 sequence aligned to the mouse, rat, and human genomes. **(B)** Drp1 relative mRNA expression in control and shDrp1-transfected mouse 3T3 cells. **(C)** Schematic of shMiro1 and shMfn1 sequences aligned to the mouse genome. **(D and E)** Miro1 and (E) Mfn1 relative mRNA expression in control and shDrp1-transfected mouse 3T3 cells. Data are mean ± SEM. *n* = 3 independent experiments. Unpaired two-tailed *t* test.

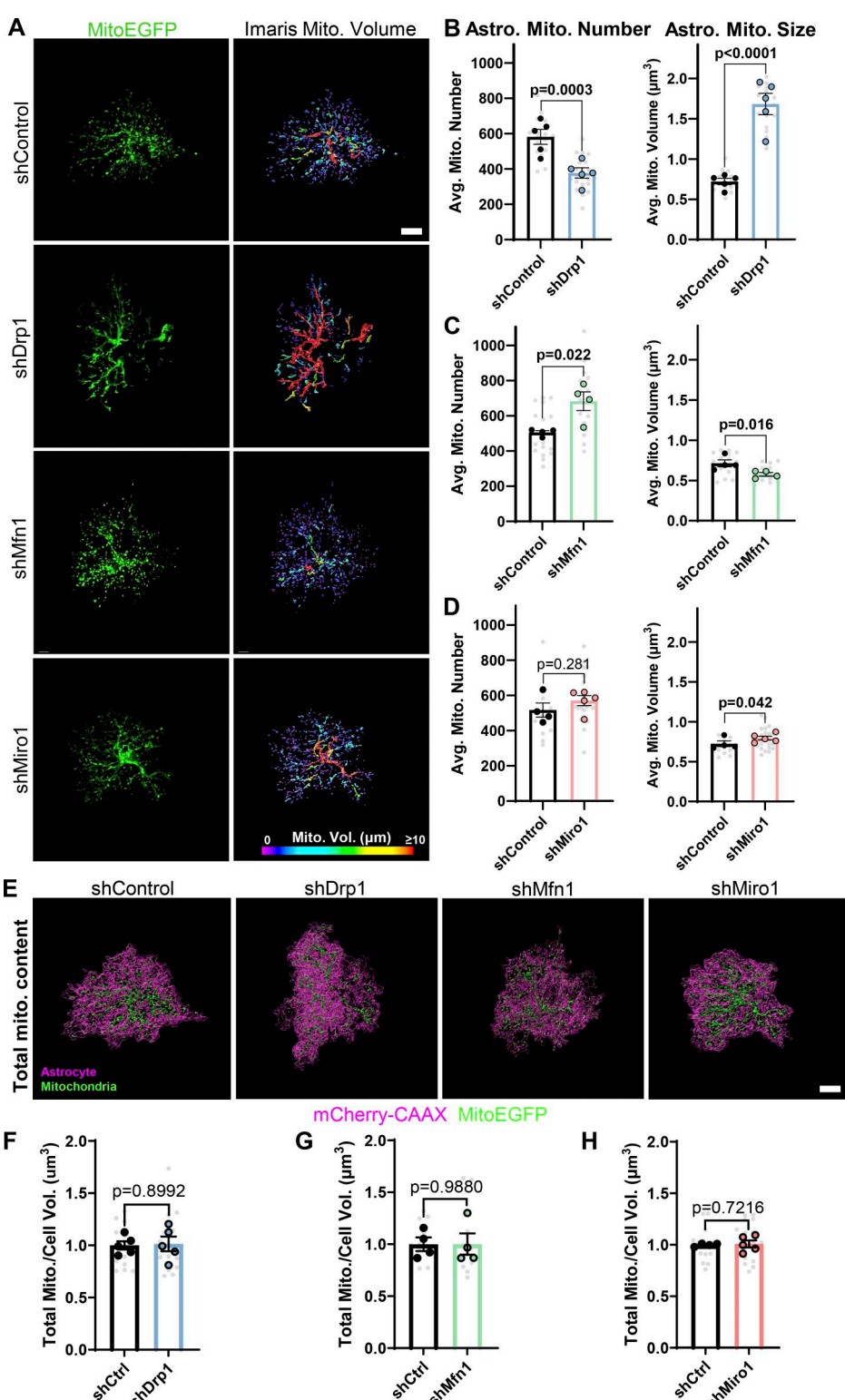

Figure S2.   **Genetic manipulation of mitochondrial dynamics *in vivo* alters mitochondria number and size. (A)** Representative images (left column) and Imaris reconstructions (right column) of V1 P21 astrocyte mitochondria expressing EGFP (green) and shRNA against Drp1 (shDrp1), Mfn1 (shMfn1), Miro1 (shMiro1), or scrambled control (shControl). Scale bar, 10 μm. **(B–D)** Quantification of astrocyte average mitochondrial number (left graphs) and average mitochondrial size (right graphs) in shDrp1, shMfn1, and shMiro1 conditions, respectively. *N* = 4–5 male and female mice/condition (large circles), *n* = 2–4 cells/mouse, 10–15 cells total/condition (small gray dots). Data are mean ± SEM. Nested *t* test. **(E)** Representative images of Imaris reconstructed V1 P21 astrocytes expressing mCherry-CAAX–tagged shRNA (magenta) against Drp1 (shDrp1), Mfn1 (shMfn1), Miro1 (shMiro1), or scrambled control (shControl) and their EGFP mitochondria (green). Scale bar, 10 μm. **(F–H)** Quantification of total mitochondrial volume normalized to cell volume in (F) shDrp1, (G) shMfn1, and (H) shMiro1 conditions, respectively. *N* = 4–5 male and female mice/condition (large circles), *n* = 2–4 cells/mouse, 10–15 cells total/condition (small gray dots). Data are mean ± SEM. Nested *t* test.

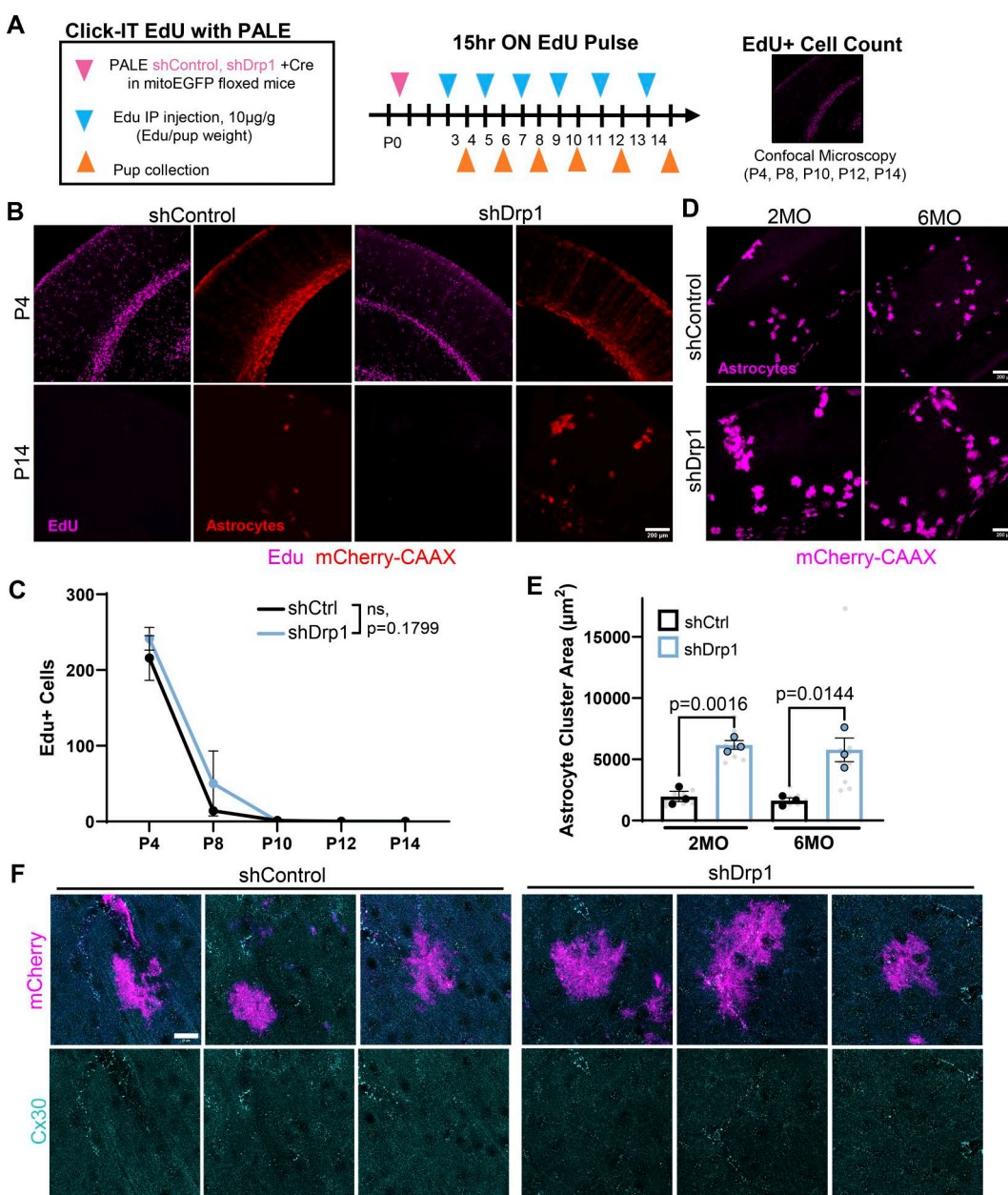

Figure S3. **Characterization of shDrp1-induced astrocyte clustering. (A)** Overview of PALE combined with EdU labeling. **(B)** Representative images of V1 cortices with astrocytes expressing mCherry-CAAX shRNA (red) against Drp1 (shDrp1) or scrambled control (shControl) and EdU labeling (magenta) at P4 and P14. Scale bar, 200 μm. **(C)** Quantification of Edu+ cells in shControl or shDrp1 cortices at P4, P8, P10, P12, and P14. N = 3–4 male and female mice/condition, three cortical section images/mouse. Data points are time point averages ± SEM. Two-way ANOVA with Sidak's multiple comparisons test. **(D)** Representative images of V1 cortices with astrocytes expressing mCherry-CAAX shRNA (magenta) against Drp1 (shDrp1) or scrambled control (shControl) at 2 and 6 mo. Scale bar, 200 μm. **(E)** Quantification of astrocyte cell or cluster area per field of view at 2 and 6 mo. N = 3 male and female mice/condition (large circles), three cortical section images/mouse (small gray dots). Data points are mean ± SEM. Nested t test. **(F)** Representative images of V1 astrocytes from three mice per condition expressing mCherry-CAAX and shControl (top two rows) and shDrp1 (bottom two rows) stained with an antibody against Cx30 (cyan) at P21. Scale bars, 20 μm.

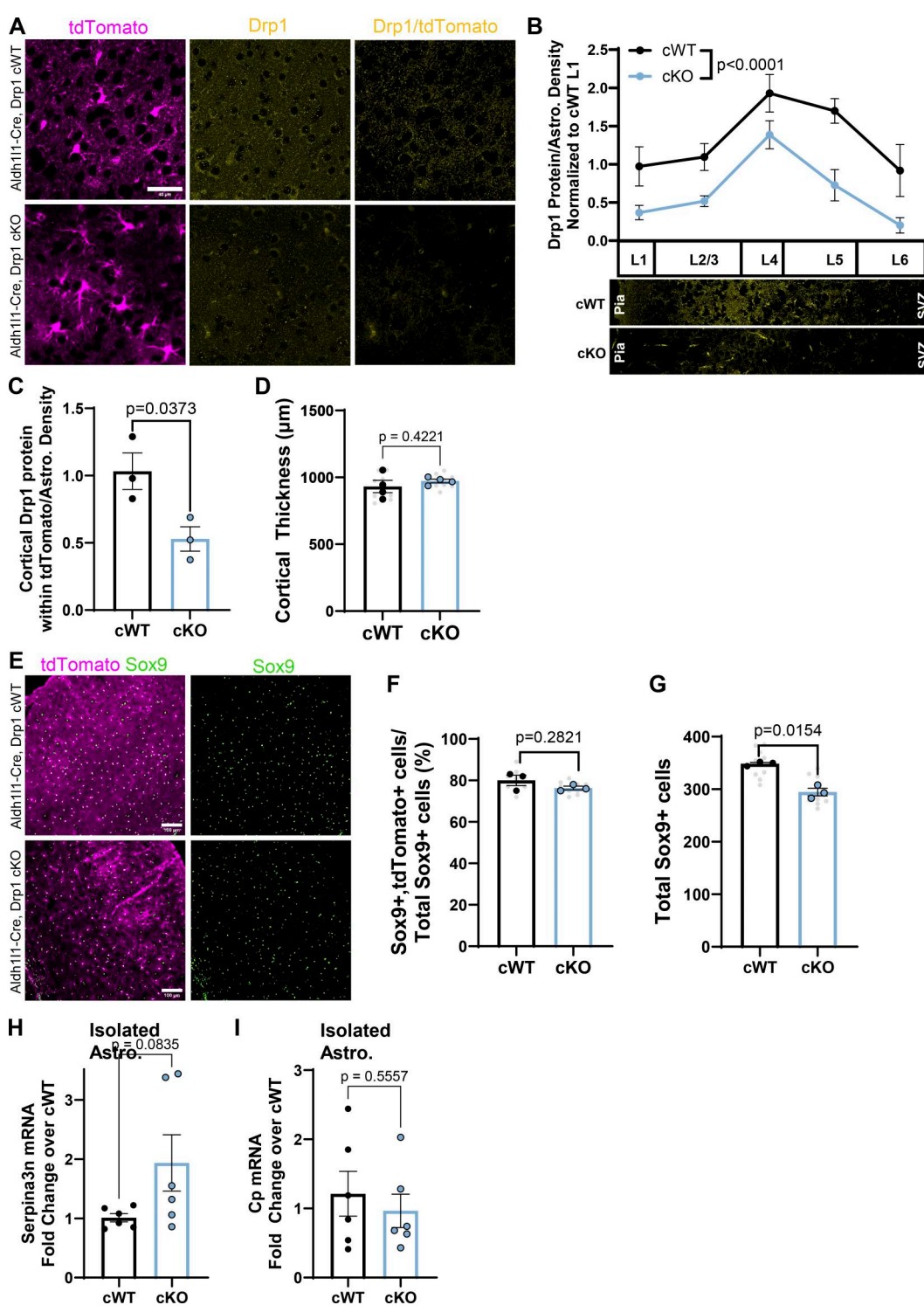

Figure S4.  **Astrocyte-specific Drp1 cKO mouse characterization. (A)** Representative images of Drp1 cWT and cKO V1 cortices with tdTomato+ astrocytes (magenta) stained with Drp1 (yellow) and showing Drp1 masked by tdTomato (last column) at P21. Scale Bar, 40 μm. **(B and C)** Quantification of Drp1 mean gray value within tdTomato area normalized to astrocyte density per cortical layer (B) and across the whole cortex (C). N = 3–4 male and female mice/condition, 2–3 cortical images/mouse. Data points are mean ± SEM. Two-way ANOVA with Sidak's multiple comparisons test for B, and an unpaired, two-tailed t test for C. **(D)** Quantification of cortical thickness in Drp1 cKO compared with cWT mice. N = 4 male and female mice/condition (large circles), n = 2–3 cortical images/mouse. Data are mean ± SEM. Unpaired, two-tailed t test. **(E)** Representative images of Drp1 cWT and cKO V1 cortices with tdTomato+ astrocytes (magenta) stained with Sox9 (green) at P21. Scale Bar, 100 μm. **(F and G)** Quantification of (F) percent Sox9 and tdTomato double-positive astrocytes out of total Sox 9-positive nuclei and (G) total Sox 9-positive nuclei per cortical field of view in Drp1 cWT and cKO mice. N = 3 male and female mice/condition, n = 3 cortical images/mouse (small gray dots). Data points are mean ± SEM. Nested t test. **(H and I)** Quantification of (H) Serpina3n and (I) Cp mRNA from immunopurified astrocytes from Drp1 cWT and cKO mice. N = 6 male and female mice/condition. Data are mean ± SEM. Unpaired two-tailed t test.

