## [Peer Review File · The Journal of Cell Biology]

Mitochondrial fission controls astrocyte morphogenesis and organization in the cortex

Maria Pia Rodriguez Salazar, Sprihaa Kolanukuduru, Valentina Ramirez, Boyu Lyu, Gracie Manigault, Gabrielle Sejourne, Hiromi Sesaki, Guoqiang Yu, and Cagla Eroglu

Corresponding Author(s): Cagla Eroglu, Duke University School of Medicine

Review Timeline:

Submission Date:	2024-10-22
Editorial Decision:	2025-01-03
Revision Received:	2025-05-16
Editorial Decision:	2025-06-18
Revision Received:	2025-07-02

Monitoring Editor: Marc Freeman

Scientific Editor: Andrea Marat

Transaction Report:

DOI: <https://doi.org/10.1083/jcb.202410130>

January 3, 2025

Re: JCB manuscript #202410130

Cagla Eroglu
Duke University School of Medicine

Dear Cagla,

Thank you for submitting your manuscript entitled "Mitochondrial fission controls astrocyte morphogenesis and organization in the cortex". The manuscript was assessed by expert reviewers, whose comments are appended to this letter. We invite you to submit a revision if you can address the reviewers' key concerns, as outlined here.

You will see that the reviewers are overall positive about the quality and potential impact of your manuscript. They have provided constructive suggestions, which we hope you agree will further improve your study and that you will be able to address in a revised study.

GENERAL GUIDELINES:

Text limits: Character count for an Article is < 40,000, not including spaces. Count includes title page, abstract, introduction, results, discussion, and acknowledgments. Count does not include materials and methods, figure legends, references, tables, or supplemental legends.

Figures: Articles may have up to 10 main text figures. Figures must be prepared according to the policies outlined in our Instructions to Authors, under Data Presentation, <https://jcb.rupress.org/site/misc/ifora.xhtml>. All figures in accepted manuscripts will be screened prior to publication.

Supplemental information: There are strict limits on the allowable amount of supplemental data. Articles may have up to 5 supplemental figures. Up to 10 supplemental videos or flash animations are allowed. A summary of all supplemental material should appear at the end of the Materials and methods section.

Please note that JCB now requires authors to submit Source Data used to generate figures containing gels and Western blots with all revised manuscripts. This Source Data consists of fully uncropped and unprocessed images for each gel/blot displayed in the main and supplemental figures. File names for Source Data figures should be alphanumeric without any spaces or special characters (i.e., SourceDataF#, where F# refers to the associated main figure number or SourceDataFS# for those associated with Supplementary figures). The lanes of the gels/blots should be labeled as they are in the associated figure, the place where cropping was applied should be marked (with a box), and molecular weight/size standards should be labeled wherever possible. Source Data files will be made available to reviewers during evaluation of revised manuscripts and, if your paper is eventually published in JCB, the files will be directly linked to specific figures in the published article.

The typical timeframe for revisions is three to four months. If you anticipate any difficulties in meeting this aforementioned revision time limit, please contact us and we can work with you to find an appropriate time frame for resubmission. Please note that papers are generally considered through only one revision cycle, so any revised manuscript will likely be either accepted or rejected.

Thank you for this interesting contribution to Journal of Cell Biology. You can contact us at the journal office with any questions at cellbio@rockefeller.edu.

Sincerely,

Marc Freeman
Monitoring Editor

Andrea L. Marat
Deputy Editor

Journal of Cell Biology

Reviewer #1 (Comments to the Authors (Required)):

In the current manuscript "Mitochondrial fission controls astrocyte morphogenesis and organization in the cortex', Salazar et al., describe a mechanism whereby during the process of astrocyte maturation and morphogenesis, mitochondria increase in number and decrease in size to facilitate organization into the tiny sub compartments of astrocytes (PAPs) which enwrap neuronal synapses. If mitochondrial fission is disrupted by the conditional deletion of Drp1, a protein critical for mitochondrial fission, PAP formation and cortical astrocyte organization is disrupted. Overall, this is a well-written and interesting manuscript that provides novel information related to astrocyte mitochondrial dynamics during early brain development. However, there are concerns that limit enthusiasm for the manuscript as written. These concerns are noted below.

- 1) The relationship between Drp1, mitochondrial fission and the specificity or link to Cnx43 needs further clarification. Cnx43 is a PAP protein, and PAPs are disrupted in the model. The authors need to clarify if the effects on Cnx43 are specific, or if PAP proteins in general are disrupted by performing immunohistochemistry and Western blotting for other PAP related proteins (ATPase, Glt1, Kir4.1, Glst, etc...). The findings would be interesting either way.
- 2) The manipulations are performed very early in astrocyte development, when proliferation, local expansion and migration are still occurring. Data indicates that KO of Drp1 leads to altered total number of astrocytes (as assessed by soma counts). However, manipulating Drp1, may also impact astrocyte migration to their proper location and thus nearest neighbor distance/astrocyte organization. The way the manuscript is written, this is directly linked to Cnx43 expression, which may or may not have anything to do with this observation. Could this be explored in a culture system (i.e. do astrocytes clump in a simplified culture system with no neurons around)? At a minimum, this possibility should be discussed.
- 3) Related to the above comment, it would be useful to manipulated Drp1 a later in development when proliferation and migration are complete, and morphogenesis occurs
- 4) Given that GFAP and vimentin are directly tied to astrocyte morphology, additional astrocyte reactive targets need to be validated to demonstrate these astrocytes are indeed reactive.
- 5) The analysis done in figure 1 c-d, should be repeated in at least one of the in vivo Drp1 manipulations to show that Drp1 KO results in larger, fewer, less distal mitochondria when fission is disrupted, particularly because total mito # was not observed to change in Drp1 shRNA experiments.
- 6) There is little consideration given to how and if Drp1 manipulation, particularly globally is likely to impact the astrocyte cellular metabolism, oxidative phosphorylation, glutamine/glutamate metabolism, ATP production, etc. This may be directly related to astrocyte reactivity, yet this is not discussed either.

Minor:

- 7) Please include citation - introduction, lines 55-56, '...ramified processes make up more than one-half the brain parenchyma by infiltrating the neuropil'

Reviewer #2 (Comments to the Authors (Required)):

The manuscript of Rodriguez Salazar et al., presents an important and novel exploration of how mitochondrial fission, regulated by Dynamin-related protein 1 (Drp1), impacts astrocyte morphogenesis and cortical organization during postnatal brain development. The authors convincingly demonstrate that Drp1 loss disrupts mitochondrial dynamics, reduces astrocyte complexity, induces clustering, and dysregulates the gap junction protein Connexin 43 (Cx43). The study is well-executed, employing both in vivo and in vitro models, as well as advanced imaging and quantification tools.

While the study makes significant contributions to the understanding of mitochondrial dynamics in astrocytes, a key mechanistic gap limits the impact of the findings. Specifically, the role of Cx43 in rescuing Drp1-related astrocyte phenotypes remains

unexplored, leaving an important question unanswered.

Major Concerns

1. Lack of Functional Rescue Experiments with Cx43:

The authors convincingly show that Drp1 loss results in a significant reduction in Cx43 protein expression, leading to astrocyte clustering and disorganization. However, the causal role of Cx43 dysregulation in driving these phenotypes is not directly tested. To strengthen the mechanistic link between Drp1, Cx43, and astrocyte organization, the authors should perform rescue experiments by overexpressing Cx43 (full-length or specific isoforms like Cx43-20kDa) in Drp1-deficient astrocytes.

2. Cx43 Functional Analysis is Missing:

While the authors measure Cx43 protein levels and localization, they do not assess functional gap junction coupling. Functional tests, such as dye-transfer or dual-patch clamp experiments, could demonstrate whether Drp1 loss affects astrocytic communication via gap junctions.

Reviewer #3 (Comments to the Authors (Required)):

Summary

In this work, Rodriguez Salazar and colleagues studied the role of mitochondrial fission and fusion dynamics on the morphological maturation of astrocytes and their tiling in the cortex. To manipulate mitochondrial dynamics, they knocked down the expression of Drp1, a key regulator of mitochondrial fission, using techniques such as Piggybac methods for the sparse labeling and gene knock down, and astrocyte-specific conditional mutant mice. They performed extensive immunocytochemical analysis on astrocytes in primary cultures and cortical brain slices and studied how blocking Drp1 function can alter astrocyte morphology and mitochondrial structure. These studies found that the number of mitochondria increases with the maturation of astrocytes, with more mitochondria reaching the fine processes. They suggest that knockdown of Drp1 reduced astrocyte morphology complexity, induced aberrant clustering of astrocytes, and disrupted astrocyte tiling. All these phenotypes were highly variable, with more severe phenotypes seen in cultured astrocytes than in cell-type specific Drp1 conditional mutants. In addition, the authors suggest that the lack of Drp1 reduced the expression of gap-junction protein Cx40 in astrocyte and reactive astrogliosis. Although some of the provided data are exciting and offer deeper insights into the role of astrocyte mitochondria in cell morphogenesis, this study didn't provide strong experimental support to their claim in several places. One of the major caveats of this study is that throughout the manuscript, the authors correlate their findings to the impaired mitochondrial fission/fusion dynamics but never show that their genetic manipulation disrupted mitochondrial dynamics. Also, there is little or no mechanistic insight into how the breakdown of mitochondrial fission can lead to astrocyte tiling and gap-function coupling disruption. This work currently provides some interesting observations but requires significant work to fully solidify their observation and support their hypothesis.

Major concerns:

1. The number of mitochondria and the parameters quantified to describe mitochondrial structure are poorly defined. For example, (a) what is the unit for average mitochondrial size (Fig. 2F); in case it is in μm^3 (as in Fig. 1), then mitochondria are unbelievably large. (b) without proper segmentation of individual mitochondria, it is difficult to judge how the mitochondrial numbers were estimated in secondary and fine/terminal branches (Fig. 2C). (c) the authors claim that astrocytes acquire more mitochondria in fine processes as they mature. However, the data provided doesn't support this conclusion. When astrocytes mature, they become complex, and their fine branches increase by 3-4-fold (Fig. 2D); if carefully noted, the mitochondrial number increases by 3-4-fold. Once the number of mitochondria is normalized to the total cell morphology parameters (such as volume, etc.), it seems the number of mitochondria is linearly scaled as astrocytes matured (between 4-24 hours), and suggested there were no significant changes in the mitochondrial density.
2. Time-lapse imaging is essential to characterize mitochondrial dynamics (fission, fusion, and motility). In this study, the authors use genetic tools to systematically disrupt mitochondrial dynamics in astrocytes, e.g., shRNA against Drp1 (fission), Mfn1 (fusion), and Miro1 (transport and motility). However, the authors don't provide evidence that their genetic manipulation disrupted the mitochondrial dynamics in question. The main focus of this study is to investigate the role of mitochondrial fission in regulating astrocyte morphology, tiling, and gap junction coupling. However, no single dataset in the manuscript shows that astrocytes' mitochondrial fission/fusion dynamics are indeed affected when Drp1/Mfn1 proteins are knocked down.
3. If shRNA against Drp1 and Mfn1 worked as expected, the total number of mitochondria should also be affected. The expected results should be decreased mitochondria in the Drp1 KD and an increased number in the Mfn1 KD. If the numbers are unaffected (as in Suppl Fig. 2), then at least intrinsic mitochondrial properties (volume, length) should be affected while balancing fusion and fission kinetics. How do the authors explain that KD of both Drp1 and Mfn1 do not affect mitochondrial total content?
4. In addition, due to a lack of fission dynamics data, it becomes highly correlative and difficult to believe that the reduced

complexity in astrocyte morphology (Fig. 2 and 3) is due to impaired mitochondrial fission and not due to break down of other cellular/mitochondrial function on Drp1 knockdown. For example, in culture, when compared to in vivo (Fig. 2 vs. 3), the effect of Drp1 knockdown on astrocyte morphology is severe, implying that disruption of Drp1 might not affect fission but may be other critical cellular functions. Additionally, when the number of mitochondria is normalized to the number of fine branches in Drp1 control vs mutant astrocytes (Fig. 3D, E), possibly there might be either minor or no difference in the density of mitochondria, suggesting Drp1 knockdown didn't affect mitochondria number (as presented in Suppl 2). Also, from the images shown, it is tough to judge the localization of mitochondria to the distal/perisynaptic processes. Thus, at least from the data provided, there is little evidence that fission is required for the mitochondrial recruitment to the perisynaptic processes and the morphogenesis of fine and terminal processes.

5. The authors suggest that disruption of mitochondrial fission resulted in aberrant clustering and impaired tiling of astrocytes (Fig. 5 and 7). They measure the distance between Sox9+ nuclei in cell clusters where the phenotype is evident. But this doesn't indicate there is a tiling defect, and it is essential to show how the overall distribution of astrocytes was affected by performing tiling analysis on Drp1 KD astrocytes with the unaffected neighboring astrocytes. In addition, in some clusters, astrocyte nuclei are virtually touching each other (Fig. 5C); how could this be possible without changes in the astrocyte territory (Fig. 4B). It is difficult to visualize how the astrocyte clustering could occur, without terminal processes of neighboring astrocyte not significantly overlapping each other? Current data don't provide strong support that Drp1 KD leads to a disorganized astrocyte morphology in vivo.

6. Contrary to the PALE sparse labeling, in the Aldh111-Drp1-tdTomato cKO mice, the authors use tdTomato+ cells to estimate the tiling (Fig. 7) and tdTomato+ intensity to study coverage (Fig. 6F). It is known that not all astrocytes will be recombined in Aldh111-Drp1-tdTomato mice, and there will be animal-to-animal variability in the Cre-ER recombination efficiency. Therefore, the tiling analysis should be performed using the Sox9+ nuclei (as in Fig. 5). Based on PALE Drp1 shRNA transfections (Fig 5C), the authors should see similar multi-nuclei clusters in the Aldh111-Drp1-tdTomato cKO as well. If such clusters are absent, how could this difference between the two models (PALE vs cKO) be explained? Similarly, just looking at the intensity of tdTomato doesn't suffice to estimate cell coverage.

7. The authors must present absolute numbers for different cortical layers. The arbitrarily normalized data presented is confusing (Fig. 6D and E). For an unbiased estimation of the astrocyte density in controls and cKOs, Sox9 should be used; it is critical to use a marker independent of recombination efficiency variations (tdTomato) and astrocyte reactive state (GFAP). At present, it is difficult to get a clear picture of what is going on in the cortex of KO mice. The only conclusion one can confidently draw is that in the absence of Drp1, the astrocytes become reactive and upregulate GFAP.

8. Is there a difference between the expression of Cx43 in clustered astrocytes vs individual astrocytes in which Drp1 is knocked down (Fig. 5)? How does Cx43 staining look like in the regions where KD of Drp1 is stronger e.g., SVZ Fig. 5A. The finding that Drp1 disruption affects Cx40 expression is an interesting one, but still, it is not clear how Cx40 in Aldh111-Drp1 cKO mice only few Drp1- astrocytes show loss/reduced Cx40, and most of them have normal or even slightly increased levels. Also, the immunoblotting data showed a considerable change in the levels of Cx43 between controls and Aldh111-Drp1 cKO, but the immunohistochemistry data show modest changes in Cx40.

9. Do Aldh111-Drp1 cKO mice display any motor or other behavior phenotype compared to controls? It is highly recommended in the discussion to compare and contrast phenotypes of Drp1 null mutants and Aldh111-Drp1 cKO mice.

Minor concerns:

- A better analysis of fine astrocyte processes should be performed. The quantification presented as NIV (Fig. 4C), doesn't convincingly show the morphological changes in the fine astrocyte processes and its relation to the mitochondrial localization. It is known that astrocytes might not even reach the fine processes below 100-200nm in size.
- Representative images should also be shown at higher magnification (including Fig. 7D). It is crucial as the targets analyzed are expressed in subcellular compartments (mitochondria or gap junctions), and differences in observed expression seem cell-specific. By doing this, authors could rule out if there is any correlation between Drp1 expression levels (or Drp1 KD or KO efficiency) and decreased Cx43 expression
- It is uncommon to quantify just one experiment (Fig. 2). The authors should mention the number of biological and technical replicates used in their study. Ideally, statistical tests should be performed on a number of biological replicates and not technical replicates.
- Line 216 typo - distal.

We thank the reviewers for their careful review and constructive feedback on our manuscript, "Mitochondrial fission controls astrocyte morphogenesis and organization". We were pleased that all the reviewers found our study of high interest and novelty for the field, acknowledging the study's strengths in using *in vitro* and *in vivo* models, coupled with advanced imaging and extensive quantification to study Drp1's role in astrocyte maturation.

The reviewers also identified a few concerns and provided important suggestions to improve the study. The three major concerns of the reviewers centered on validation that our mitochondrial dynamics manipulation tools modify mitochondrial number and size, expansion of the characterization of Drp1 cKO astrocytes (cre recombination efficiency of the model, further reactive state targets, and PAP protein expression), and clarification of the Drp1-Cx43 relationship or discussion of alternative mechanisms by which mitochondrial fission may be regulating astrocyte maturation. In this revised version of the manuscript, we included new data addressing these major concerns and some of the minor concerns raised by all three reviewers.

In the revised manuscript, we included 6 new experiments, 13 new data analyses, and 23 new figure panels.

Below, we include the reviewer's comments and the experiments we included in the revised version of the manuscript, addressing each concern.

For ease of reviewing, the edited text in the manuscript and our responses below are highlighted in blue.

Reviewer #1:

In the current manuscript "Mitochondrial fission controls astrocyte morphogenesis and organization in the cortex", Salazar et al., describe a mechanism whereby during the process of astrocyte maturation and morphogenesis, mitochondria increase in number and decrease in size to facilitate organization into the tiny sub compartments of astrocytes (PAPs) which enwrap neuronal synapses. If mitochondrial fission is disrupted by the conditional deletion of Drp1, a protein critical for mitochondrial fission, PAP formation and cortical astrocyte organization is disrupted. Overall, this is a well-written and interesting manuscript that provides novel information related to astrocyte mitochondrial dynamics during early brain development. However, there are concerns that limit enthusiasm for the manuscript as written. These concerns are noted below.

We thank the reviewer for their thoughtful assessment of our manuscript. In this revised version of the manuscript, we included new data addressing the reviewer's major and minor concerns and provided further insight into the reactive state of Drp1 cKO astrocytes as well as the specificity of Cx43 protein regulation (versus other PAP proteins) when mitochondria fission is disrupted.

1. *The relationship between Drp1, mitochondrial fission and the specificity or link to Cnx43 needs further clarification. Cnx43 is a PAP protein, and PAPs are disrupted in the model. The authors need to clarify if the effects on Cnx43 are specific, or if PAP proteins in general are disrupted by performing immunohistochemistry and Western blotting for other PAP related proteins (ATPase, Glt1, Kir4.1, Glast, etc...). The findings would be interesting either way.*

We thank the reviewer for their important question. Following the reviewer's suggestion, we performed immunoblotting for three PAP proteins (Glut1, Kir4.1, and Glast) in immunopurified Drp1 cWT and cKO astrocytes from our Aldh1l1-Drp1 cKO model. We found that all three PAP proteins were significantly decreased in Drp1 cKO astrocytes compared to control, supporting our findings that distal astrocyte processes (which house PAP proteins) fail to properly form when mitochondrial fission is inhibited. **These new data are now included in Fig. 7L-Q (lines 400-411).** Regarding the specificity of Drp1 regulation of Cx43, these results indicate that Drp1 not only regulates Cx43 expression in astrocytes but also PAP proteins, thus **we have updated the figure and section title to be more generic to PAP protein dysregulation** in Drp1 cKO astrocytes. However, Cx43 is much more significantly decreased by Drp1 loss than all other PAP proteins tested, indicating Cx43 is more affected by dysfunctional mitochondrial dynamics. Lastly, in **Sup. Fig. 3F**, we show another gap junction protein expressed in astrocytes, Cx30, is not qualitatively decreased in our sparse Drp1 KD model (we do not see 'holes' of Cx30 expression within Drp1 KD astrocytes as we do with Cx43 (Fig. 5F)). These data show us that Drp1 does not modulate all astrocyte gap junctions similarly, and that Cx43 is significantly decreased and Cx30 is not. This is noted in the results section (line 310).

- 2. The manipulations are performed very early in astrocyte development, when proliferation, local expansion and migration are still occurring. Data indicates that KO of Drp1 leads to altered total number of astrocytes (as assessed by soma counts). However, manipulating Drp1, may also impact astrocyte migration to their proper location and thus nearest neighbor distance/astrocyte organization. The way the manuscript is written, this is directly linked to Cnx43 expression, which may or may not have anything to do with this observation. Could this be explored in a culture system (i.e. do astrocytes clump in a simplified culture system with no neurons around)? At a minimum, this possibility should be discussed.*

We agree with the reviewer that Drp1 loss has been shown to impact cell migration in other cell types and could likewise directly affect astrocyte migration to some degree during development. Further, we agree that our data do not demonstrate a causal relationship between Drp1-deficient astrocyte disorganization and Cx43 expression. We did not intend to claim that Cx43 disruption is *the* mechanism through which Drp1 may be regulating astrocyte organization but rather highlighted that both loss of Cx43 expression and astrocyte disorganization happen concurrently in two different models of Drp1 loss. We do suggest that Cx43 (which has cell migration functions in astrocytes, line 297 and 469) *could* be *one* of the ways in which Drp1 modulates astrocyte organization during development. Unfortunately, we could not study this question in a simpler culture system, as we have observed that astrocytes do not tile or properly express gap junction proteins in culture (either on their own or in co-culture with neurons). We have further clarified this throughout the text and **added discussion regarding Drp1's role in cell migration as a potential direct mechanism in our findings (lines 473-478).**

- 3. Related to the above comment, it would be useful to manipulated Drp1 a later in development when proliferation and migration are complete, and morphogenesis occurs*

We agree with the reviewer that early manipulation of Drp1 could affect astrocyte migration and proliferation. Indeed, studying Drp1 loss in astrocytes at later developmental timepoints is the immediate next question we have for subsequent publications, but for this manuscript we are

particularly interested the early developmental role of Drp1 in astrocyte maturation. Regardless, in **Sup. Fig. 3A-C**, we performed extensive time-course investigation of proliferation dynamics in Drp1 WT vs. KD astrocytes in our sparse model and found no significant differences. This data suggests that early postnatal loss of Drp1 does not affect astrocyte proliferation. Lastly, reducing Drp1 in astrocytes has proven to be quite difficult in our hands even in our early postnatal manipulations, and therefore this question would require longer than the time we have for revisions to address. For example, the early postnatal deletion of Drp1 in astrocytes took us two years to optimize and validate using two different astrocyte-specific Cre driver lines. Dosing sufficient tamoxifen to induce Drp1 loss in early postnatal timepoints (P0-P3) took several months of trial and error. We have added the potential of **Drp1's role in cell migration as a consideration in our study in the discussion (lines 473-478)**.

4. *Given that GFAP and vimentin are directly tied to astrocyte morphology, additional astrocyte reactive targets need to be validated to demonstrate these astrocytes are indeed reactive.*

We appreciate the reviewer's comment. To further characterize whether Drp1 cKO astrocytes are indeed reactive, we performed qPCR on isolated Drp1 cWT and cKO astrocytes for 4 additional markers of astrocyte reactivity: LCN2, Cxcl10, Serpina3n, and Cp (Liddel et al., 2017). These data show a significant increase in LCN2 and Cxcl10, a trending increase in Serpina3n, and no significant difference in Cp in Drp1 cKO astrocytes compared to control. This data further support that Drp1 cKO astrocytes upregulated multiple markers of astrocyte reactivity (5 out of 6 tested, lines 359-362). **These new data are now included in Fig. 6K-L (LCN2 and Cxcl10) and Sup. Fig. 4H-I (Serpina3n and Cp)**.

5. *The analysis done in figure 1 c-d, should be repeated in at least one of the in vivo Drp1 manipulations to show that Drp1 KO results in larger, fewer, less distal mitochondria when fission is disrupted, particularly because total mito # was not observed to change in Drp1 shRNA experiments.*

We appreciate the reviewer's comment. The original data the reviewer mentions (now **Sup. Fig. 2E-H**) was not a quantification of total mitochondrial number but rather total mitochondrial content, which is the total volume of mitochondria within a cell normalized to total cell volume. The reviewer is correct, shDrp1 did not significantly change total mitochondrial content within astrocytes (neither did any of our 3 mitochondrial dynamics manipulations). This data indicates that astrocyte morphogenesis (distal process volume for shDrp1 or territory size for shMfn1) is inhibited in proportion to changing mitochondrial dynamics (size and number) to normalize mitochondrial content/occupancy. We apologize for the confusion on the communication of our methods and have now further clarified this in the text (lines 251-254).

In addition, following the reviewer's suggestion, we performed the mitochondrial analyses done in Figure 1 (mitochondrial number and size) using Imaris for all three mitochondrial dynamics manipulations (Drp1, Mfn1, and Miro1) in our sparse knockdown model. This analysis demonstrated that shDrp1 results in a significant increase in mitochondrial size and decrease in mitochondrial number throughout astrocytes as we would expect when inhibiting mitochondrial fission. **This new data is now included in Sup. Fig. 2A-D (lines 223-226)**. Further, in **Fig. 4D** we showed that shDrp1 astrocytes have a significant decrease of distal mitochondria specifically. Lastly, we have also added a representative image to show that shDrp1 astrocytes in culture likewise present with hyperfused/elongated mitochondria, **now in Fig. 3C (lines 185-186)**.

6. *There is little consideration given to how and if Drp1 manipulation, particularly globally is likely to impact the astrocyte cellular metabolism, oxidative phosphorylation, glutamine/glutamate metabolism, ATP production, etc. This may be directly related to astrocyte reactivity, yet this is not discussed either.*

We thank the reviewer for their important point and agree that mitochondrial dynamics not only control the localization of mitochondria within cells, but also mitochondrial function itself. We have **added discussion** on this point throughout the manuscript, primarily in the discussion (lines 496-508).

Minor:

7. *Please include citation - introduction, lines 55-56, '...ramified processes make up more than one-half the brain parenchyma by infiltrating the neuropil'*

Thank you for catching this missing citation, we have updated the statement and added the citation.

Reviewer #2:

The manuscript of Rodriguez Salazar et al., presents an important and novel exploration of how mitochondrial fission, regulated by Dynamin-related protein 1 (Drp1), impacts astrocyte morphogenesis and cortical organization during postnatal brain development. The authors convincingly demonstrate that Drp1 loss disrupts mitochondrial dynamics, reduces astrocyte complexity, induces clustering, and dysregulates the gap junction protein Connexin 43 (Cx43). The study is well-executed, employing both in vivo and in vitro models, as well as advanced imaging and quantification tools. While the study makes significant contributions to the understanding of mitochondrial dynamics in astrocytes, a key mechanistic gap limits the impact of the findings. Specifically, the role of Cx43 in rescuing Drp1-related astrocyte phenotypes remains unexplored, leaving an important question unanswered.

We thank Reviewer 2 for their helpful comments and for pointing out exciting questions for follow-up studies.

Major Concerns:

1. *Lack of Functional Rescue Experiments with Cx43: The authors convincingly show that Drp1 loss results in a significant reduction in Cx43 protein expression, leading to astrocyte clustering and disorganization. However, the causal role of Cx43 dysregulation in driving these phenotypes is not directly tested. To strengthen the mechanistic link between Drp1, Cx43, and astrocyte organization, the authors should perform rescue experiments by overexpressing Cx43 (full-length or specific isoforms like Cx43-20kDa) in Drp1-deficient astrocytes.*

We thank the reviewer for this important comment and agree that rescue experiments introducing Cx43 into shDrp1 astrocytes are necessary to establish a causal role of Drp1-induced Cx43 dysregulation and astrocyte disorganization. First as a point of clarification, we do not claim that

our data demonstrates a causal relationship between Drp1, Cx43, and astrocyte disorganization, and Cx43 expression. Instead, we demonstrate both the disruption of astrocyte organization and the dysregulation of Cx43 expression are *simultaneously* observed in Drp1-deficient astrocytes and suggest Cx43 disruption could be one of the mechanisms by which astrocyte disorganization is observed. Indeed, it is likely that disrupting mitochondrial fission could have multiple converging effects on astrocyte biology during development that would lead to dysfunctional astrocyte migration and organization. Namely, Drp1 itself has been shown to have direct roles in regulating cell migration (Kim et al., 2015), thus Drp1 loss in our model could directly result in astrocyte disorganization independently or synergistically with Cx43 loss. Therefore, we do not believe that Cx43 is the only mechanism at play in our work, and we have further **clarified this throughout the manuscript, particularly in the discussion (lines 473-478)**.

Nonetheless, we robustly attempted to perform Cx43 rescue experiments as suggested by the reviewer. We considered 3 different approaches and thoroughly pursued one approach for the past 6 months without success:

1. *In vitro* Cx43 rescue experiments in shDrp1 astrocytes in culture: We could not perform these rescue experiments *in vitro* because astrocytes do not disperse into organized domains or tile in culture and therefore would have no output metric to assess the relationship between Drp1, Cx43, and astrocyte disorganization.
2. Viral overexpression of Cx43 in Drp1 cKO mice: Viruses only begin to express at P7 in the cortex, and do not fully express until P14, when astrocytes have finished their migration and are almost completely occupying their final territories (Clavreul et al., 2019). Therefore, this approach would miss the developmental timeframe in which we would need to restore Cx43 expression (before astrocytes form an organized network).
3. Sparse overexpression of Cx43 in shDrp1 astrocytes in PALE mice: This approach required us to inject and electroporate 3 separate plasmids (two plasmids for piggyBac shDrp1-mCherry expression, and 1 Cx43 overexpression plasmid (Baldwin et al., 2021)) into the ventricles of P0 mice. Further, we had to rely on the probabilistic nature of these 3 plasmids electroporating into single astrocytes to rescue Cx43 in shDrp1 astrocytes. The same cell would have to be electroporated with 3 separate plasmids, one of which (the Cx43 overexpression plasmid) has very low expression efficiency *in vivo*.

Based on the limitations above, method # 3 is the approach we thought was the most feasible out of the three possible options, and below is a summary of our attempts. As the reviewer will see, we were technically limited by the very low number of cells expressing all 3 plasmids, despite trying different iterations of plasmid ratios electroporated, and thus were unfortunately unable to collect enough data to analyze and address this question.

	Experiment	Animals	Plasmid ratio	Rescue cells
	1	9	PBase: 0.8ug, shRNA: 0.7ug, Cx43TY: 1ug	0
	2	8	PBase: 0.9ug, shRNA: 0.8ug, Cx43TY: 1ug	0
	3	7	PBase: 0.9ug, shRNA: 0.7ug, Cx43TY: 1ug	0
	4	5	PBase: 0.9ug, shRNA: 0.7ug, Cx43TY: 1ug	0
	5	6	PBase: 0.9ug, shRNA: 0.7ug, Cx43TY: 1ug	1
	6	8	PBase: 0.9ug, shRNA: 0.7ug, Cx43TY: 1ug	2
	7	7	PBase: 0.9ug, shRNA: 0.7ug, Cx43TY: 1ug	0 litter died
	8	12	PBase: 0.9ug, shRNA: 0.7ug, Cx43TY: 1.4ug	TBD/ongoing
	9	12	PBase: 0.9ug, shRNA: 0.7ug, Cx43TY: 1.4ug	0 litter died

Total	9 Experiments	74 Animals	4 plasmid ratios	3 rescue cells
-------	---------------	------------	------------------	----------------

While we were technically unable to investigate the direct link between Cx43 and astrocyte disorganization with rescue experiments, we tried an alternative approach to shed some light into this relationship. Since Cx43 is a perisynaptic astrocyte process (PAP) protein, we probed the specificity of the link between Cx43, Drp1, and astrocyte organization by assessing levels of other astrocyte PAP proteins in Drp1-deficient astrocytes. To do so, we performed immunoblotting for three PAP proteins (Glut1, Kir4.1, and Glast) in immunopurified Drp1 cWT and cKO astrocytes from our Aldh111-Drp1 cKO model. We found that all three PAP proteins were significantly decreased in Drp1 cKO astrocytes compared to control, supporting our findings that distal astrocyte processes (which house PAP proteins) fail to properly form when mitochondrial fission is inhibited. **These new data are now included in Fig. 7L-Q (lines 400-413).** These results also indicate that Drp1 not only regulates Cx43 expression in astrocytes but also PAP proteins, thus **we have updated the figure and section title to be more generic to PAP protein dysregulation** in Drp1 cKO astrocytes. Nevertheless, Cx43 is much more significantly decreased by Drp1 loss than all other PAP proteins tested (Fig. 7J and Fig. 7L-Q). These findings suggest that mitochondrial fission is required for the proper expression of multiple PAP proteins in developing astrocytes (not only Cx43), but Cx43 protein expression is more vulnerable to aberrant mitochondrial fission. Thus, it is still plausible that disrupted Cx43 with its known roles in cell migration, and in concert with decreases in other PAP proteins, contributes to the disorganization of astrocyte domains in Drp1 cKO mice.

2. Cx43 Functional Analysis is Missing: While the authors measure Cx43 protein levels and localization, they do not assess functional gap junction coupling. Functional tests,

such as dye-transfer or dual-patch clamp experiments, could demonstrate whether Drp1 loss affects astrocytic communication via gap junctions.

We agree with the reviewer that functional astrocyte-astrocyte coupling tests would be interesting, however, we unfortunately do not have the technical expertise in our lab to currently perform these complex studies within the revision timeframe. We have **added discussion** on the limitation of our studies to Cx43 protein disruption but not function, and the need for functional studies in the future to address astrocyte coupling function in Drp1 KO conditions (**lines 479-481**).

Reviewer #3:

In this work, Rodriguez Salazar and colleagues studied the role of mitochondrial fission and fusion dynamics on the morphological maturation of astrocytes and their tiling in the cortex. To manipulate mitochondrial dynamics, they knocked down the expression of Drp1, a key regulator of mitochondrial fission, using techniques such as Piggybac methods for the sparse labeling and gene knock down, and astrocyte-specific conditional mutant mice. They performed extensive immunocytochemical analysis on astrocytes in primary cultures and cortical brain slices and studied how blocking Drp1 function can alter astrocyte morphology and mitochondrial structure. These studies found that the number of mitochondria increases with the maturation of astrocytes, with more mitochondria reaching the fine processes. They suggest that knockdown of Drp1 reduced astrocyte morphology complexity, induced aberrant clustering of astrocytes, and disrupted astrocyte tiling. All these phenotypes were highly variable, with more severe phenotypes seen in cultured astrocytes than in cell-type specific Drp1 conditional mutants. In addition, the authors suggest that the lack of Drp1 reduced the expression of gap-junction protein Cx40 in astrocyte and reactive astrogliosis. Although some of the provided data are exciting and offer deeper insights into the role of astrocyte mitochondria in cell morphogenesis, this study didn't provide strong experimental support to their claim in several places. One of the major caveats of this study is that throughout the manuscript, the authors correlate their findings to the impaired mitochondrial fission/fusion dynamics but never show that their genetic manipulation disrupted mitochondrial dynamics. Also, there is little or no mechanistic insight into how the breakdown of mitochondrial fission can lead to astrocyte tiling and gap-function coupling disruption. This work currently provides some interesting observations but requires significant work to fully solidify their observation and support their hypothesis.

We thank Reviewer 3 for their detailed comments and suggestions to improve our manuscript. In this revised manuscript, we addressed each of the major and minor comments below.

Major concerns:

1. *The number of mitochondria and the parameters quantified to describe mitochondrial structure are poorly defined. For example, (a) what is the unit for average mitochondrial size (Fig. 2F); in case it is in μm^3 (as in Fig.1), then mitochondria are unbelievably large. (b) without proper segmentation of individual mitochondria, it is difficult to judge how the mitochondrial numbers were estimated in secondary and fine/terminal branches (Fig. 2C). (c) the authors claim that astrocytes acquire more mitochondria in fine processes as they mature. However, the data provided doesn't support this conclusion. When astrocytes*

mature, they become complex, and their fine branches increase by 3-4-fold (Fig. 2D); if carefully noted, the mitochondrial number increases by 3-4-fold. Once the number of mitochondria is normalized to the total cell morphology parameters (such as volume, etc.), it seems the number of mitochondria is linearly scaled as astrocytes matured (between 4-24 hours), and suggested there were no significant changes in the mitochondrial density.

(a) We thank the reviewer for bringing this error to our attention. The reviewer is correct, the scaling of our raw images going into Seg_Astro was incorrect (wrongly scaled in 150 pixels per inch, when they should have been 6.18 pixels per μm) which incorrectly scaled up our mitochondrial area measurement. We have corrected this error by converting the Seg_Astro area output to pixels/ μm and have updated the data accordingly. Since this was a scaling issue, the trends and take-aways of the data have not changed, but the average areas are now accurately representing area (μm^2) of mitochondria and **updated in Figure 2F**.

(b) We agree with the reviewer that mitochondrial overlap throughout processes can be a limitation of quantification of mitochondrial numbers. However, this is a limitation of the study of most organelles, and any bias in the limitation of imaging and mitochondrial segmentation would apply equally to all our conditions. Furthermore, this is the reason we excluded primary branches from our *in vitro* analyses, as perinuclear mitochondria form an interconnected network that is impossible to quantify. We used single optical sections of images when quantifying mitochondria *in vitro*, which is the best strategy available. As mitochondria become more distal, so do their separation, thus allowing us to more accurately quantify mitochondria as described in our methods.

(c) We do not claim fine processes contain more mitochondria than other process types, but rather that both fine branches and mitochondria within fine branches demonstrate the greatest increase in number compared to other types of branches. **We have further clarified this in the text (lines 163-167)**

2. *Time-lapse imaging is essential to characterize mitochondrial dynamics (fission, fusion, and motility). In this study, the authors use genetic tools to systematically disrupt mitochondrial dynamics in astrocytes, e.g., shRNA against Drp1 (fission), Mfn1 (fusion), and Miro1 (transport and motility). However, the authors don't provide evidence that their genetic manipulation disrupted the mitochondrial dynamics in question. The main focus of this study is to investigate the role of mitochondrial fission in regulating astrocyte morphology, tiling, and gap junction coupling. However, no single dataset in the manuscript shows that astrocytes' mitochondrial fission/fusion dynamics are indeed affected when Drp1/Mfn1 proteins are knocked down.*

We appreciate the reviewer's comment and agree that time-lapse imaging would have been helpful to our studies. However, since we performed our mitochondrial dynamics studies *in vivo*, time-lapse imaging is not possible to do in our lab *in vivo*. To address the reviewer's comment, we have now added mitochondrial number and size quantification from our *in vivo* PALE experiments for the three shRNA mitochondrial dynamics manipulations, and these new data demonstrate that the mitochondrial network within astrocytes is robustly remodeled differentially by each regulator of mitochondrial dynamics. For example, shDrp1 astrocytes have significantly larger and fewer mitochondria while shMfn1 astrocytes have significant smaller and more mitochondria compared to control. **These new data are now included in Sup. Fig. 2 A-D, (lines 223-227, 240-243)**. Furthermore, we have added clearer representative images of the profound

mitochondrial hyperfusion and elongation we consistently observe using the same shRNA against Drp1 in our *in vitro* studies. **This new data is now included in Fig. 3C (line 185-186).**

3. *If shRNA against Drp1 and Mfn1 worked as expected, the total number of mitochondria should also be affected. The expected results should be decreased mitochondria in the Drp1 KD and an increased number in the Mfn1 KD. If the numbers are unaffected (as in Suppl Fig. 2), then at least intrinsic mitochondrial properties (volume, length) should be affected while balancing fusion and fission kinetics. How do the authors explain that KD of both Drp1 and Mfn1 do not affect mitochondrial total content?*

We appreciate the reviewer's comment. The original data the reviewer mentions (now **Sup. Fig. 2E-H**) was not a quantification of total mitochondrial number but rather total mitochondrial content, which is the total volume of mitochondria within a cell normalized to total cell volume. The reviewer is correct, shDrp1 did not significantly change total mitochondrial content within astrocytes (neither did any of our 3 mitochondrial dynamics manipulations). This data indicates that astrocyte morphogenesis (distal process volume for shDrp1 or territory size for shMfn1) is inhibited in proportion to changing mitochondrial dynamics (size and number) to normalize mitochondrial occupancy. We apologize for the confusion on the communication of our methods and have now further **clarified this in the text (lines 251-254).**

In addition, following the reviewer's #2 comment above, we performed the mitochondrial analyses done in Figure 1 (mitochondrial number and size) using Imaris for all three mitochondrial dynamics manipulations (Drp1, Mfn1, and Miro1) in our sparse knockdown model. This analysis demonstrated that shDrp1 results in a significant increase in mitochondrial size and decrease in mitochondrial number throughout astrocytes as we would expect when inhibiting mitochondrial fission, and the inverse trend when inhibiting mitochondrial fusion. **This new data is now included in Sup. Fig. 2A-D (lines 223-226).**

4. *In addition, due to a lack of fission dynamics data, it becomes highly correlative and difficult to believe that the reduced complexity in astrocyte morphology (Fig. 2 and 3) is due to impaired mitochondrial fission and not due to break down of other cellular/mitochondrial function on Drp1 knockdown. For example, in culture, when compared to in vivo (Fig. 2 vs. 3), the effect of Drp1 knockdown on astrocyte morphology is severe, implying that disruption of Drp1 might not affect fission but may be other critical cellular functions. Additionally, when the number of mitochondria is normalized to the number of fine branches in Drp1 control vs mutant astrocytes (Fig. 3D, E), possibly there might be either minor or no difference in the density of mitochondria, suggesting Drp1 knockdown didn't affect mitochondria number (as presented in Suppl 2). Also, from the images shown, it is tough to judge the localization of mitochondria to the distal/perisynaptic processes. Thus, at least from the data provided, there is little evidence that fission is required for the mitochondrial recruitment to the perisynaptic processes and the morphogenesis of fine and terminal processes.*

We appreciate the reviewer's comment. We address the reviewer's concerns point by point below:

1. *"Due to a lack of fission dynamics data, it becomes highly correlative and difficult to believe that the reduced complexity in astrocyte morphology": We have now provided new data that indicate defective mitochondrial fission in shDrp1 astrocytes (reviewer comment 2 and 3*

above, Sup. Fig. 2A-D), providing more clarity on the link between mitochondrial fission and astrocyte morphogenesis.

2. *"In culture, when compared to in vivo, the effect of Drp1 knockdown on astrocyte morphology is severe"*: It is common in our lab to see astrocyte morphology phenotypes much more strongly *in vitro* than when we perform the same manipulation *in vivo*, likely due to the increased complexity of an *in vivo* model system that could trigger compensatory mechanisms, especially when studying such a fundamental cellular function such as mitochondrial dynamics.
3. *"When the number of mitochondria is normalized to the number of fine branches in Drp1 control vs mutant astrocytes, possibly there might be either minor or no difference in the density of mitochondria, suggesting Drp1 knockdown didn't affect mitochondria number"*: 'Mitochondrial number' is a raw quantification of the number of mitochondria within an astrocyte branch type (independent of normalization to cell volume), while 'mitochondrial density' is a normalized measure of mitochondrial number normalized to cell volume (or branch area). We believe there is a miscommunication with this terminology and have **updated the text throughout to clarify these separate metrics**, as mitochondrial number and mitochondrial density cannot be used interchangeably. With the development of Seg_Astro, we robustly show that shDrp1 astrocytes do indeed have decreased mitochondrial numbers across all process types quantified (secondary, fine, and terminal, **Fig. 3E**). The reviewer is correct, that fine branch number also decreases in shDrp1 astrocytes to a proportional degree to mitochondrial number loss in fine branches. This parallel decrease of both mitochondrial and branch numbers would result in a near equivalent mitochondrial *density* as the reviewer rightly points out, but it does not change the fact that mitochondrial *numbers* are decreased in shDrp1 astrocytes. Lastly, the parallel decrease of both fine branch and mitochondrial number in shDrp1 astrocytes further supports the premise of our manuscript's findings- that astrocyte morphology is inhibited in concert with mitochondrial dynamics' defects. **This has been clarified throughout the text and in lines 191-195.**
4. *"There is little evidence that fission is required for the mitochondrial recruitment to the perisynaptic processes and the morphogenesis of fine and terminal processes"*: We provide several lines of evidence, *in vitro* and *in vivo*, that support the finding that mitochondrial fission controls mitochondrial recruitment to distal processes and the morphogenesis of fine/distal processes specifically as outlined below:
 1. Mitochondria increase in number and decrease in size during astrocyte morphogenesis (Fig. 1F-G) while robustly occupying nascent astrocyte fine/distal branches (Fig. 2D-F), which house exclusively small/fragmented mitochondria throughout morphogenesis (Fig. 2F).
 2. Astrocyte fine process morphogenesis is uniquely affected by Drp1 loss *in vitro*. For example, while astrocyte secondary branches also display a significant decrease in mitochondrial number in shDrp1 conditions, the number of secondary branches (and primary branches) is unaffected by Drp1 loss (Fig. 3E-F). Conversely, the robust decrease in mitochondrial numbers in astrocyte fine branches is proportional to fine branch loss in shDrp1 conditions. These data demonstrate that Drp1-induced mitochondrial fission is uniquely required for fine and distal astrocyte process formation. Further, Drp1 does not affect higher order process formation, despite decreases in mitochondria numbers in those larger astrocyte processes. **We have further clarified this in the text (lines 191-198)**

3. Astrocyte distal process morphogenesis is uniquely affected by Drp1 loss *in vivo*. Drp1 loss in astrocytes *in vivo* results in decreased distal mitochondrial volume, accompanied by decreased volume of distal neuropil infiltrating astrocyte processes (Fig. 4C-D). Again here, this effect is specific to distal astrocyte processes as Drp1 loss does not affect astrocyte territory size (Fig. 4B), which is established by higher order astrocyte processes (primary and secondary branches).

Lastly, we agree with the reviewer that Drp1 loss can parallelly likely affect other cellular functions (such as cell metabolism or migration) that could contribute to the phenotypes we see. We have now **added discussion** regarding mitochondrial fission's roles in cell migration and cell metabolism as potential parallel contributing mechanisms in our study (**lines 473-479, 496-508**).

5. *The authors suggest that disruption of mitochondrial fission resulted in aberrant clustering and impaired tiling of astrocytes (Fig, 5 and 7). They measure the distance between Sox9+ nuclei in cell clusters where the phenotype is evident. But this doesn't indicate there is a tiling defect, and it is essential to show how the overall distribution of astrocytes was affected by performing tiling analysis on Drp1 KD astrocytes with the unaffected neighboring astrocytes. In addition, in some clusters, astrocyte nuclei are virtually touching each other (Fig. 5C); how could this be possible without changes in the astrocyte territory (Fig. 4B). It is difficult to visualize how the astrocyte clustering could occur, without terminal processes of neighboring astrocyte not significantly overlapping each other? Current data don't provide strong support that Drp1 KD leads to a disorganized astrocyte morphology in vivo.*

We thank the reviewer for their detailed comment. We address the reviewer's concerns point by point below:

1. *"But this doesn't indicate there is a tiling defect, and it is essential to show how the overall distribution of astrocytes was affected by performing tiling analysis on Drp1 KD astrocytes with the unaffected neighboring astrocytes."* Our data demonstrates that Drp1 loss disrupts astrocyte organization amongst Drp1 KD astrocytes *only*, as we would not expect organization of distances between wild-type astrocytes to be affected since PALE is a sparse Drp1 KD model (Fig. 3C-E). However, we do measure distances between all astrocyte neighbors in our Drp1 cKO model for a more global approach to investigate the role of Drp1 in astrocyte organization, and in this model also find disruptions to astrocyte organization in Drp1 cKO mice (Fig. 7A-C).
2. *"In addition, in some clusters, astrocyte nuclei are virtually touching each other; how could this be possible without changes in the astrocyte territory. It is difficult to visualize how the astrocyte clustering could occur, without terminal processes of neighboring astrocyte not significantly overlapping each other?"* We performed astrocyte morphology analyses on Drp1 KD astrocytes that were on the periphery of clusters due to the limitations the reviewer highlights and **have further clarified this in the methods (lines 952-954)**. The reviewer is correct, and it is highly likely that astrocytes within the shDrp1 clusters greatly overlap in their territories. Hence our finding that Drp1 and mitochondrial fission regulate astrocyte organization in the cortex and likely tiling of astrocyte domains.
3. *"Current data don't provide strong support that Drp1 KD leads to a disorganized astrocyte morphology in vivo."* We apologize for the confusion, we do not claim a disorganization of astrocyte *morphology* but a disruption in the *organization* of astrocytes in relationship to other astrocytes, as noted by their distances to neighboring astrocytes.

6. *Contrary to the PALE sparse labeling, in the Aldh1l1-Drp1-tdTomato cKO mice, the authors use tdTomato+ cells to estimate the tiling (Fig. 7) and tdTomato+ intensity to study coverage (Fig. 6F). It is known that the not all astrocytes will be recombined in Aldh1l1-Drp1-tdTomato mice, and there will be animal-to-animal variability in the Cre-ER recombination efficiency. Therefore, the tiling analysis should be performed using the Sox9+ nuclei (as in Fig. 5). Based on PALE Drp1 shRNA transfections (Fig 5C), the authors should see similar multi-nuclei clusters in the Aldh1l1-Drp1-tdTomato cKO as well. If such clusters are absent, how could this difference between the two models (PALE vs cKO) be explained? Similarly, just looking at the intensity of tdTomato doesn't suffice to estimate cell coverage.*

We thank the reviewer for their important comments and address their concerns point by point below:

1. *“It is known that the not all astrocytes will be recombined in Aldh1l1-Drp1-tdTomato mice, and there will be animal-to-animal variability in the Cre-ER recombination efficiency. Therefore, the tiling analysis should be performed using the Sox9+ nuclei”* To address concerns that varying efficiency in recombination between our cWT and cKO conditions may be confounding our astrocyte domain organization findings, we have performed a recombination efficiency study quantifying the number of Sox9+ and tdTomato+ double positive cells out of total Sox9+ cells in our Drp1 cWT and cKO animals to measure recombination efficiency in our model. This data demonstrates both Drp1 cWT and cKO conditions have an equivalent recombination efficiency in astrocytes (~ %80) and are not significantly different from each other. Therefore, non-recombined astrocytes make up an equivalent and small proportion of the astrocytes in Drp1 cWT and cKO conditions, making tdTomato an appropriate proxy for our measures of astrocyte organization. **This new data is now included in Sup. Fig. 4E-F (line 327-331).**
2. *“Based on PALE Drp1 shRNA transfections (Fig 5C), the authors should see similar multi-nuclei clusters in the Aldh1l1-Drp1-tdTomato cKO as well. If such clusters are absent, how could this difference between the two models (PALE vs cKO) be explained?”* Sparse knock down models are often different than conditional knock out of a protein in all cells. In PALE, we only knock down Drp1 in a few cells that end up clustering, while all other cells remain wild type. It is possible that the wildtype cells “reject” Drp1 KD astrocytes from their network and thus “push” them away from other wild-type cells into a cluster of KD astrocytes. This is not the case when Drp1 is knocked out from most astrocytes in the cKO model and thus, the disorganization is observed throughout the cortex. Nonetheless, we do see a significant decrease in the nearest neighbor distance in our Drp1 cKO model as well, accompanied by a significantly more varied distribution of distances between multiple neighbors (Fig.7 B-C). This data indicate that Drp1 cKO astrocytes both cluster together and are further apart, meaning disorganized, compared to Drp1 cWT astrocytes.
3. *“Similarly, just looking at the intensity of tdTomato doesn't suffice to estimate cell coverage.”* The use of fluorescent cell reporters is a well-established proxy for cell coverage, and the best proxy available to estimate astrocyte coverage across the cortex (Stogsdill et al., 2017).
7. *The authors must present absolute numbers for different cortical layers. The arbitrarily normalized data presented is confusing (Fig. 6D and E). For an unbiased estimation of the astrocyte density in controls and cKOs, Sox9 should be used; it is critical to use a marker independent of recombination efficiency variations (tdTomato) and astrocyte*

reactive state (GFAP). At present, it is difficult to get a clear picture of what is going on in the cortex of KO mice. The only conclusion one can confidently draw is that in the absence of Drp1, the astrocytes become reactive and upregulate GFAP.

We apologize for the lack of clarity in our cell density quantification method. Raw cell numbers per cortical layer are an inaccurate representation of cell density, as cortical layer area size where cells are counted vary and thus must be taken into account to report density. Layer 1, for example, is much smaller in area compared to layer 4 of the cortex, therefore their area size must be used to normalize the raw cell numbers when reporting cell density.

Following the reviewer's suggestion, we have now also quantified the total number of Sox9+ cells in both cWT and cKO mice across the cortex and recapitulated our density findings using tdTomato (Fig. 6E). **This new data is now included in Sup. Fig. 4G.** Both pieces of data indicate a slight but significant decrease in astrocyte density in Drp1 cKO mice (**lines 339-340**).

8. Is there a difference between the expression of Cx43 in clustered astrocytes vs individual astrocytes in which Drp1 is knocked down (Fig. 5)? How does Cx43 staining look like in the regions where KD of Drp1 is stronger e.g., SVZ Fig. 5A. The finding that Drp1 disruption effects Cx40 expression is an interesting one, but still, it is not clear how Cx40 in Aldh111-Drp1 cKO mice only few Drp1- astrocytes show loss/reduced Cx40, and most of them have normal or even slightly increased levels. Also, the immunoblotting data showed a considerable change in the levels of Cx43 between controls and Aldh111-Drp1 cKO, but the immunohistochemistry data show modest changes in Cx40.

We thank the reviewer for their insightful questions. We have quantified Cx43 in single Drp1 KD astrocytes and found a similar reduction of Cx43 in single shDrp1 astrocytes as we did in shDrp1 astrocyte clusters. **This new data is now included in Fig. 5F-G (lines 304-309).** There is no SVZ labeling at P21, when we are doing our Cx43 quantifications. SVZ labeling is present at earlier timepoints when radial glial cells are still differentiating and extending their cortical processes. By P21, radial glial cells are gone and there is little to no SVZ labeling, therefore we are unable to measure Cx43 in this region at P21.

We agree with the reviewer and were likewise intrigued by the difference between Cx43 expression in Drp1 cKO astrocytes by IHC vs. WB. However, the data clearly show that Cx43 expression is robustly decreased in Drp1 KD astrocyte clusters (Fig. 5F and G) and in isolated Drp1 cKO astrocytes (Fig. 7I), but heterogeneously dysregulated by IHC in Drp1 cKO mice (Fig. 7D-G). One plausible explanation is that Drp1 KD clusters are "excluded" from the otherwise Drp1 WT surrounding astrocyte network and therefore more easily lose their Cx43 protein. Inversely, in the cKO model where most astrocytes are Drp1-deficient, it is possible that compensatory mechanisms arise to attempt to produce new Cx43 protein given the fundamental importance of Cx43 for astrocyte function when the whole astrocyte network is compromised. Given that Cx43 is a rapidly recycled protein (Beardslee et al., 1998; Laird et al., 1991), this could explain the heterogeneous expression via IHC. Once we isolate astrocytes to perform WB and separate them from each other and their network environment, Drp1 cKO astrocyte Cx43 protein may be rapidly degraded compared to Drp1 cWT astrocytes as observed via WB.

9. *Do Aldh111-Drp1 cKO mice display any motor or other behavior phenotype compared to controls? It is highly recommended in the discussion to compare and contrast phenotypes of Drp1 null mutants and Aldh111-Drp1 cKO mice.*

We appreciate the reviewer's point and agree that these behavioral studies would have been an exciting next step in our work. However, in this study we did not perform these experiments but have **added discussion on comparisons between Drp1 null mice and our model (line 509-514)**.

Minor concerns:

- *A better analysis of fine astrocyte processes should be performed. The quantification presented as NIV (Fig. 4C), doesn't convincingly show the morphological changes in the fine astrocyte processes and its relation to the mitochondrial localization. It is known that astrocytes might not even reach the fine processes below 100-200nm in size.*

We appreciate the reviewer's comment. However, astrocyte fine processes *in vivo* are so fine and densely complex that even super resolution microscopy cannot resolve single processes accurately. The only truly certain measure of astrocyte fine processes would be EM, which we do not have the technical ability or revision timeline to accomplish. Neuropil infiltration volume has been the standard in the field as an estimate for the study of distal astrocyte process morphogenesis (Stogsdill et al., 2017).

- *Representative images should also be shown at higher magnification (including Fig. 7D). It is crucial as the targets analyzed are expressed in subcellular compartments (mitochondria or gap junctions), and differences in observed expression seem cell-specific. By doing this, authors could rule out if there is any correlation between Drp1 expression levels (or Drp1 KD or KO efficiency) and decreased Cx43 expression*

We have **added higher magnification images for Fig. 7D** and previously reported higher magnification images for all other conditional deletion data (Fig. 6D, Sup. Fig. 4A).

- *It is uncommon to quantify just one experiment (Fig. 2). The authors should mention the number of biological and technical replicates used in their study. Ideally, statistical tests should be performed on a number of biological replicates and not technical replicates.*

We appreciate the reviewer's comment. We do report all of our biological and technical replicates in our figure legends, and all of our experiments are at least a biological N of 3. We agree that this Fig. 2 is unique because it is a biological N of 1- however, this experiment is a thorough time-course assessment of wild-type (non-manipulated) astrocytes in culture. This experiment uses cells from 10 different male and female wildtype rat pups combined, a technical n of >10 cells from 3 different cell culture wells per timepoint, for 4 timepoints total, with each cell undergoing 8 analyses (4 branch types and 4 mitochondrial measurements). This results in >80 analyses per timepoint. Additionally, the conclusions found in this *in vitro* experiment recapitulate and expand upon our *in vivo* findings (Fig. 1). Nonetheless if the reviewer prefers, we can move this figure to the supplement.

- *Line 216 typo - distal.*

Thank you for bringing this to our attention, we have fixed this typo.

References

- Baldwin, K.T., C.X. Tan, S.T. Strader, C. Jiang, J.T. Savage, X. Elorza-Vidal, X. Contreras, T. Rüdlicke, S. Hippenmeyer, R. Estévez, R.R. Ji, and C. Eroglu. 2021. HepaCAM controls astrocyte self-organization and coupling. *Neuron*. 109:2427-2442.e2410.
- Beardslee, M.A., J.G. Laing, E.C. Beyer, and J.E. Saffitz. 1998. Rapid turnover of connexin43 in the adult rat heart. *Circ Res*. 83:629-635.
- Clavreul, S., L. Abdeladim, E. Hernández-Garzón, D. Niculescu, J. Durand, S.-H. Ieng, R. Barry, G. Bonvento, E. Beaurepaire, J. Livet, and K. Loulier. 2019. Cortical astrocytes develop in a plastic manner at both clonal and cellular levels. *Nature Communications*. 10:4884.
- Kim, H.J., M.R. Shaker, B. Cho, H.M. Cho, H. Kim, J.Y. Kim, and W. Sun. 2015. Dynamin-related protein 1 controls the migration and neuronal differentiation of subventricular zone-derived neural progenitor cells. *Scientific Reports*. 5:15962.
- Laird, D.W., K.L. Puranam, and J.P. Revel. 1991. Turnover and phosphorylation dynamics of connexin43 gap junction protein in cultured cardiac myocytes. *Biochem J*. 273(Pt 1):67-72.
- Liddel, S.A., K.A. Guttenplan, L.E. Clarke, F.C. Bennett, C.J. Bohlen, L. Schirmer, M.L. Bennett, A.E. Münch, W.-S. Chung, T.C. Peterson, D.K. Wilton, A. Frouin, B.A. Napier, N. Panicker, M. Kumar, M.S. Buckwalter, D.H. Rowitch, V.L. Dawson, T.M. Dawson, B. Stevens, and B.A. Barres. 2017. Neurotoxic reactive astrocytes are induced by activated microglia. *Nature*. 541:481-487.
- Stogsdill, J.A., J. Ramirez, D. Liu, Y.-H. Kim, K.T. Baldwin, E. Enustun, T. Ejikeme, R.-R. Ji, and C. Eroglu. 2017. Astrocytic Neuroligins Control Astrocyte Morphogenesis and Synaptogenesis. *Nature*. 551:192-197.

June 18, 2025

RE: JCB Manuscript #202410130R

Cagla Eroglu
Duke University School of Medicine

Dear Cagla,

Thank you for submitting your revised manuscript entitled "Mitochondrial fission controls astrocyte morphogenesis and organization in the cortex". We would be happy to publish your paper in JCB pending final revisions necessary to meet our formatting guidelines (see details below). In your final revision, please address reviewer #3's final concerns with text edits as appropriate and necessary.

A. MANUSCRIPT ORGANIZATION AND FORMATTING:

- 1) Text limits: Character count for Articles is < 40,000, not including spaces. Count includes abstract, introduction, results, discussion, and acknowledgments. Count does not include title page, figure legends, materials and methods, references, tables, or supplemental legends.
- 2) Figures limits: Articles may have up to 10 main text figures.
- 3) Figure formatting: Scale bars must be present on all microscopy images, including inset magnifications. Molecular weight or nucleic acid size markers must be included on all gel electrophoresis. Aspect ratios of images may not be altered.
- 4) Statistical analysis: Error bars on graphic representations of numerical data must be clearly described in the figure legend. The number of independent data points (n) represented in a graph must be indicated in the legend. Statistical methods should be explained in full in the materials and methods. For figures presenting pooled data the statistical measure should be defined in the figure legends. Please also be sure to indicate the statistical tests used in each of your experiments (either in the figure legend itself or in a separate methods section) as well as the parameters of the test (for example, if you ran a t-test, please indicate if it was one- or two-sided, etc.). Also, if you used parametric tests, please indicate if the data distribution was tested for normality (and if so, how). If not, you must state something to the effect that "Data distribution was assumed to be normal but this was not formally tested."
- 5) Abstract and title: The abstract should be no longer than 160 words and should communicate the significance of the paper for a general audience. The title should be less than 100 characters including spaces. Make the title concise but accessible to a general readership.
- 6) Materials and methods: Should be comprehensive and not simply reference a previous publication for details on how an experiment was performed. Please provide full descriptions in the text for readers who may not have access to referenced manuscripts.
- 7) All antibodies, cell lines, animals, and tools used in the manuscript should be described in full, including accession numbers for materials available in a public repository such as the Resource Identification Portal. Please be sure to provide the sequences for all of your primers/oligos and RNAi constructs in the materials and methods. You must also indicate in the methods the source, species, and catalog numbers (where appropriate) for all of your antibodies. Please also indicate the acquisition and quantification methods for immunoblotting/western blots.
- 8) Microscope image acquisition: The following information must be provided about the acquisition and processing of images:
 - a. Make and model of microscope
 - b. Type, magnification, and numerical aperture of the objective lenses
 - c. Temperature
 - d. Imaging medium
 - e. Fluorochromes
 - f. Camera make and model
 - g. Acquisition software
 - h. Any software used for image processing subsequent to data acquisition. Please include details and types of operations

involved (e.g., type of deconvolution, 3D reconstitutions, surface or volume rendering, gamma adjustments, etc.).

10) Supplemental materials: There are strict limits on the allowable amount of supplemental data. Articles may have up to 5 supplemental figures. Please also note that tables, like figures, should be provided as individual, editable files. A summary of all supplemental material should appear at the end of the Materials and methods section.

13) ORCID IDs: ORCID IDs are unique identifiers allowing researchers to create a record of their various scholarly contributions in a single place. Please note that ORCID IDs are now *required* for all authors. At resubmission of your final files, please be sure to provide your ORCID ID and those of all co-authors.

Please note that JCB now requires authors to submit Source Data used to generate figures containing gels and Western blots with all revised manuscripts. This Source Data consists of fully uncropped and unprocessed images for each gel/blot displayed in the main and supplemental figures. For assays performed using capillary electrophoresis and/or immunoassay-based detection, authors should instead provide the electropherogram graph(s) for each experiment, plotting fluorescence/chemiluminescence intensity vs. molecular weight/size. Please be sure to provide one Source Data file for each figure gels, blots, and/or capillary electrophoresis assays along with your revised manuscript files. File names for Source Data figures should be alphanumeric without any spaces or special characters (i.e., SourceDataF#, where F# refers to the associated main figure number or SourceDataFS# for those associated with Supplementary figures). For traditional gels and blots, the lanes of the gels/blots should be labeled as they are in the associated figure, the place where cropping was applied should be marked (with a box), and molecular weight/size standards should be labeled wherever possible. For capillary electrophoresis assays, each trace in the graph should be color-coded and labeled to indicate which protein, gene, or sample is being measured (please try to avoid red/green combinations to accommodate our color-blind readers).

Journal of Cell Biology now requires a data availability statement for all research article submissions. These statements will be published in the article directly above the Acknowledgments. The statement should address all data underlying the research presented in the manuscript. Please visit the JCB instructions for authors for guidelines and examples of statements at (<https://rupress.org/jcb/pages/editorial-policies#data-availability-statement>).

B. FINAL FILES:

****It is JCB policy that if requested, original data images must be made available to the editors. Failure to provide original images upon request will result in unavoidable delays in publication. Please ensure that you have access to all original data images prior to final submission.****

****The license to publish form must be signed before your manuscript can be sent to production. A link to the electronic license to publish form will be sent to the corresponding author only. Please take a moment to check your funder requirements before choosing the appropriate license.****

Thank you for your attention to these final processing requirements. Please revise and format the manuscript and upload materials within 7 days. If you need an extension for whatever reason, please let us know and we can work with you to determine a suitable revision period.

Thank you for this interesting contribution, we look forward to publishing your paper in Journal of Cell Biology.

Sincerely,

Marc Freeman
Monitoring Editor

Andrea L. Marat
Deputy Editor

Journal of Cell Biology

Reviewer #1 (Comments to the Authors (Required)):

In the current manuscript "Mitochondrial fission controls astrocyte morphogenesis and organization in the cortex", Salazar et al., describe a mechanism whereby during the process of astrocyte maturation and morphogenesis, mitochondria increase in number and decrease in size to facilitate organization into the tiny sub compartments of astrocytes (PAPs) which enwrap neuronal synapses. If mitochondrial fission is disrupted by the conditional deletion of Drp1, a protein critical for mitochondrial fission, PAP formation and cortical astrocyte organization is disrupted. Overall, this is a well-written and interesting manuscript that provides novel information related to astrocyte mitochondrial dynamics during early brain development. My concerns were addressed appropriately in the revised manuscript.

Reviewer #2 (Comments to the Authors (Required)):

The authors have thoroughly addressed my comments and concerns. In my opinion, the manuscript is now suitable for acceptance in JCB.

Reviewer #3 (Comments to the Authors (Required)):

First of all, I thank the authors for carefully addressing some concerns raised before. However, the key concern raised by this reviewer is still not appropriately addressed. The data do not robustly support the hypothesis put forth by the authors that "mitochondrial fission controls astrocyte morphogenesis" (see details below). The authors must modify their conclusions and claims according to the data provided, and tone down their unsubstantiated claims. In the current state, the claims in the manuscript and the data don't match.

The hypothesis that mitochondrial fission promotes the extension of astrocyte arbors and formation of fine and distal branches is not supported by the presented data. Specifically, if mitochondrial dynamics are central to astrocyte morphogenesis, why does an increase in mitochondrial number in Mfn1 knockdown (KD) mice not correspond to an increase in astrocyte size? If mitochondrial fission regulates morphogenesis, enhanced fission in Mfn1 KD mice would be expected to proportionally increase

astrocyte size. However, the data show that increased mitochondrial number does not result in expanded astrocyte territory; instead, territory size is reduced (see Fig. 4F). Alternatively, one can argue that altering fission/fusion dynamics may impair mitochondrial health, leading to metabolically poorly supported astrocytes with altered (or atrophied) cell morphology. At least based on the data presented in the manuscript, this reviewer doesn't see a direct relationship between mitochondrial fission and fine astrocyte morphogenesis.

Cx43 was mistakenly referred to as Cx40 in my previous comments, apologies for the typographical error. The connection between Cx43 and Drp1 still remains weak. The authors do not provide compelling evidence that Drp1 deficiency directly affects Cx43 trafficking, which in turn affects astrocyte morphology. Furthermore, reductions in other proteins highly expressed in fine processes (GLAST, Glut1, Kir4.1) suggest that decreased Cx43 may be secondary to a potential loss of perisynaptic astrocytic processes (PAPs) in Drp1 mutant astrocytes. Thus, a claim that Drp1 loss leads to dysregulation of Cx43 and disrupted astrocyte organization is therefore not substantiated. As mentioned above, it is equally plausible that Drp1 deficiency impairs mitochondrial health, induces astrocyte reactivity, and thereby indirectly affects fine branching and associated protein expression. This argument can be further supported by the data presented in this manuscript, that astrocytes become highly reactive in Drp1 cKO mice - as indicated by upregulation of reactivity" markers such as GFAP, LCN2, Cxcl10, and Serpina3n.

Lastly, the authors suggest that in sparse Drp1 KD experiments astrocytes cluster, and in Drp1 cKO mice such clustering is absent. They suggest that "wildtype cells "reject" Drp1 KD astrocytes from their network and thus "push" them away from other wild-type cells into a cluster of KD astrocytes". Again, this hypothesis sounds interesting but has no scientific basis or data supporting it.

Point by point response to Reviewer #3:

We appreciate the reviewer's careful consideration of our manuscript and thoughtful suggestions. Below, we address the reviewer's comments point by point, clarifying our findings and hypotheses, and respectfully explaining why additional experiments fall beyond the scope of this manuscript.

1. The hypothesis that mitochondrial fission promotes the extension of astrocyte arbors and formation of fine and distal branches is not supported by the presented data. Specifically, if mitochondrial dynamics are central to astrocyte morphogenesis, why does an increase in mitochondrial number in Mfn1 knockdown (KD) mice not correspond to an increase in astrocyte size? If mitochondrial fission regulates morphogenesis, enhanced fission in Mfn1 KD mice would be expected to proportionally increase astrocyte size. However, the data show that increased mitochondrial number does not result in expanded astrocyte territory; instead, territory size is reduced (see Fig. 4F). Alternatively, one can argue that altering fission/fusion dynamics may impair mitochondrial health, leading to metabolically poorly supported astrocytes with altered (or atrophied) cell morphology. At least based on the data presented in the manuscript, this reviewer doesn't see a direct relationship between mitochondrial fission and fine astrocyte morphogenesis.

We respectfully clarify that in the Mfn1 KD condition, increased mitochondrial numbers are accompanied by significantly reduced mitochondrial size. This reduction in mitochondrial size is likely to impact astrocyte territory expansion negatively. We have explicitly included this clarification and hypothesis in the results section of our manuscript. However, investigating why mitochondrial fusion deficits influence astrocyte territory size is beyond our study's specific scope, as our primary focus is distinctly on mitochondrial fission. Our data robustly demonstrate that mitochondrial fission through Drp1, distinct from fusion or trafficking dynamics, uniquely regulates astrocyte distal process formation and cortical astrocyte network organization.

The reviewer further suggests an alternative explanation involving impaired mitochondrial health due to altered fission/fusion dynamics underlies the morphological deficiencies we observed. It is plausible that disrupted mitochondrial health could influence astrocyte morphology. However, we have already taken into consideration this possibility in our previous revision. Our evidence strongly supports that impaired mitochondrial fission specifically disrupts morphogenesis of distal astrocyte processes, which is different from the morphological phenotypes observed in Mfn1 and Miro1 KD astrocytes. If a simple explanation of mitochondrial dysfunction is causing disrupted astrocyte morphology, then we would have expected overlapping phenotypes in Drp1, Mfn1 and Miro1 KD astrocytes.

- Why does an increase in mitochondrial number in Mfn1 knockdown (KD) mice not correspond to an increase in astrocyte size?

Mfn1 KD astrocytes do not only have increased mitochondrial numbers; they also have robustly decreased mitochondrial size. This reduction of mitochondrial size in Mfn1 KD astrocytes may negatively impact astrocyte territory outgrowth. **We have added this hypothesis in the results section of our manuscript.**

- If mitochondrial fission regulates morphogenesis, enhanced fission in Mfn1 KD mice would be expected to proportionally increase astrocyte size. However, the data show that increased mitochondrial number does not result in expanded astrocyte territory; instead, territory size is reduced.

Our findings reveal that a *balance* of appropriate levels of mitochondrial fission and fusion is best for astrocyte morphogenesis. As mentioned above, impaired mitochondrial fusion in astrocytes resulted not only in an increased number of mitochondria but also in a significantly reduced average mitochondrial size. It has been extensively shown that overactive fission is detrimental to cell health. Thus, in the absence of Mfn1, which leads to overactive fission, there may be negative impacts on astrocyte development, resulting in reduced astrocyte territory. **We have added this hypothesis to the results section of our manuscript.**

Furthermore, our manuscript focuses on the role of mitochondrial fission in astrocyte development, not mitochondrial fusion. We used manipulation of mitochondrial fusion as a tool to investigate if mitochondrial fission had *unique* effects on astrocyte development or if manipulating other mitochondrial dynamics would similarly affect astrocyte distal process development and astrocyte organization. In our manuscript, we extensively demonstrate that mitochondrial fission, not fusion or trafficking, uniquely affects astrocyte distal process formation and astrocyte network organization. We believe **why mitochondrial fusion affects astrocyte territory size are beyond the scope of the study.**

- Alternatively, one can argue that altering fission/fusion dynamics may impair mitochondrial health, leading to metabolically poorly supported astrocytes with altered (or atrophied) cell morphology. **We have previously added this additional consideration of altered mitochondrial function contributing to our phenotypes in our revised submission of our manuscript both in the results and discussion.** However, even with the consideration of this parallel mechanism at play, our data still clearly show that the astrocyte development phenotypes we discovered are *exclusively* a result of impaired mitochondrial fission, not fusion or trafficking. Therefore, supporting our manuscript title and hypothesis, “Mitochondrial fission controls astrocyte morphogenesis and organization in the cortex”

2. Cx43 was mistakenly referred to as Cx40 in my previous comments, apologies for the typographical error. The connection between Cx43 and Drp1 still remains weak. The authors do not provide compelling evidence that Drp1 deficiency directly affects Cx43 trafficking, which in turn affects astrocyte morphology. Furthermore, reductions in other proteins highly expressed in fine processes (GLAST, Glut1, Kir4.1) suggest that decreased Cx43 may be secondary to a potential loss of perisynaptic astrocytic processes (PAPs) in Drp1 mutant astrocytes. Thus, a claim that Drp1 loss leads to dysregulation of Cx43 and disrupted astrocyte organization is therefore not substantiated. As mentioned above, it is equally plausible that Drp1 deficiency impairs mitochondrial health, induces astrocyte reactivity, and thereby indirectly affects fine branching and associated protein expression. This argument can be further supported by the data presented in this manuscript, that astrocytes become highly reactive in Drp1 cKO mice - as indicated by upregulation of reactivity" markers such as GFAP, LCN2, Cxcl10, and Serpina3n.

We appreciate the reviewer's observations regarding Drp1 deficiency and Cx43. To clarify, our data indicate that Drp1 loss leads to dysregulation of Cx43 protein levels; however, we have

intentionally not claimed a causal relationship between disrupted Cx43 and astrocyte disorganization. Instead, our findings support the conclusion that both astrocyte disorganization and dysregulated Cx43 expression are parallel consequences of Drp1 loss. We have refined the manuscript text to clearly emphasize this distinction and avoid overstating causality.

- The connection between Cx43 and Drp1 still remains weak. The authors do not provide compelling evidence that Drp1 deficiency directly affects Cx43 trafficking, which in turn affects astrocyte morphology.

We show several lines of evidence that Drp1 loss dysregulates Cx43 protein levels. We do not claim that Cx43 disruption in Drp1 KD or cKO astrocytes is the *cause* of astrocyte network disorganization. Rather that both Cx43 dysregulation and astrocyte network disorganization are parallel observed consequences of Drp1 loss in astrocytes. **In the previous revision, we thoroughly modified the text to clarify this data and not overstate a causal role of Cx43 in our findings.**

- As mentioned above, it is equally plausible that Drp1 deficiency impairs mitochondrial health, induces astrocyte reactivity, and thereby indirectly affects fine branching and associated protein expression.

Regarding mitochondrial health and astrocyte reactivity as potential indirect mechanisms, we agree that these are important considerations, and **we have already incorporated them into our previous revision.** Nevertheless, our experimental evidence continues to robustly support that impaired mitochondrial fission uniquely disrupts astrocyte morphogenesis and cortical organization.

3. Lastly, the authors suggest that in sparse Drp1 KD experiments astrocytes cluster, and in Drp1 cKO mice such clustering is absent. They suggest that "wildtype cells "reject" Drp1 KD astrocytes from their network and thus "push" them away from other wild-type cells into a cluster of KD astrocytes". Again, this hypothesis sounds interesting but has no scientific basis or data supporting it. They suggest that "wildtype cells "reject" Drp1 KD astrocytes from their network and thus "push" them away from other wild-type cells into a cluster of KD astrocytes". Again, this hypothesis sounds interesting but has no scientific basis or data supporting it.

We thank the reviewer for highlighting the need for clarity on this point. The reviewer compares two distinct experimental models—sparse Drp1 KD astrocytes versus global Drp1 cKO astrocytes—which inherently differ and are not directly comparable. Nevertheless, astrocyte clustering in Drp1 cKO mice is indeed likely, as demonstrated by our data showing significantly reduced nearest-neighbor distances among astrocytes. In the cKO condition, astrocytes within clusters appear closer, while astrocytes across clusters are further apart, yielding a broader range of distances overall. **We have now explicitly clarified this point in the results section.**

The hypothesis regarding wild-type cells excluding Drp1 KD astrocytes into clusters was proposed solely in response to the reviewer's direct query during revision discussions. We emphasize this hypothesis was exploratory, not experimentally tested, and importantly, not included in the manuscript.